# Transient histone deacetylase inhibition induces cellular memory of gene expression and 3D genome folding

Flora Paldi [1], Michael-Florian Szalay [1,3], Solène Dufau[1,3], Marco Di Stefano [1,3], Hadrien Reboul[1], Daniel Jost [2], Frédéric Bantignies [1] & Giacomo Cavalli [1]✉

Epigenetic memory enables the propagation of gene expression patterns following transient stimuli. Although three-dimensional chromatin organization is emerging as a key regulator of genome function, it is unknown whether it contributes to cellular memory. Here we establish that acute perturbation of the epigenome can induce cellular memory of gene expression in mouse embryonic stem cells. We uncover how a pulse of histone deacetylase inhibition translates to changes in transcription, histone modifications and genome folding. While most epigenomic and transcriptional changes are initially reversed once the perturbation is removed, some loci remain transcriptionally deregulated and genome architecture partially maintains its perturbed conformation. Consequently, a second pulse of transient hyperacetylation induces stronger memory of transcriptional deregulation. Using ultradeep Micro-C, we associate memory of gene expression with repressive Polycomb-mediated chromatin topology. These results demonstrate how cells can record transient stresses in their genome architecture, thereby enabling an enhanced response to subsequent perturbations.

Cellular identity is established by gene regulation and epigenetic mechanisms that shape the transcriptional landscape. The information to maintain transcriptional programs is stored in alternative chromatin states that provide means for cellular plasticity to respond to developmental and environmental cues. This is particularly true for embryonic stem cells (ESCs) that have distinctive permissive chromatin where activating and repressive configurations often co-exist[1]. These bivalent or poised states are mainly established around developmental genes and require a fine balance between opposing signals to keep gene expression sufficiently low but also prime genes for future activation[2,3]. An important feature of functional chromatin states is the ability to convert short-lived signals to long-lived changes in gene expression—a concept commonly referred to as cellular memory[4], that is, a sustained cellular response to a transient stimulus. Cellular memory is widely accepted as an important aspect underlying development and often involves a complex interplay among different epigenetic layers to stabilize gene expression programs after cellular state transitions[5]. However, the crosstalk between epigenetic mechanisms and the extent to which they can contribute to memory has been difficult to study due to functional redundancy between components of the epigenetic machinery, as well as the lack of experimental approaches that uncouple gene regulation from cellular memory.

In this study, we sought to understand the dynamics of the epigenome during a short-lived disruption of chromatin state balance. To this end, we pulsed mouse ESCs (mESCs)[6] with the histone deacetylase (HDAC) inhibitor trichostatin A (TSA)[6], which has rapid, global, yet reversible effects on histone acetylation. Using a combination of RNA sequencing (RNA-seq), chromatin immunoprecipitation followed by

[1]Institute of Human Genetics (IGH), University of Montpellier, CNRS, Montpellier, France. [2]Laboratoire de Biologie et Modélisation de la Cellule, Ecole Normale Supérieure de Lyon, CNRS UMR 5239, Inserm Université Claude Bernard Lyon, Lyon, France. [3]These authors contributed equally: Michael-Florian Szalay, Solène Dufau, Marco Di Stefano. ✉e-mail: giacomo.cavalli@igh.cnrs.fr

sequencing (ChIP–seq) and ultradeep Micro-C, we investigate the interplay among gene expression, histone modifications and genome folding. We place particular emphasis on three-dimensional (3D) genome folding, which is emerging as a key contributor to cellular identity through its role in gene expression control[7], and also uncover a link between 3D genome organization and cellular memory.

## Results

### HDAC inhibition alters the epigenome and the transcriptome

To disrupt the chromatin state balance of mESCs, we pulsed them with TSA for 4 h (Fig. 1a). To minimize pleiotropic effects, we optimized treatment conditions so that histone hyperacetylation is strongly induced, but the bulk of the acetylome and cell cycle progression remain minimally affected (Extended Data Fig. 1a–f). Calibrated ChIP–seq indicated that acute TSA treatment induced genome-wide H3K27 hyperacetylation (Fig. 1b and Extended Data Fig. 1g) that occurred ubiquitously around *cis*-regulatory elements, along gene bodies and in intergenic regions (Extended Data Fig. 1h).

As functional chromatin states are maintained by an interplay between active and repressive histone modifications[5,8], we characterized secondary changes in the histone landscape. By categorizing genomic intervals as active (enriched in H3K27ac, H3K4me1 or H3K4me3) and repressive (enriched in H3K9me3, H3K27me3 or H2AK119ub), we found that TSA treatment caused a larger fraction of the genome to be in an active state (Extended Data Fig. 1i). We identified differential peaks for most histone modifications, with activating marks in general gaining signal (Fig. 1c and Extended Data Fig. 1h). Gene annotation of differential ChIP–seq peaks pointed to an amplification of developmental processes and a suppression of pluripotency (Fig. 1d).

Next, we performed bulk RNA-seq to discern whether the effects of HDAC inhibition on the transcriptome corroborate change in the histone landscape. As HDAC1 is an important regulator of early development[9,10], we focused on transcriptional changes related to ESC identity. As suggested by the re-organization of the histone modification landscape, downregulated genes were associated with stem cell population maintenance (Fig. 1e,f), while upregulated genes were linked to the developmental maturation of the neural lineage, as previously described[10–13].

Thus, acute HDAC inhibition leads to widespread accumulation of H3K27ac and global changes in the histone modification landscape that promote a gene expression program associated with exit from pluripotency.

### Global and local architectural changes mark the TSA chromatin state

Imaging studies have shown that TSA treatment leads to chromatin decompaction both at global[14,15] and local[15,16] scales. To understand how these alterations translate to changes in chromatin contacts, we generated ultradeep Micro-C maps with 8.5 and 6.6 billion interactions for control (dimethyl sulfoxide (DMSO)) and TSA conditions, respectively. Our datasets show uniform genomic coverage over both active and inactive genomic intervals[17,18], providing unbiased genome-wide interaction maps (Extended Data Fig. 2a,b). Unlike recent high-resolution capture studies, we do not observe microcompartments at previously described loci[19] (Extended Data Fig. 2c,d), but it remains possible that microcompartments would emerge upon deeper sequencing.

When we compared contact matrices generated from control and TSA-treated cells, we detected a dramatic increase in *trans* contacts upon TSA treatment (Extended Data Fig. 2e) concomitantly to a marked decrease in *cis*-interactions (Extended Data Fig. 2f). TSA treatment also led to a loss of prominent A compartment interactions—characteristic of ESCs[20]—without major changes in compartment identity (Fig. 2a and Extended Data Fig. 2g,h). Conversely, B–B interactions—feature of differentiated cells[20]—became prominent in *cis* (Fig. 2a). This prompted us to examine if A and B compartments were asymmetrically impacted by

TSA treatment. Indeed, the gain in activating histone marks in A compartment exceeded that in B compartment, and most gene expression deregulation corresponded to transcription start sites (TSSs) located in the A compartment (Fig. 2b,c and Extended Data Fig. 2i).

It has been previously suggested that elevated transcriptional activity[21,22] or acetylation level[22,23] increases the stiffness of the chromatin fiber, leading to an increase in *trans*-interactions. We tested this idea by simulating TSA-induced changes using mechanistic 3D polymer modeling[24]. First, we used a single-chain block copolymer model (Fig. 2d) made of A and B chromatin regions to infer self-attraction energies ($E_{AA}$, $E_{BB}$) that best reproduce the A/B compartment strength in DMSO condition (Extended Data Fig. 2j,k and Supplementary Methods). Because A and B compartments are asymmetrically affected by TSA (Fig. 2c), we simulated the effect of TSA treatment by changing the stiffness differentially within A and B domains in a multichain system. By optimizing the values of stiffness to match the median *trans*-interaction ratio (Fig. 2e and Extended Data Fig. 2l–n), the TSA conformation was compatible with a greater increase of stiffness in A domains than in B domains ($K_\theta^A = 18.0\,K_\theta$ versus $K_\theta^B = 3.0\,K_\theta$). Interestingly, the change in stiffness we applied to reproduce the *trans*-contact ratio predicted the swap in compartment strength that we observed experimentally, without the need of modifying A–A and B–B attraction energies (Fig. 2f and Extended Data Fig. 2m). Additionally, our model predicted a peripheral displacement of A domains in the simulated nucleus[25] (Extended Data Fig. 2o,p). This was confirmed by H3K27ac immunofluorescence that indicated increased proximity of H3K27ac foci to the nuclear periphery in TSA-treated cells (Extended Data Fig. 2q,r).

Subsequently, we characterized the changes in chromatin looping. We identified >3,000 focal interactions with differential loop strength (Fig. 2g–i) that were strongly associated with developmental loci (Fig. 2j and Extended Data Fig. 2s). As we detected ~3,500 loci with increased CTCF binding in TSA (Extended Data Fig. 2t), we tested if looping increased at CTCF-bound anchor sites. However, we found that CTCF-mediated loops globally became weaker. Instead, non-CTCF loops—carrying either active (H3K27ac, H3K4me1) or repressive (H3K9me3, H3K27me3) chromatin signatures—became stronger upon TSA treatment (Fig. 2k,l).

To rule out potential off-target effects of the nonselective HDAC inhibitor TSA, we assayed changes in gene expression and chromatin architecture induced by the nuclear HDAC inhibitor, romidepsin[26] (Extended Data Fig. 3a). This revealed a nearly identical transcriptional response and similar changes in 3D genome folding among the two inhibitors (Extended Data Fig. 3b–h).

Taken together, TSA treatment promotes interchromosomal contacts and decreases A–A compartment interactions. Such changes are compatible with a compartment-specific increase of chromatin stiffness in biophysical modeling simulations. In parallel, TSA treatment causes fine-scale restructuring where CTCF-dependent and epigenetic-state-driven loops behave differently. Notably, annotation of differential looping sites mirrors ongoing developmental processes identified from transcriptomic and histone modification changes.

### Epigenomic and architectural changes govern gene expression changes

We then asked what epigenomic changes underlie TSA-induced gene expression deregulation. Upregulated TSSs were enriched for H3K27ac peaks that gained signal in TSA, whereas peaks with decreased H3K27ac signal were more frequently found at downregulated TSSs (Fig. 3a). Gain in H3K27ac was concomitant with a modest increase in chromatin accessibility and a substantial gain of H3K4me1 (Fig. 3b). We detected overlapping H3K4me3 and H3K27me3 signals at a large subset of upregulated genes, implying that bivalent genes are susceptible to TSA-mediated gene derepression, without the loss of H3K27me3 (Fig. 3c and Extended Data Fig. 4a).

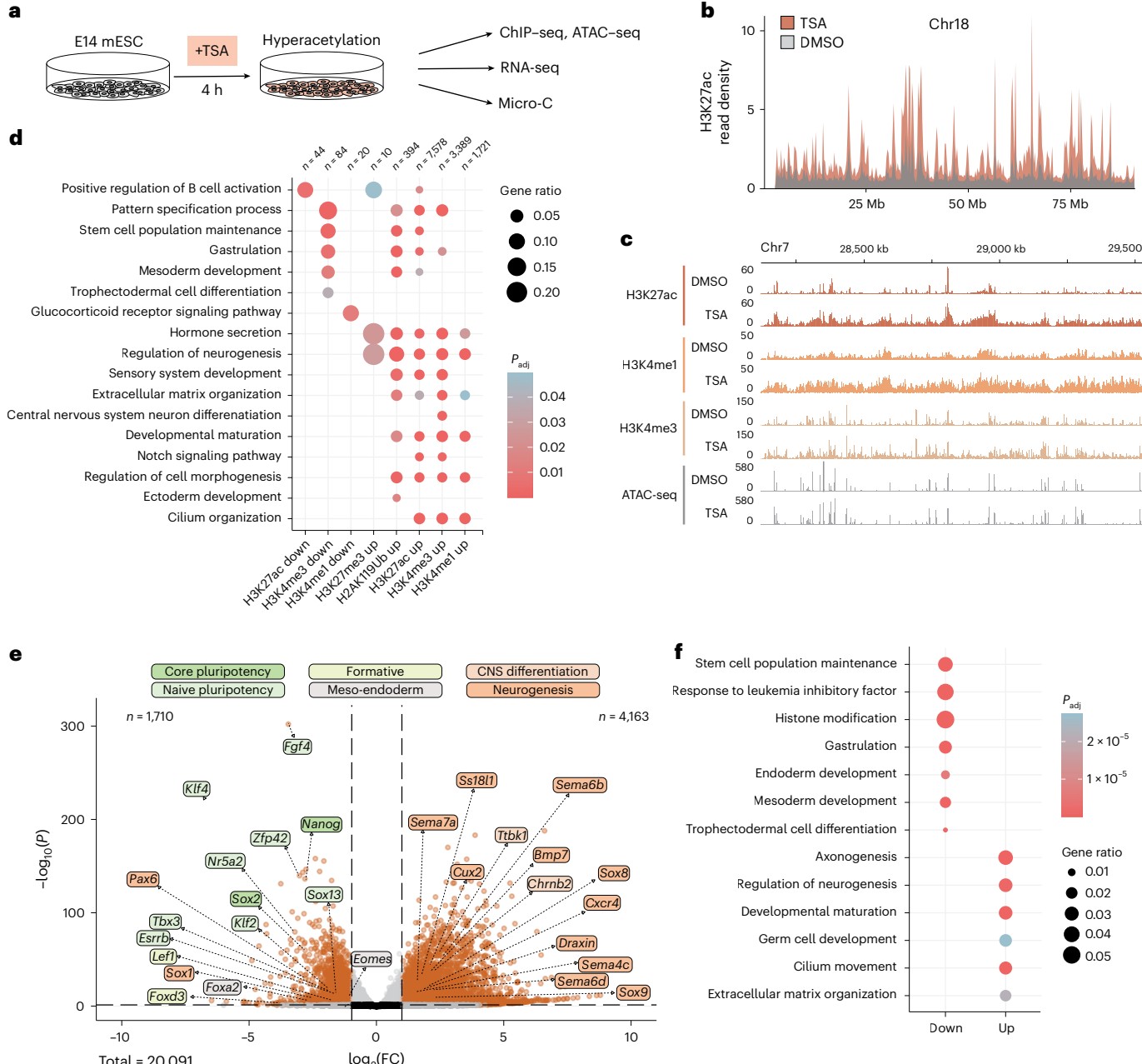

**Fig. 1 | Acute HDAC inhibition leads to global changes in the histone landscape and gene expression. a**, E14 mESCs were pulsed for 4 h with TSA and assayed for changes in chromatin states by multi-omics. **b**, Plot showing normalized H3K27ac ChIP–seq read density on chromosome 18 (bin size = 10 kb). **c**, Genomic snapshot of spike-in normalized ChIP–seq and ATAC–seq signal over a typical hyperacetylated region. **d**, GO enrichment of terms related to development among genes located within 1-kb distance from differential ChIP–seq peaks. *P* values were derived from one-sided Fisher's exact test and adjusted to FDR with the BH method. **e**, Volcano plot showing

differential gene expression (significance cutoffs (dashed lines)−adjusted *P* value (derived from Wald test, corrected for multiple testing with BH approach) <0.05, absolute log$_2$(FC) > 1) upon TSA treatment. Labels correspond to core and naive pluripotency, formative, meso-endodermal, CNS differentiation and neurogenesis marker genes. **f**, GO enrichment of terms related to development among upregulated and downregulated genes. Adjusted *P* values were derived as in **d**. BH, Benjamini−Hochberg; FDR, false discovery rate; FC, fold change; CNS, central nervous system.

To understand if transcriptional upregulation could be caused by ectopic enhancer activation, we examined the linear proximity of genomic regions that gain H3K4me1 to TSSs. This revealed that such peaks form closer to upregulated gene promoters than promoters of expression-matched control genes that do not undergo upregulation (Fig. 3d). Similarly, upregulated TSSs were found to be in closer proximity to previously described primed (H3K4me1$^+$, H3K27ac$^-$) and poised (H3K4me1$^+$, H3K27me3$^+$, H3K27ac$^-$) enhancers[27,28] (Extended

Data Fig. 4b). As enhancer activation and subsequent gene expression are often accompanied by increased enhancer−promoter (E−P) contacts[20,29,30], we analyzed changes in chromatin looping at upregulated TSSs. Although we detected an increased linear proximity of loops to upregulated TSSs, gene expression upregulation occurred without changes in promoter contacts (Fig. 3e,f).

Interestingly, active marks were also gained in the vicinity of downregulated TSSs, without the accumulation of repressive histone

modifications (Fig. 3b,c). However, closer inspection of differential ChIP–seq peak distribution revealed that regions that lose signal enrichment for activating histone marks (H3K27ac, H3K4me3) are more frequently located near downregulated TSSs (Fig. 3a and Extended Data Fig. 4d), indicating functional coupling between epigenomic and transcriptomic changes. Sites of increased H2AK119ub signal were found equally around both upregulated and downregulated TSS, consistent with recent evidence implicating H2AK119ub in both gene repression and activation[31–33]. A subset of downregulated TSSs were found to be strongly enriched for Myc and YY1 binding (Extended Data Fig. 4c), transcriptional regulators that are HDAC targets and whose acetylation state can modulate their molecular function[34,35]. This suggests that TSA-induced gene downregulation may be partially due to effects on nonhistone targets of HDACs. Unlike upregulated TSSs, promoter loops around downregulated TSSs became stronger (Fig. 3f). We also noticed the formation of prominent de novo loops around some of the most strongly downregulated genes, among existing H3K9me3 sites, with only a mild increase in H3K9me3 level (Extended Data Fig. 4e–i). These suggest that repressive chromatin contacts could represent a widespread molecular process linked to gene downregulation.

Altogether, TSA-induced gene upregulation occurs with the accumulation of activating signals, whereas gene downregulation occurs without the gain of repressive chromatin marks. Instead, while gene upregulation is potentially associated with enhancer overactivation without changes in E–P contacts, downregulation is linked to repressive chromatin looping.

## ESCs recover transcriptional and chromatin states after TSA removal

We then postulated that if the perturbation induced cellular memory, chromatin and gene expression changes should outlast the initial causative event. To this end, we washed TSA-treated cells and let them recover for 24 h (about two cell doubling times; Fig. 4a). After 24 h of TSA removal protein acetylation was restored to the unperturbed level (Extended Data Fig. 1a–c and Extended Data Fig. 5a), and cell generation tracing showed that all cells have divided during the recovery period (Extended Data Fig. 5b). The effects of TSA on the histone landscape were likewise readily reversible (Extended Data Fig. 5c–g). Excess H3K27ac, H3K4me1 and chromatin accessibility were restored at once (Fig. 4b) and H3K27ac peaks regained their enrichment around TSSs (Fig. 4c), with the exception of H3K9ac, where quantitative analysis indicated mild persistent enrichment at several genomic loci (Extended Data Fig. 5h–j). Consistent with the restoration of chromatin marks, TSA-induced transcriptional deregulation was nearly completely reversed with only few genes (n = 164) showing residual deregulation (Fig. 4d). Although certain genomic loci retained

slightly increased H3K27ac, they did not strongly colocalize with TSSs that remained deregulated (Fig. 4e). Altogether, these data show that histone marks and gene expression are generally restored 24 h after TSA removal, but a minor fraction of the genes maintains a memory of the perturbation.

To exclude the possibility that the near-complete recovery is due to an insufficiently long recovery, we assayed gene expression changes 48, 72 and 96 h after the TSA removal. Interestingly, the number of differentially expressed genes increased with longer recovery times (Extended Data Fig. 6a–d). Conversely, assessing transcription at shorter recovery times (16 and 20 h) showed that gene expression changes were the lowest at 24 h. Crucially, we did not find any link between nonhistone targets of HDACs and sustained gene expression deregulation (Extended Data Fig. 6e–h). Although we cannot fully exclude it, this minimizes the possibility that long-term transcriptional consequences would be due to pleiotropic effects of HDAC inhibition.

After 24-h recovery, pluripotency network activity was efficiently restored, and most developmental processes were downregulated again. As mESC culture conditions actively suppress differentiation, we asked whether efficient recovery is a general feature of pluripotent cells, or whether it results from growth conditions that impose dominant cell states. Thus, we grew mESCs into gastruloids[36,37], which we treated with TSA for 4 h immediately before the Chiron pulse (Fig. 4f and Extended Data Fig. 7a,b). The effects of TSA on the transcriptome strongly correlated between mESCs and gastruloids (Fig. 4g), with Gene Ontology (GO) enrichment analysis showing similar developmental deregulation in both conditions (Extended Data Fig. 7c). After washes—which restored H3K27ac within 24 h (Extended Data Fig. 7a,b,d)—we let control and TSA-treated gastruloids develop for 3 days. Although the area staining positive for the neuroectodermal marker Sox2 expanded in TSA-treated gastruloids, the transcriptome was largely re-established (Fig. 4h,i and Extended Data Fig. 7e). To enhance the relevance of our findings beyond pluripotent cells, we tested the ability of neural progenitor cells (NPCs) to recover from a TSA pulse. This revealed that, similarly to ESCs, transcriptional differences were small after 24-h TSA removal and increased at 48 h (Extended Data Fig. 7f–h). While the transcriptional response to TSA is partially linked to similar regulatory pathways in ESCs and NPCs, cell-type-specific differences exist (Extended Data Fig. 7i).

In sum, mESCs possess a remarkable capacity to recover their transcriptional and histone modification landscapes following a hyperacetylation pulse. Our findings in gastruloids and NPCs support these observations, namely that the effect of HDAC inhibition on the transcriptome is profound, but transcriptional recovery from it is efficient. Nevertheless, the data also indicate that cells maintain a partial memory of the TSA pulse.

**Fig. 2 | Global and fine-scale architectural changes characterize the TSA chromatin state. a**, Aggregate plots of homotypic interactions between A and B compartments in *cis*. **b**, Metaplots showing normalized H3K27ac ChIP–seq read density over A and B compartments (bin size = 1 kb). Shading represents s.d. of the mean. **c**, Distribution of H3K27ac ChIP–seq reads (left) and upregulated and downregulated TSSs (right) by compartment. **d**, Schematic representation of biophysical modeling where $E_{AA}$ and $E_{BB}$ correspond to the attraction energies and $K_θ^A$ and $K_θ^B$ correspond to the stiffness of the chromatin fiber, in A and B domains, respectively. Each chromosome was modeled by a 20-Mb chain with beads representing 5-kb DNA. A and B domains were set at 1.5 Mb in size to match the mean compartment size derived from the Micro-C data, resulting in six A domains, six B domains and two telomeric regions of 1 Mb at the extremities of the chain. A nucleus was modeled using 20 chains. **e**, *Trans*-contact ratio in DMSO and TSA in the Micro-C data (n = 19 chromosomes) and in the model (n = 600 corresponding to 20 simulated chains in 30 replicates). Box plots show median (central line), the 25th and 75th percentiles (box limits) and 1.5× IQR (whiskers). Outliers are not shown. **f**, Compartment strength in DMSO (left) and TSA (right) in the Micro-C data and in the model (n = 150 corresponding to number of values in average compartment profiles). Box plot elements are as in **e**. **g**, Volcano

plot of differential loops between DMSO and TSA (significance cutoffs (dashed lines)—adjusted *P* value (derived from Wald test, corrected for multiple testing with BH approach) <0.05, absolute FC > 1.5). Positive log$_2$(FC) indicates stronger interaction in TSA. **h**, Micro-C contact maps showing differential looping at the *Bcar1*, *Zfp462* and *Tbx3* loci. **i**, Aggregate plots of Micro-C signal around differential loops at 4-kb resolution. **j**, GO enrichment among genes closest to differential loop anchors. *P* values correspond to one-sided Fisher's exact test corrected for FDR with the BH method. **k**, Pile-ups showing Micro-C signal around all loops identified in DMSO (top) and TSA (bottom) (resolution = 4 kb). Quantification of aggregate loop signal (n = 9 corresponding to the central 3 × 3 pixels) is shown on the right (paired two-tailed *t* test; **P < 0.01). Box plot elements are as in **e**. **l**, Pile-ups of Micro-C signal around loops stratified by the presence (CTCF loops) or absence (non-CTCF loops) of CTCF ChIP–seq peaks at loop bases (resolution = 4 kb). Non-CTCF loops have been further divided into active and repressive based on the presence of activating (H3K27ac, H3K4me1) or repressive (H3K27me3, H3K9me3) ChIP–seq signal at loop bases. Quantification of aggregate loop strength (n = 9 corresponding to the central 3 × 3 pixels) is shown on the right (paired two-tailed *t* test; *P < 0.05, ***P < 0.001). Box plot elements are as in **e**. IQR, interquartile range; Res, resolution.

## Genome architecture retains partial memory of its past conformation

To further explore whether the TSA pulse could be recorded by mESCs, we analyzed 3D genome folding after recovery. Surprisingly, we found that chromatin conformation did not fully recover—the *cis–trans* ratio was partially restored (Extended Data Fig. 8a) and *cis*-contact depletion persisted, particularly in the A compartment (Extended Data Fig. 8b). Additionally, while A–A interactions were efficiently

recovered in *trans* and showed some increase in *cis*, B–B interactions in *cis* remained prominent (Fig. 5a). We could equally detect sustained changes in genome conformation at the gene level. High-resolution eigenvector decomposition[38] revealed local instances where transcriptional and architectural recovery became uncoupled. For example, gene expression upregulation at the *F11*, *Klkb1* and *Cyp4v3* loci in TSA shifted the ~200-kilobase (kb) encoding genomic segment to the A compartment. After recovery, repression of the genes was

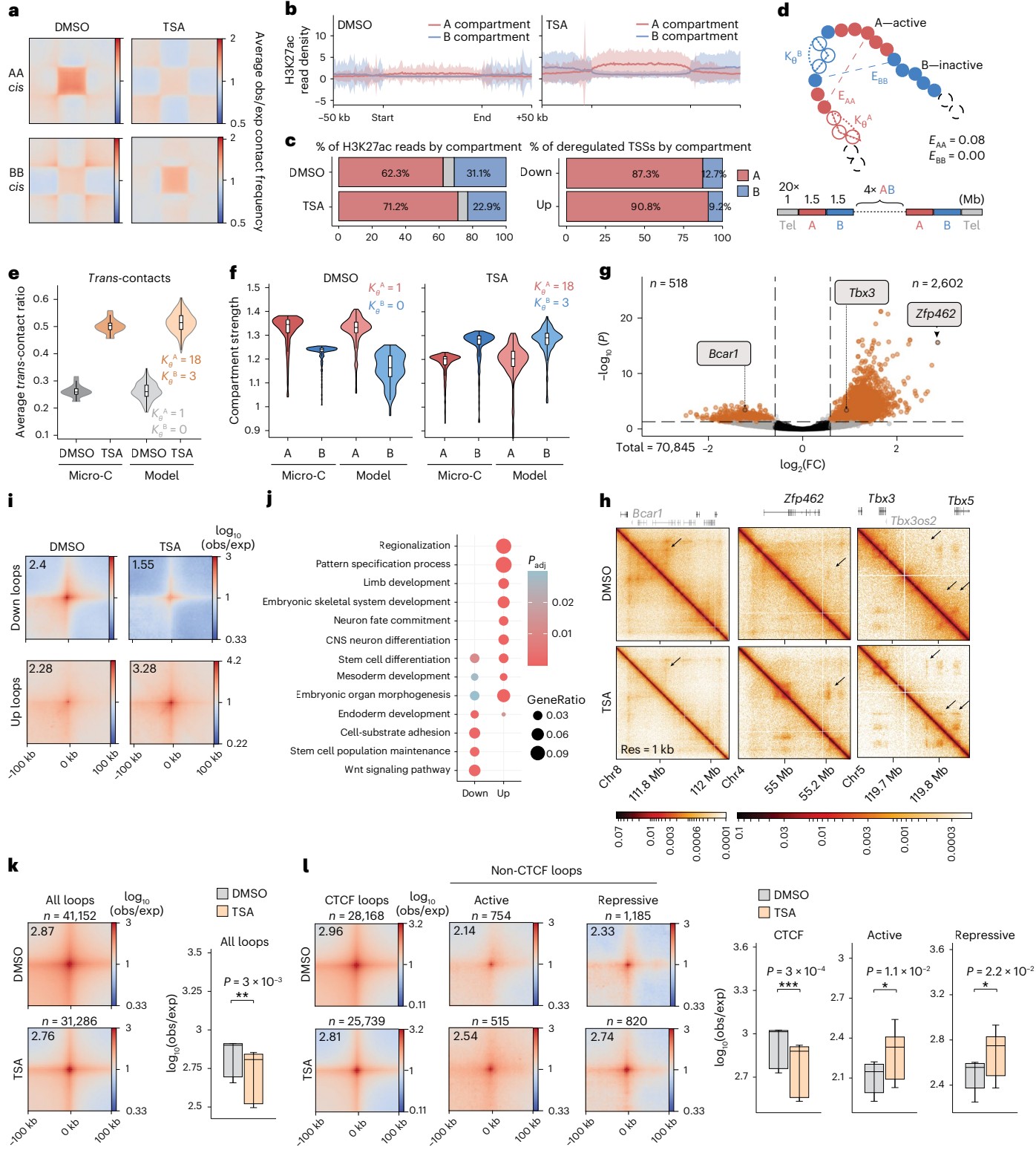

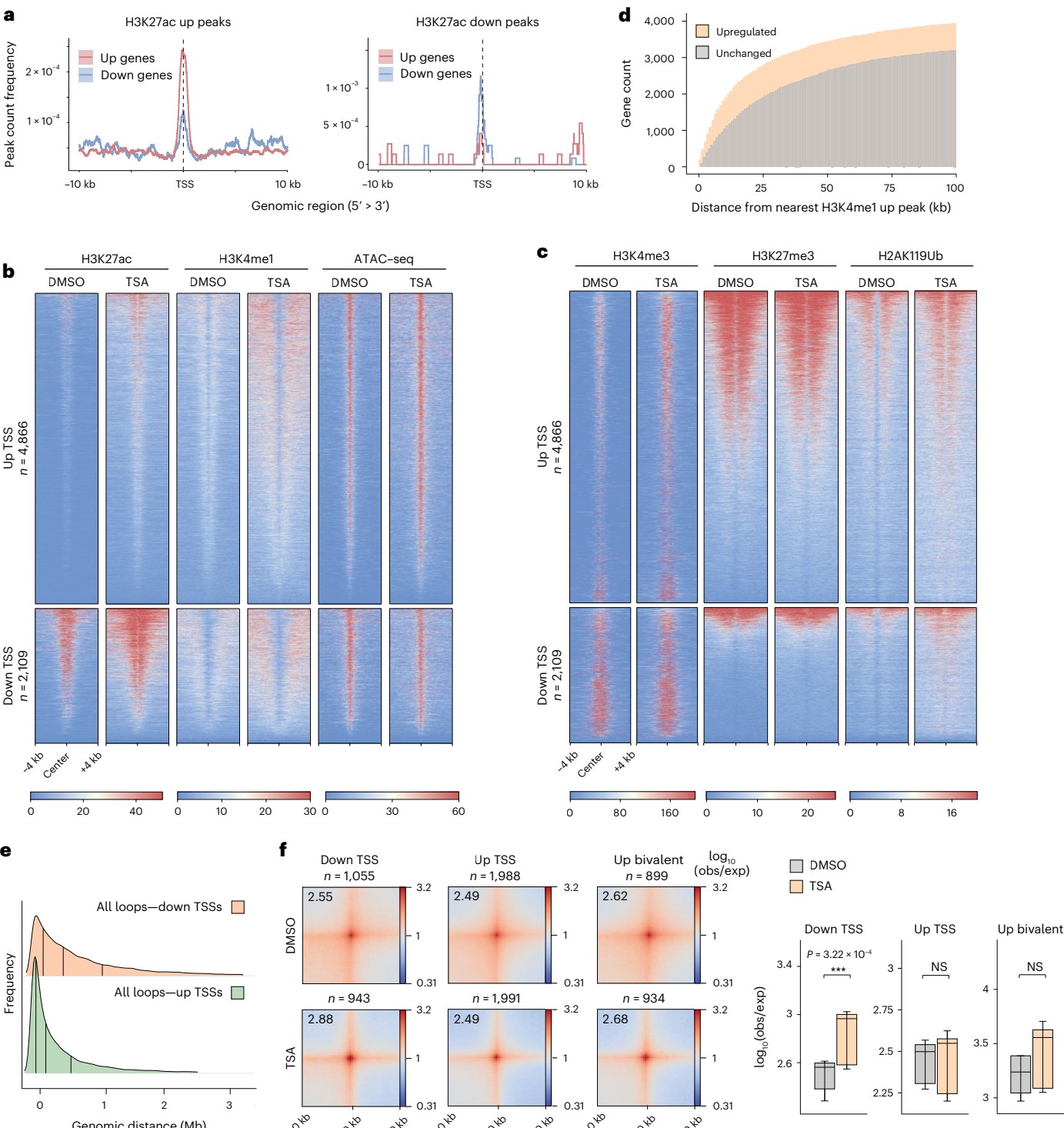

**Fig. 3 | Changes in histone landscape and chromatin looping underlie differential gene expression. a**, H3K27ac TSA up (left) and down (right) peak-count frequency distributions relative to TSSs of upregulated and downregulated genes in TSA. **b,c**, Heatmaps showing normalized H3K27ac, H3K4me1 and ATAC–seq signal (**b**) or H3K4me3, H3K27me3 and H2AK119ub signal (**c**) around TSSs of differentially expressed genes. **d**, Cumulative histogram showing genomic distance between upregulated gene promoters and the nearest increased H3K4me1 peak. Control genes represent an expression-matched gene set that does not increase in expression. **e**, Ridge plot showing the frequency of loop anchors in the function of genomic distance from the nearest deregulated TSS. **f**, Aggregate plots of Micro-C signal around loops where anchors overlap with downregulated (left) or upregulated (middle) TSSs, as well as bivalent (right) TSSs that undergo upregulation (resolution = 4 kb). Quantification of piled-up loop signal ($n = 9$ corresponding to the central $3 \times 3$ pixels) is shown on the right (paired two-tailed $t$ test; NS > 0.05, ***$P < 0.001$). Data shown are the median, with hinges corresponding to IQR and whiskers extending to the lowest and highest values within 1.5× IQR.

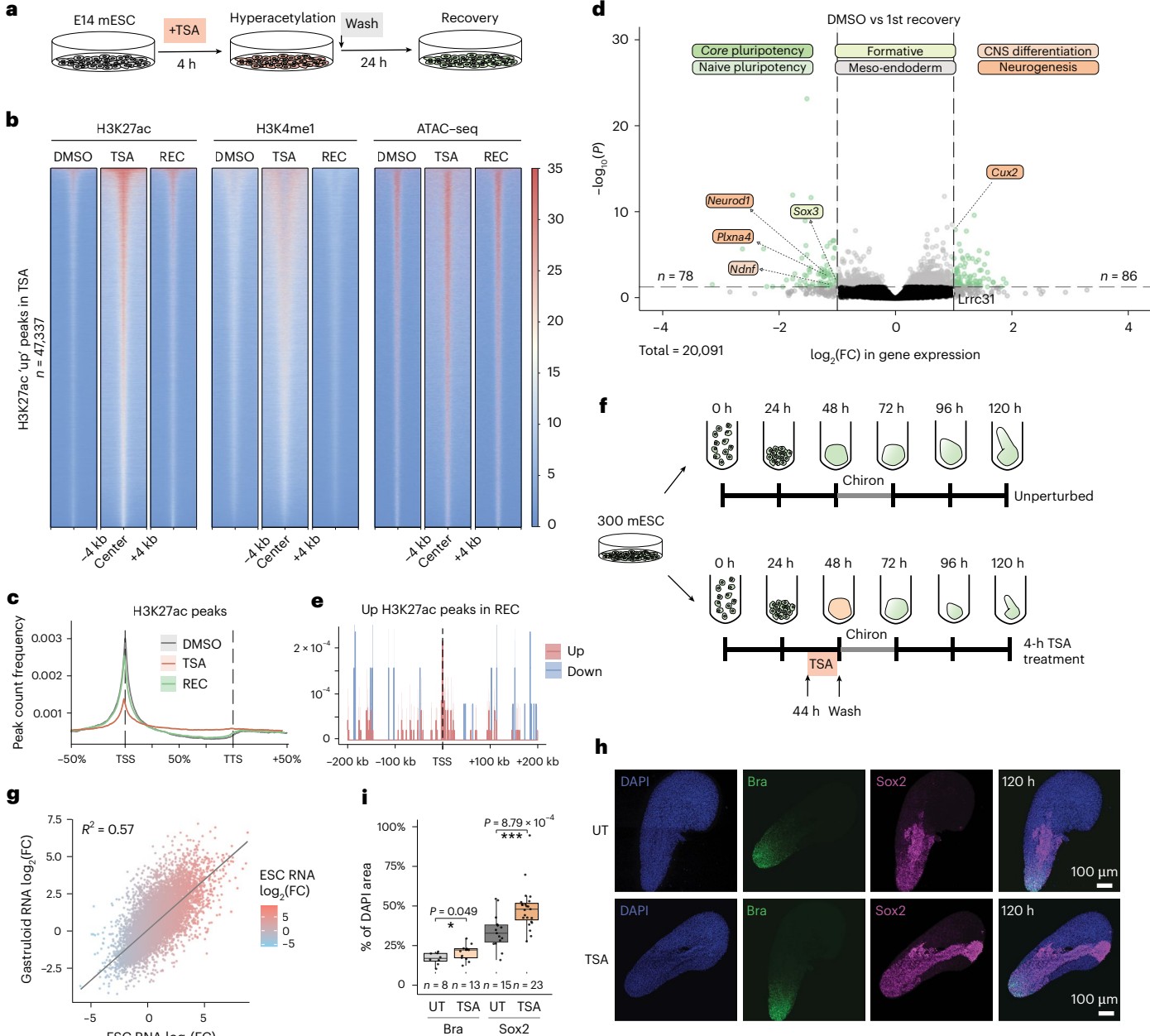

**Fig. 4 | Pluripotent cells recover their transcriptional identity and chromatin states following the removal of HDAC inhibition. a**, TSA-treated mESCs were washed and re-assayed for their chromatin states 24 h later. **b**, Heatmaps of H3K27ac, H3K4me1 and ATAC–seq signal in DMSO, TSA and REC at differential H3K27ac peaks identified in TSA. **c**, H3K27ac peak-count frequency distribution in DMSO, TSA and REC relative to gene bodies of all genes. **d**, Volcano plot showing differential gene expression 24 h after TSA washout. Labels show developmental marker genes that remain above the significance thresholds (dashed lines, adjusted $P$ value (derived from Wald test, corrected for multiple testing with BH approach) <0.05, absolute $\log_2(FC)$ > 1). **e**, H3K27ac up peak-count frequency distribution in REC relative to gene bodies that remain upregulated and downregulated. **f**, Schematics of TSA treatment between

44 h and 48 h and subsequent washout during gastruloid differentiation. **g**, Scatterplot showing the correlation of TSA-induced transcriptomic changes in mESCs and in 48-h gastruloids. Fitted line shows linear regression. **h**, Representative images of Sox2 and Bra immunostaining in 120-h untreated gastruloids (top) or gastruloids 3 days following transient TSA treatment (bottom). Scale bar = 100 μm. **i**, Quantification of Bra and Sox2 area over DAPI in 120-h gastruloids with or without TSA treatment (*n* corresponds to number of gastruloids measured). Data shown are the median, with hinges corresponding to IQR and whiskers extending to the lowest and highest values within 1.5× IQR (unpaired two-tailed *t* test; *$P$ < 0.05, ***$P$ < 0.001). TTS, transcription termination site; REC, recovery; Bra, Brachyury.

successfully restored; however, the encoding genomic segment maintained A compartment identity, like in the TSA condition (Fig. 5b). Additionally, incomplete architectural recovery was visible at certain genomic loci where increased loop strength was maintained throughout the recovery period (Fig. 5c and Extended Data Fig. 8c) without any persisting changes in histone modifications. Finally, we found that while differential loops that lost strength in TSA were fully restored, loops that became stronger upon TSA treatment remained enhanced (Fig. 5d).

In sum, genome architecture carries a memory of its TSA-induced conformation that is visible at the level of *cis*-contact frequencies, compartment interactions and chromatin loops.

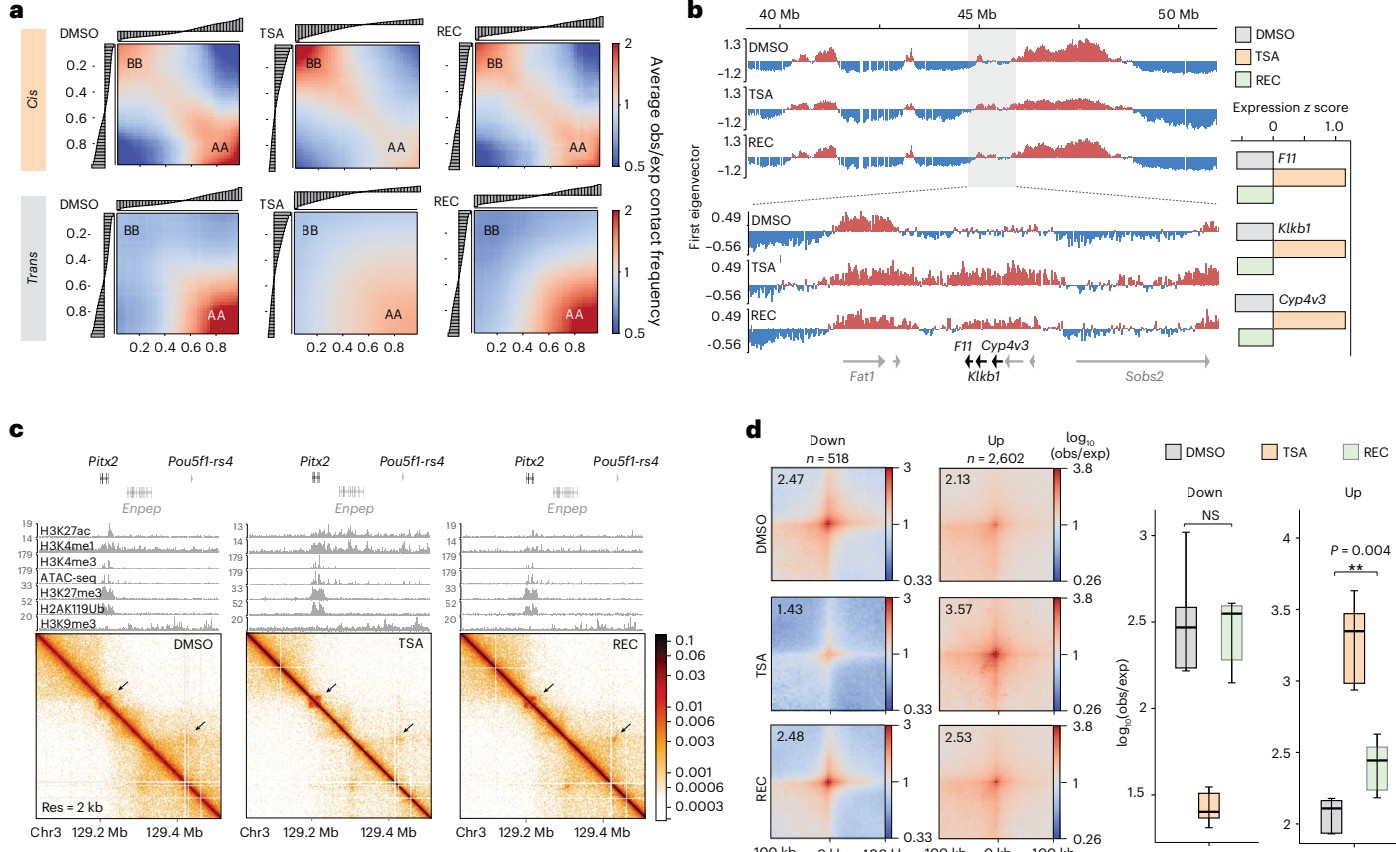

**Fig. 5 | Genome architecture retains partial memory of the past conformation. a**, Saddle plots of compartment interactions in *cis* (top) and in *trans* (bottom) in DMSO, TSA and REC Micro-C. **b**, High-resolution eigenvector tracks of the Micro-C data showing small-scale compartment switch around the *F11–Klkb1–Cyp4v3* that persists in recovery. Gene expression *z* score is shown on the right. **c**, Micro-C maps at the *Pitx2* locus showing incomplete architectural recovery.

Normalized ChIP–seq tracks of the corresponding condition are shown above. **d**, Pile-up of Micro-C signal around differential loops in DMSO, TSA and recovery (resolution = 4 kb). Quantification of piled-up loop signal (*n* = 9 corresponding to the central 3 × 3 pixels) is shown on the right (paired two-tailed *t* test; NS > 0.05, **\**P* < 0.01). Data shown are the median, with hinges corresponding to IQR and whiskers extending to the lowest and highest values within 1.5× IQR.

## Lasting gene expression changes are linked to regulatory 3D contacts

We then reasoned that if the persisting minor architectural and transcriptional changes signified cellular memory, then repeated exposure to TSA should have more severe consequences. Therefore, we subjected mESCs to a second cycle of TSA treatment and recovery (Fig. 6a). While the effects of the second TSA treatment were comparable to the first (Fig. 6b and Extended Data Fig. 8d,e), recovery from the second treatment was less complete (Fig. 6c). Hundreds of genes remained strongly deregulated (*n* = 767) and showed association with developmental processes, suggesting that repeat exposure had a greater impact on cellular identity (Fig. 6d). We note that this might be partially due to increasing gene expression deregulation with longer times following the first TSA pulse (Extended Data Fig. 6b). Next, we stratified differentially expressed genes based on their abilities to recover from either TSA treatments, and we plotted their expression over the double treatment course. While most deregulated genes oscillated between their native and ectopic expression states, a subset of genes showed progressively aggravating gene expression deregulation (Fig. 6e), indicative of cellular memory. Additionally, transcriptional response and recovery were similar when the first recovery period was increased from 24 to 48 h (Extended Data Fig. 8f–h), implying that the cellular memory effect can persist through multiple cell generations.

We aimed to identify chromatin features that distinguish genes that recover from those that retain memory of their TSA-induced expression state. First, we analyzed histone acetylation and chromatin

accessibility around upregulated TSSs. This showed no difference between recovered genes and genes that did not recover—both groups were characterized by a gain of activating chromatin signals that were efficiently restored during both recoveries (Fig. 6f). H3K9ac peaks that remained enriched after the first recovery were equally enriched around TSSs that recovered and that did not recover (Extended Data Fig. 8i), indicating that H3K9 acetylation is not responsible for the memory effect. Instead, we found diffuse but strong pre-existing E–P contacts at nonrecovery genes compared to recovery genes (Fig. 6g and Extended Data Fig. 9a), and nonrecovery genes also exhibited higher propensity to form loops (Extended Data Fig. 9b). Next, similar TSS analyses among the downregulated genes revealed that genes that did not recover were targets of the developmental repressor Polycomb, as indicated by abundant H3K27me3 signal around promoters (Fig. 6h). The same was found to be true in NPCs (Extended Data Fig. 9e), indicating that sustained gene downregulation might be linked to Polycomb activity in multiple cell types. Interestingly, downregulation happened without an apparent change in H3K27me3. Rather, we found that Polycomb loops gained substantial strength at nonrecovery TSSs (Extended Data Fig. 9c,d), as well as genome wide (Fig. 6i and Extended Data Fig. 9f). Critically, increased Polycomb-mediated looping persisted after the first recovery period, without major change in H3K27me3 level at loop anchors (Extended Data Fig. 9g). Sustained loop strengthening was a feature specific to Polycomb rather than repressive loops in general, as we found de novo H3K9me3 loops to recover efficiently (Extended Data Fig. 9h).

In conclusion, repeated transient HDAC inhibition triggers cellular memory of gene expression that is associated with strong architectural features surrounding deregulated TSSs—in case of upregulated genes prominent preformed E–P contacts, at downregulated TSSs bolstered repressive Polycomb loops might perpetuate altered activity states.

## PRC1-mediated chromatin loops govern continued gene downregulation

As sustained transcriptional downregulation of Polycomb target genes happened without the accumulation of the H3K27me3 mark deposited by Polycomb repressive complex 2 (PRC2), we sought to understand whether memory relies on the activity of the PRC1 complex. We performed calibrated ChIP–seq analysis of Ring1B, which revealed increased PRC1 binding at >8,000 sites after recovery from TSA (Fig. 7a). Neither H3K27me3 nor H2AK119ub showed sustained increase at these sites, consistent with our finding that the Polycomb histone modification landscape remains largely unchanged. To test whether transcriptional memory relied on PRC1-mediated spatial clustering, we disrupted Polycomb-mediated looping through the depletion of two subunits (PCGF2 and PCGF4) of canonical PRC1 (Fig. 7b), using an ESC line (termed *Pcgf2*$^{fl/fl}$) in which *Pcgf4* is deleted and *Pcgf2* can be removed by tamoxifen (OHT). This line was previously used to demonstrate that PCGF2 and PCGF4 are responsible for creating interactions among Polycomb domains[39,40] (Extended Data Fig. 10a,b) and that this architectural role is independent from H3K27me3 and H2AK119ub (Extended Data Fig. 10c,d). Although sustained PCGF2 depletion led to mild derepression of Polycomb targets, the transcriptional response to the TSA–recovery double treatment course was highly similar among E14, *Pcgf2*$^{fl/fl}$ –OHT and *Pcgf2*$^{fl/fl}$ +OHT cells (Extended Data Fig. 10e–h). PCGF2 depletion, however, decreased the number of genes that did not recover from the TSA pulse by more than twofold (Fig. 7c), indicating a weaker memory response. Critically, most genes (321/366) that showed sustained downregulation in the −OHT condition did not exhibit sustained downregulation in the +OHT condition (Fig. 7d), indicating that the disruption of Polycomb loops rewrites the memory response to TSA. We nevertheless identified a smaller subset of genes that showed continued downregulation in +OHT cells that were also Polycomb targets (Fig. 7d,e). However, while the gene set that remained downregulated only in −OHT cells showed increased looping at TSSs, the nonrecovery genes unique to +OHT cells (152/197 genes) did not, and increase in Ring1B at +OHT TSS loop anchors occurred to a lesser degree (Extended Data Fig. 9b and Fig. 7f–h), indicating that their downregulation relies on a mechanism that is independent from PRC1-mediated chromatin looping.

In sum, disruption of Polycomb domain interactions modulates the memory response, demonstrating that PRC1-mediated spatial clustering is responsible for the TSA-induced sustained downregulation at Polycomb target genes.

## Discussion

The interplay between epigenetic layers and whether they function synergistically or antagonistically is an area of active research. The findings presented in this study describe a crosstalk between epigenetic modifications and genome folding, which together modulate the mESC transcriptional program. Namely, acute perturbation of histone acetylation rapidly translates to changes in histone methylation and 3D chromatin organization. Although the majority of epigenomic changes are reversible, we find that certain alterations in 3D genome folding persist and associate with a transcriptional memory effect at a subset of genomic loci.

In addition to the general opening and activation of chromatin at promoters, we observe widespread H3K4me1 deposition upon TSA treatment, suggesting the deployment of new enhancers. These are likely to be major drivers of gene upregulation, as previous studies have shown that the enhancer landscape—rather than promoter activity—is more substantial for lineage determination[41,42]. Accordingly, enhancers are the most epigenetically dynamic regions of the genome[43,44], explaining their susceptibility to the disruption of chromatin state balance. Interestingly, we find that H3K27 acetylation seems to precede H3K4me1 deposition, which questions the commonly accepted sequence of events in enhancer activation where H3K4me1 is supposed to precede H3K27ac[45,46]. Once triggered, the maintenance of enhancer activity is an active process[47–49], explaining the efficient recovery of the enhancer landscape and transcriptional program once the acetylation state is restored. We find that gene upregulation occurs without changes in E–P contacts, which agrees with recent studies that uncouple gene activation from a need to increase the frequency of physical contact[50–52], or find that they are coupled only during terminal tissue differentiation but not in cell-state transitions[53]. Instead, we find pre-existing E–P contacts that correlate with the memory effect. Indeed, it is thought that preformed E–P contacts may prime some genes for activation[20,27,54–57], but additional triggers are required for transcription to take place. We speculate that excess histone acetylation activates enhancers, and those that are structurally in a high-contact probability with their promoter targets can maintain active transcription after the removal of ectopic acetylation.

While chromatin looping is commonly discussed in the context of E–P contacts, our study highlights the importance and the potency of repressive chromatin loops. Counterintuitively, we find that ectopic chromatin activation enhances looping between loci marked by repressive chromatin signatures. One such class of loops corresponds to Polycomb contacts that are central to TSA-induced sustained downregulation of gene expression. In neural progenitors, Polycomb loci are known to exhibit transcriptional memory in *cis*, and this memory is linked to antagonism between PRC2 and activating signals[58]. Thus, one possible explanation is that ectopic genome-wide chromatin activation draws activating complexes away from Polycomb targets, shifting the equilibrium toward gene downregulation. Crucially, as continued gene downregulation involves minimal—if any—change in the H3K27me3–H3K27ac balance at promoters, enhanced spatial sequestration of Polycomb loci appears to be central to the mechanisms of repression, constituting an architecture-based memory. Indeed, disruption of Polycomb-mediated spatial clustering modulates

---

**Fig. 6 | Sustained gene expression deregulation is associated with strong regulatory 3D contacts. a**, After the recovery period, cells were exposed to a second TSA pulse, wash and recovery cycle. **b**, Scatterplot showing the correlation of transcriptomic changes induced by the first and second TSA treatments. Fitted line shows linear regression. **c**, Volcano plot showing differential gene expression (significance cutoffs (dashed lines)—adjusted *P* value (derived from Wald test, corrected for multiple testing with BH approach) <0.05, absolute log$_2$(FC) >1) after recovery from a second TSA (reREC) treatment. Labels correspond to core and naive pluripotency, formative, meso-endodermal, CNS differentiation and neurogenesis marker genes. **d**, Development-related GO term enrichment among genes that remain misregulated after the first and second recoveries from TSA treatment. *P* values correspond to one-sided Fisher's exact test corrected for FDR with the BH method. **e**, Gene expression *z* scores of recovered and not recovered genes (*n* = number or genes in group)

through the TSA–recovery time course. Data shown are the median, with hinges corresponding to IQR and whiskers extending to the lowest and highest values within 1.5× IQR. **f**, H3K27ac ChIP–seq and ATAC–seq signal in DMSO, TSA, REC, reTSA and reREC at upregulated TSSs that recover (top) and do not recover (bottom). **g**, Pile-up of E–P contacts in DMSO, TSA and REC around upregulated TSS that recover (left) and do not recover (right). **h**, H3K27ac and H3K27me3 ChIP–seq signal in DMSO, TSA, REC, reTSA and reREC at downregulated TSSs that recover (top) and do not recover (bottom). **i**, Aggregate plots of Micro-C signal in DMSO, TSA and REC at all (left) and non-CTCF (right) loops with H3K27me3 ChIP–seq signal at loop anchors (resolution = 4 kb). Quantification of piled-up loop strength (*n* = 9 corresponding to the central 3 × 3 pixels) is shown on the right (paired two-tailed *t* test; NS > 0.05, **$P$ < 0.01, ****$P$ < 0.0001). Box plot elements are as in **e**. reREC, re-Recovery.

the transcriptional memory response triggered by TSA. This is consistent with prior findings showing that, besides local chromatin compaction[59,60], long-range contacts are a mechanism by which Polycomb complexes confer silencing[55,61–66]. We also detect prominent looping between H3K9me3-marked loci as a potential mechanism of gene downregulation. It might be interesting to investigate if H3K9me3 contacts are mediated by chromatin-binding proteins such as HP1 (ref. 67) and its associated partners.

Acute disruption of the acetylation landscape also led to important changes in global genome folding. The increase in *trans* contact points to the possibility that histone acetylation might be an important determinant of intrachromosomal interactions and chromosome territories. Indeed, it has been shown that long, highly transcribed genes or gene-dense regions extend from chromosome territories[21,68–70], although this has been attributed to binding of ribonucleoproteins to nascent transcripts rather than to the acetylation state per se. Using

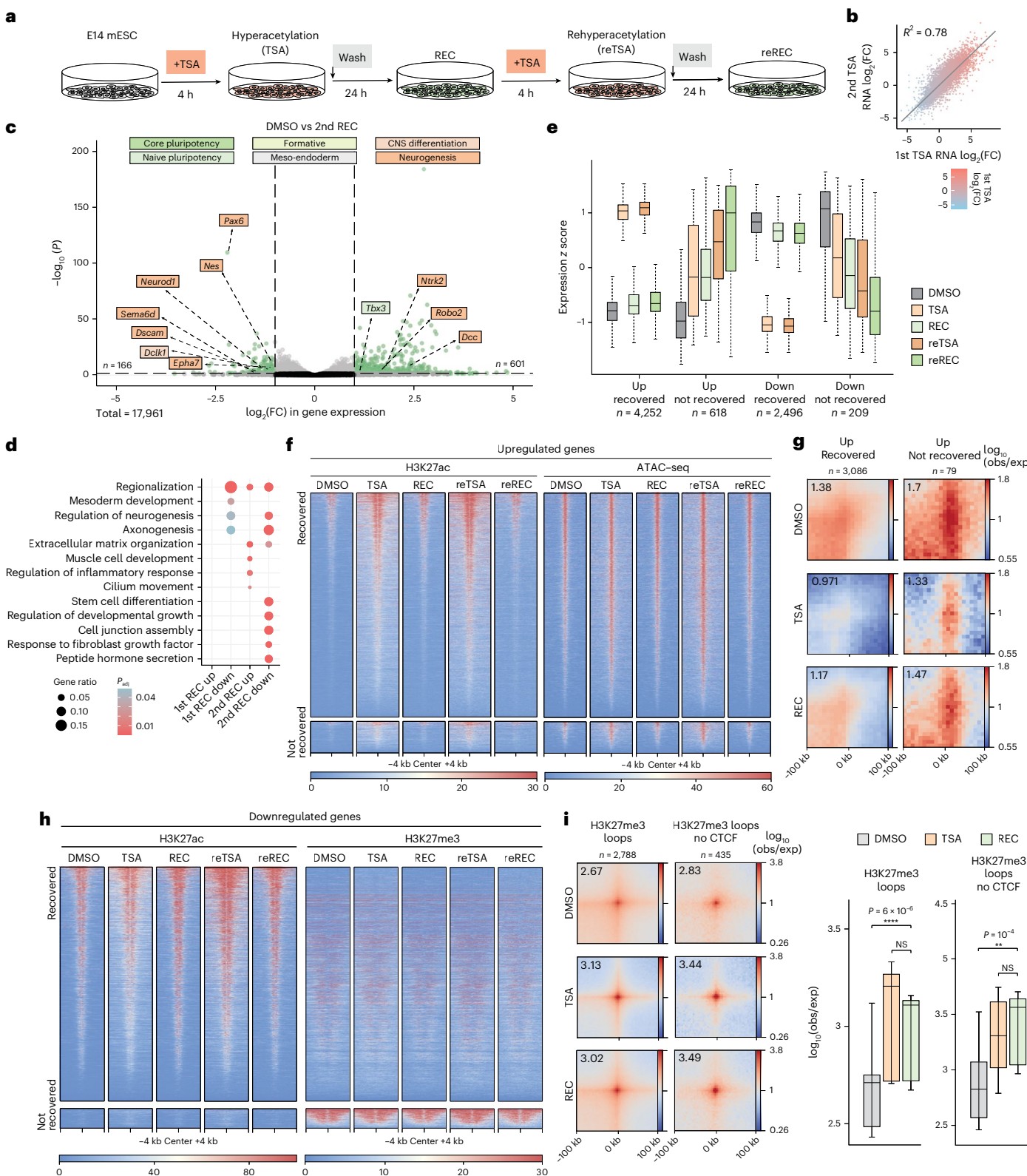

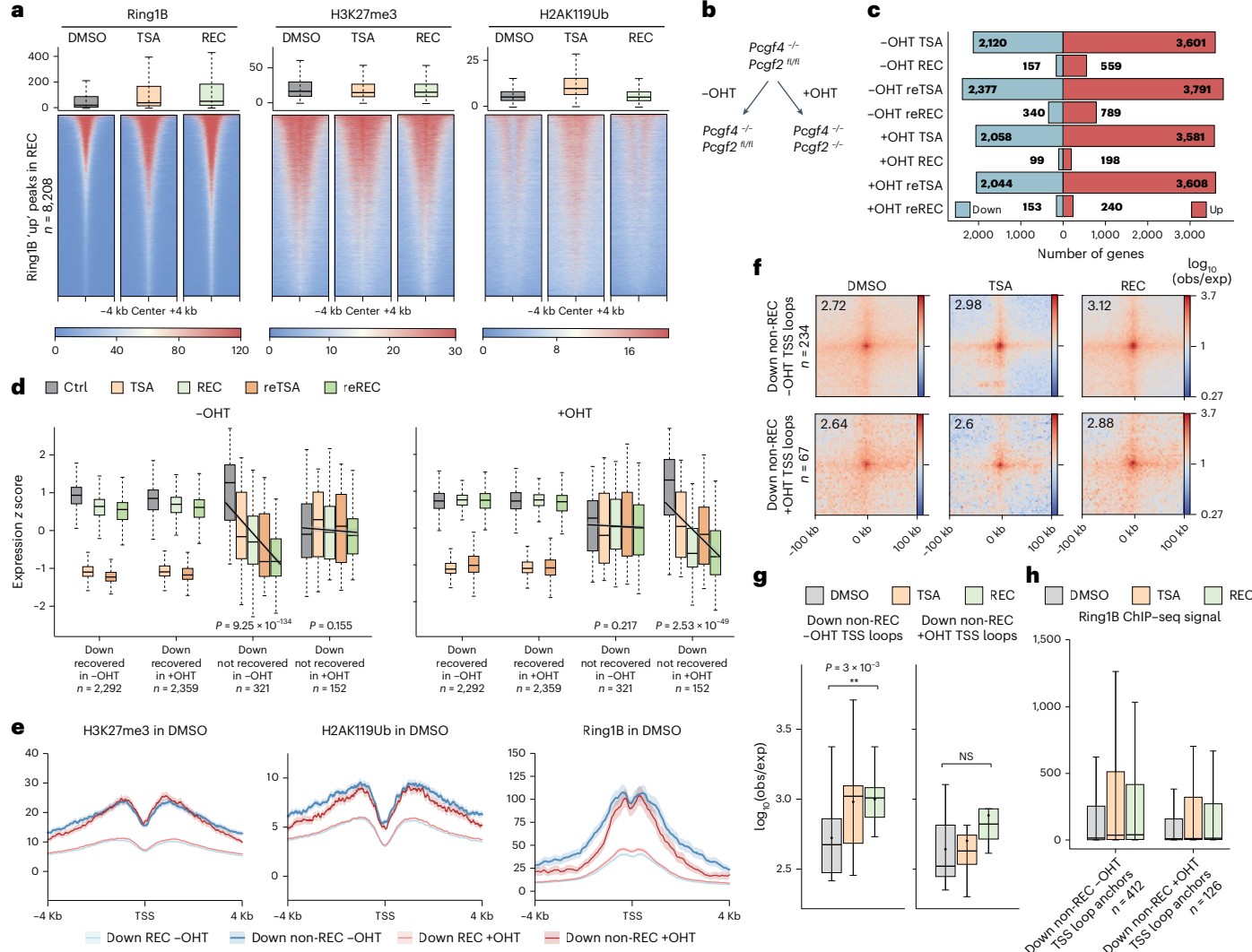

**Fig. 7 | Disruption of PRC1-mediated 3D chromatin contacts changes memory response triggered by TSA treatment. a**, Box plots (top) and heatmaps (bottom) of normalized Ring1B, H3K27me3 and H2AK119ub ChIP-seq signal in DMSO, TSA and REC at genomic sites with increased Ring1B binding in REC condition ($n$ = 8,208 ChIP-seq peaks). Data on box plots shown are the median, with hinges corresponding to IQR and whiskers extending to the lowest and highest values within 1.5× IQR. **b**, Schematics of tamoxifen (OHT) treatment in a *Pcgf4*[−/−]; *Pcgf2*[fl/fl] mESC cell line. **c**, Diverging bar plot showing the number of differentially expressed genes during the TSA–recovery treatment course in the *Pcgf4*[−/−]; *Pcgf2*[fl/fl] cell line with (+OHT) or without (−OHT) PCGF2 depletion. **d**, Gene expression $z$ scores of recovered and not recovered genes ($n$ = number or genes in group) through the TSA–recovery treatment course in −OHT (left) and +OHT (right) conditions. Control conditions for −OHT and +OHT are DMSO and DMSO + 120-h OHT treatment, respectively. Box plot elements are as in **a**, and

fitted lines represent linear regression with shading corresponding to standard error. $P$ values were derived using unadjusted two-sided $t$ test on regression slopes. **e**, Metaplots showing mean H3K27me3, H2AK119ub and Ring1B ChIP-seq signal in wild-type E14 mESCs at TSSs of downregulated genes that recover and do not recover in −OHT and +OHT conditions. Shading represents ±s.e.m. **f**, Aggregate plots of Micro-C signal in DMSO, TSA and REC around loops where anchors overlap with downregulated TSSs that do not recover in −OHT (top) or +OHT (bottom) conditions (resolution = 4 kb). **g**, Quantification of piled-up loop strength ($n$ = 9 corresponding to 3 × 3 central pixels) shown in **f** (paired two-tailed $t$ test; NS > 0.05, **$P$ < 0.01). Box plot elements are as in **a**. **h**, Box plot showing normalized Ring1B ChIP-seq signal at TSS loop anchors ($n$) in DMSO, TSA and REC around downregulated genes that do not recover in −OHT and +OHT conditions. Box plot elements are as in **a**.

biophysical modeling, we found that, by increasing the stiffness of the chromatin fiber[21], we can recapitulate the *trans*-contact ratio observed in TSA. It is widely accepted that homotypic interactions among domains of the same epigenetic state are the major driving force of chromosome compartmentalization[71]. Interestingly, global chromatin activation weakened rather than strengthened A–A compartment interactions to an extent that is comparable to what occurs during ESC differentiation[20]. Simulations that we carried out to understand whether increased chromatin stiffness can give rise to excess *trans* contacts efficiently predicted the change in compartment interactions that we observed in the Micro-C data. This suggests that chromatin stiffness

is an important biophysical determinant of not only interchromosomal contacts but also A/B compartmentalization.

Of note, we found that perturbed compartment interactions can partially persist beyond the recovery period, signifying that 3D structures carry a memory of their past state. This might be explained by hysteresis, the dependence of a system's behavior on its history. Hysteresis is an emerging principle in 3D genome organization[72] that has been found to be critical for modeling certain characteristics of genome folding[73,74] and has been demonstrated experimentally[75]. Additionally, biophysical modeling has shown that 3D genome folding might be a crucial element to stabilize epigenetic memory[76–79]. Our study further

supports these observations and provides empirical evidence of an architectural memory both at the global scale and at the gene level.

Finally, the ability of cells to record previous stimuli and trigger heightened responses to subsequent stimulations has potential implications for human health, as commonly used epidrugs are administered repeatedly during treatment. Thus, exposed cells will likely undergo multiple cycles of acute responses followed by recovery, potentially inducing long-term effects that warrant further study.

## Online content

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

## Methods

### ESC culture

E14GT2a p14 cells were purchased from MMRRC, UC Davis. *Pcgf4⁻/⁻*; *Pcgf2^{fl/fl}* cells were gift from R. Klose (University of Oxford), *Pcgf4* deletion and *Pcgf2* excision in response to tamoxifen (OHT) were verified by genotyping PCR. CTCF–AID–eGFP cells were a gift from R. Saldana-Meyer (Howard Hughes Medical Institute); CTCF–AID–eGFP expression was confirmed by anti-GFP immunofluorescence. ESCs were cultured on plastic plates coated with 0.1% gelatin (Sigma-Aldrich, G1890-100G) in serum-LIF medium (GMEM (Gibco, 2171002), with 15% FBS (Thermo Fisher Scientific, 26140079), 1× GlutaMAX (Thermo Fisher Scientific, 35050038), 1× MEM nonessential amino acids (Thermo Fisher Scientific, 11140035), 50 U penicillin–streptomycin (Gibco, 15140122), 0.1 mM sodium pyruvate (Gibco, 11360070), 0.1 mM 2-mercaptoethanol (Gibco, 31350010) and 1,000 U ml⁻¹ LIF (Sigma-Aldrich, ESG1107)). Cells were passaged every 2–3 days using TrypLE Express Enzyme (Gibco, 12604013). Cell lines were regularly tested for mycoplasma infection. Cell viability was assessed by staining with trypan blue (Gibco, 15250061), and cells were counted on a Countess 3 automated cell counter (Invitrogen). HDAC inhibition was performed by treating cells with 100 ng ml⁻¹ TSA (Sigma-Aldrich, 647925) for 4 h or with 0.03 nM romidepsin for 6 h. Control cells were treated with 0.01% DMSO for the same duration. PCGF2 depletion was induced by growing *Pcgf4⁻/⁻*; *Pcgf2^{fl/fl}* cells in medium supplemented with 800 nM 4-hydroxytamoxifen (OHT) for 72 h before each experiment. For recovery, cells were washed once with PBS and were incubated with fresh mESC medium for 10 min. This PBS wash/medium change was repeated twice before incubating cell for a total of 24 h.

### Differentiation of NPCs

NPCs were grown using previously published retinoic acid-based protocol[80]. Briefly, 4 × 10⁶ ESCs per replicate were cultured in suspension in Petri dishes in high glucose DMEM (Thermo Fisher Scientific, 21710025) supplemented with 1× GlutaMAX (Thermo Fisher Scientific, 35050038), 1× nonessential amino acids (Thermo Fisher Scientific, 11140035), 0.1 mM 2-mercaptoethanol (Thermo Fisher Scientific, 31350010), 1× penicillin–streptomycin (Thermo Fisher Scientific, 10378016) and 10% FBS (Thermo Fisher Scientific, 26140079). After 4 days, the medium was supplemented with 5-μM retinoic acid for an additional 4 days. The medium was changed every 2 days. Following, NPCs were replated onto gelatin-coated cell culture plates and allowed to reattach for 2 days, after which TSA treatment and washes were performed as with ESCs.

### Gastruloid culture

Gastruloids for RNA-seq and immunostaining experiments were generated as described in ref. 81. Briefly, CTCF–GFP–AID cells were collected, centrifuged and washed twice with PBS. Cells were then resuspended in N2B27 medium and counted. A total of 300 cells were seeded in each well of a round-bottomed, low-attachment 96-well plate (Greiner, 650970) in N2B27 medium. After 48 h, a 24-h pulse of 3-μM CHIR99021 (Tocris Bioscience, 4423; Chiron) was administered and medium was changed every day. HDAC inhibition was carried out by treating gastruloids with 20 ng ml⁻¹ TSA (Sigma-Aldrich, 647925) for 4 h immediately before the Chiron pulse (44–48 h). TSA was removed from the medium by changing N2B27 medium thrice with 10 min of incubation in between. Control cells were washed similarly.

### Western blotting

For western blotting ~10⁷ mESCs were dissociated, washed once in PBS, resuspended in 200 μl of cell lysis buffer (85 mM KCl; 0.5% NP40; 5 mM HEPES pH 8; 1× ethylenediaminetetraacetic acid-free protease inhibitor (Roche); 5 mM sodium butyrate) and incubated on ice for 15 min. Afterwards, nuclei were pelleted at 2000g for 5 min at 4 °C. The supernatant (cytoplasmic fraction) was separated, and nuclei were resuspended in 100-μl RIPA buffer (50 mM Tris, pH 7.5, 150 mM NaCl, 1% NP40, 0.5%

NaDoc, 0.1% sodium dodecyl sulfate, 1× ethylenediaminetetraacetic acid-free protease inhibitor (Roche, 04693132001), 5 mM sodium butyrate). After a 10-min incubation on ice, chromatin was digested for 15 min at 37 °C with 0.0125 U μl⁻¹ MNase and 1 mM CaCl₂. Extracts were cleared by 30 min of centrifugation of >10000g at 4 °C. Protein yield was quantified using the Pierce BCA protein assay kit (Thermo Fisher Scientific, A65453). Samples were mixed with 4× NuPAGE LDS sample buffer (Thermo Fisher Scientific, NP0007) and boiled for 10 min at 95 °C. Furthermore, 2-μg denatured protein extract was loaded per lane on a NuPAGE 4–12%, Bis–Tris gel (Thermo Fisher Scientific, NP0321BOX). Transfer onto nitrocellulose membranes was performed using the Trans-Blot Turbo Transfer System (Bio-Rad). Membranes were stained with Ponceau S for 5 min, then blocked for at least 30 min with 3% BSA in PBS + 0.1% Triton X-100 before incubation with primary antibody overnight at 4 °C with the following dilutions: α-H3K27ac (1:7,500; Active Motif, 39133), α-pan-acetyl lysine (1:1,000; Thermo Fisher Scientific, 66289-1-IG), α-acetyl-tubulin (1:2,000; Sigma-Aldrich, T7451), H3K4me1 (1:5,000; Active Motif, 39297), H3K4me3 (1:1,000; Milipore, 04-745), H3K9me3 (1:2,000; Abcam, ab8898), H3K27me3 (1:2,500; Active Motif, 39155), H2AK119ub (1:2,000; Cell Signaling Technology, 8240S), H3K9ac (1:7,500; Millipore, 07-352) and α-lamin B1 (1:10,000; Abcam, ab16048). Membranes were washed thrice >5 min in PBS + 0.1% Tween-20 and were incubated with secondary antibodies (α-rabbit IgG–peroxidase antibody (Sigma-Aldrich, A0545) or α-mouse IgG–peroxidase antibody (Sigma-Aldrich, A9044)) at 1:16,000 dilution for 1 h at room temperature. After three >5-min washes with PBS 0.1% Tween-20 at room temperature, membranes were developed using the SuperSignal West Dura Extended Duration Substrate solution (Thermo Fisher Scientific, 34075) for 1 min and imaged with a Bio-Rad ChemiDoc imager.

### Flow cytometry

A quantity of 1–3 × 10⁶ mESCs were dissociated with TrypLE, and pelleted and resuspended in PBS. For cell cycle analysis, dissociated mESCs were washed once in PBS and pelleted and fixed in cold 70% ethanol for 30 min at 4 °C. Cells were stained with the propidium iodide flow cytometry kit (Abcam, ab139418) according to the manufacturer's instructions. Flow cytometry was performed on a CytoFlex instrument using CytExpert (v2.4), and analysis was performed using FlowJo (v10.10). For cell proliferation tracing, dissociated mESCs were stained with 1 μM CellTrace Violet staining solution (Invitrogen, C34571) according to the manufacturer's instructions, and were plated on gelatin-coated cell culture dishes. After 24 h, TSA treatment and washes were performed as described before and cells were collected after a further 24-h incubation period. Collected cells were fixed in 4% PFA for 10 min at room temperature, washed with PBS and preserved at 4 °C until further use. Flow cytometry was performed on a NovoCyte Quanteon, and data analysis was performed using NovoExpress (v1.6.3).

### mESC immunostaining

ESCs were seeded onto glass coverslips precoated with 0.1% gelatin (Sigma-Aldrich, G1890-100G). Two hours after seeding, cells were treated with 100 ng ml⁻¹ TSA (Sigma-Aldrich, 647925) or with 0.01% DMSO for 4 h, then rinsed with PBS and fixed with 4% paraformaldehyde (Thermo Fisher Scientific, 28906) for 10 min. After three more PBS washes, cells were permeabilized for 15 min with fresh PBS + 0.3% Triton X-100. After four washes with PBT (PBS + 0.02% Tween-20), the blocking was performed using PBT + 2% BSA for at least 30 min. Then, cells were incubated with primary antibody α-H3K27ac (1:200; Active Motif, 39133) in PBT + 2% BSA for 72 h at 4 °C with orbital shaking to prevent antibody trapping[82]. Afterward, cells were washed four times with PBT and stained with secondary antibody α-rabbit 555 (Invitrogen, A31572) for 1 h at room temperature. After a new round of four washes with PBS, cells were counterstained with 0.2 μg ml⁻¹ DAPI for 10 min at room temperature on orbital shaker, before being rinsed twice

with PBS. Coverslips were mounted in ~15-μl Vectashield (Eurobio Scientific, H1000) and stored at 4 °C before imaging.

## Image acquisition and quantification

DMSO and TSA conditions in each experiment were imaged and analyzed using the same parameters. Confocal imaging was performed using a Zeiss confocal LSM980 Airyscan 2 equipped with ×63 (ESCs) or ×20 (gastruloids) objectives using ZEN Blue (v3.8-3.12). Diodes laser 405, 488, 561 and 639 nm were used for fluorophore excitations. For each gastruloid, three $z$ stacks were taken and, using Fiji (v2.14.0), maximum intensities were projected to manually define areas of H3K27ac, Sox2 and Brachyury expression and DAPI staining. For ESCs, several $z$ stacks were taken for each condition. Quantification of nucleus volumes and H3K27ac distances to periphery was performed using Imaris (v10.1.1). The option 'surfaces' was used to segment nuclei, H3K27ac signal was analyzed as 'spots' ($xy$ diameter = 0.3 μm/$z$ diameter = 0.6 μm). 'Distance transformation' was used to generate distance to periphery, and the option 'split spots into surface objects' was used to assign spots to the corresponding nuclei. The distance between each spot and the periphery is given by the intensity of the 'distance transformation' channel at the center of each spot.

## Gastruloid immunostaining

Gastruloid immunostaining protocol was adopted from ref. 83. Plastic material was precoated with blocking solution (PBS + 10% FBS + 0.2% Triton X-100). Using a cut P1000 tip, gastruloids were collected into 15-ml centrifuge tubes. After a PBS wash, gastruloids were transferred to 2 ml of 4% PFA in six-well plates and fixed overnight at 4 °C. For washes, gastruloids were transferred serially across three PBS-filled wells and were incubated for 10 min in the last one. Gastruloids were blocked in PBS + FT (PBS + 10% FBS + 0.2% Triton X-100) for 1 h at room temperature, then incubated with primary antibodies (α-brachyury (1:500; Santa Cruz Biotechnologies, sc-166962), α-Sox2 (1:500; eBioscience, 15208187), α-H3K27ac (1:200; Active Motif, 39133)) in PBS + FT and 1 μg ml$^{-1}$ DAPI overnight at 4 °C with orbital shaking. Gastruloids were washed by sequentially transferring them to three wells filled with PBS + FT and incubating them for 20 min in the last one. Staining with secondary antibody (α-rabbit Alexa Fluor Plus 488 (1:400; Thermo Fisher Scientific, A32731), α-rabbit Alexa Fluor Plus 555 (1:400; Thermo Fisher Scientific, A32794), α-rabbit Alexa Fluor Plus 647 (1:400; Thermo Fisher Scientific, A48265)), and 1 μg ml$^{-1}$ DAPI, as well as washes were carried out similarly to primary antibody. Gastruloids were mounted in ~30-μl Fluoromount-G (Thermo Fisher Scientific, 00-4958-02) and were kept at 4 °C before imaging.

## RNA isolation for RNA-seq

RNA was isolated using the RNeasy mini kit (Qiagen, 74104). Cells were detached with TrypLE, lysed in RLT buffer with β-mercaptoethanol and lysates were processed according to the manufacturer's instructions. For mESCs columns and buffers supplied with the RNeasy kit were used, while for gastruloids, the Zymo RNA Clean & Concentrator-5 (Zymo Research, R1015) reagents were used. On-column DNase-I digestion (Qiagen, 79254) was performed as recommended. RNA samples were sent to BGI Tech Solutions for strand-specific transcriptome sequencing. Samples were sequenced at a depth of 50 million 150-bp paired-end reads.

## Micro-C library preparation and sequencing

Micro-C libraries were generated with the Dovetail Micro-C Kit protocol (v1.0) with minor modifications. Briefly, $10^6$ mESCs were washed with PBS and were frozen at −80 °C for at least 1 h. Cell pellets were thawed and crosslinked first with 3 mM DSG (Thermo Fisher Scientific, A35392) in PBS for 10 min at room temperature with rotation, then formaldehyde was added at 1% final concentration for a further 10 min. The pellets were washed twice with PBS and digested with

MNase according to the kit instructions. MNase digestion was routinely verified by decrosslinking a small amount of chromatin and assessing fragment distribution on a Bioanalyzer 2100 instrument (Agilent). If the digestion profile showed 50–70% mononucleosomal DNA fraction, on-bead proximity ligation was performed, followed by crosslink reversal and DNA purification. End repair and adaptor ligation were performed using the NEBNext Ultra II DNA Library Prep Kit for Illumina (NEB, E7645). Following, DNA was purified using Solid Phase Reversible Immobilization beads (Beckman, B23318) as described in the Micro-C user manual. Finally, biotin pulldown and library amplification were performed according to the Dovetail Micro-C Kit User Guide and using Dovetail Micro-C Kit reagents, only replacing the Dovetail Primers (Universal and Index) with NEBNext primers. Libraries were pooled and sent to BGI Tech Solutions for 100-bp paired-end sequencing to obtain roughly 2–3 billion reads per replicate.

## ChIP

ChIP was performed as described previously[84]. Cells were collected with TrypLE (Thermo Fisher Scientific, 12604013) and fixed in mESC medium containing 1% methanol-free formaldehyde for 10 min with rotation at room temperature. Glycine (2.5 M glycine in PBS) was used to stop the fixation for 10 min with rotation at room temperature. Fixed cells were centrifuged at 500$g$ for 5 min at 4 °C, washed twice in 1× ice-cold PBS and snap frozen in liquid nitrogen until further use. After thawing, cells were spiked-in with 8% HEK-293 cells and chromatin extraction was performed as discussed in ref. 85 with sonication on a Covaris E220 instrument (Duty Factor 5%; PIP 140 W; cycles per Burst 200; 12 min). A total of 15 μg of chromatin was used for each replicate of histone ChIP, and 50 μg for CTCF, YY1 and Ring1B ChIP, with 6–8 μg of antibody. Because the above protocol was not suitable for YY1, we followed the protocol described in ref. 86. Briefly, fixed cells were resuspended in sodium dodecyl sulfate buffer, followed by sonication and preparation for immunoprecipitation. Next, the mixture was incubated overnight at 4 °C with Protein G beads (Invitrogen, 10004D), washed with both low and high-salt buffers, reverse-crosslinked in elution buffer, and purified using a QIAQuick PCR purification kit (Qiagen, 28104). Antibodies used in this study were as follows: H3K4me1 (Active Motif, 39297), H3K4me3 (Milipore, 04-745), H3K27ac (Active Motif, 39133), H3K9me3 (Abcam, ab8898), H3K27me3 (Active Motif, 39155), H2AK119ub (Cell Signaling Technology, 8240S), H3K9ac (Millipore, 07-352), Ring1B (Cell Signaling Technology, 5694), CTCF (Active Motif, 61311) and YY1 (Abcam, 109237). For ChIP–qPCR, the LightCycler 480 SYBR Green Master (Roche, 04887352001) was used on undiluted ChIP DNA and input DNA in 1:10 dilution. Primer sequences are provided in the Supplementary Methods. For ChIP–seq, sequencing libraries were constructed using NEBNext Ultra II DNA Library Prep Kit for Illumina (NEB, E7645), pooled and sent to BGI Tech Solutions for 100-bp paired-end sequencing to obtain roughly 30–50 million reads per replicate.

## Assay for transposase-accessible chromatin using sequencing (ATAC–seq)

For each replicate, $9 × 10^4$ mESCs were collected with TrypLE (Thermo Fisher Scientific, 12604013) and were mixed with $10^4$ HEK-293 cells. Samples were processed using the Active Motif ATAC–seq kit (Active Motif, 53150) following the manufacturer's instructions without modifications. ATAC–seq libraries were pooled and sent to BGI Tech Solutions for 100-bp paired-end sequencing, yielding approximately 30–50 million reads per replicate.

## Statistics and reproducibility

RNA-seq experiments were performed in biological triplicates. ChIP–qPCR, ChIP–seq and ATAC–seq experiments were performed in biological duplicates, except for H3K27ac ChIP–seq in DMSO, TSA and recovery where three independent replicates were performed.

Micro-C was performed in biological duplicates, except for DMSO and TSA, where five biological replicates were produced. Gastruloid immunostaining was performed in biological triplicates, mESC immunostaining in duplicates. Sample sizes are indicated in figures and/or legends. Cell viability and cell cycle profiling were performed in biological triplicates, and cell generation tracing was performed in duplicates. H3K27ac western blotting was routinely performed to verify the effects of TSA and washes. All other western blots were performed in biological duplicates, except for H3K27ac in NPCs, where a single replicate was performed.

Data collection and analyses were not performed blindly to the conditions of the experiments. No data were excluded from the analyses, except for gastruloid immunofluorescence, where gastruloids with clear morphological and/or symmetry aberrations were not imaged. The experiments were not randomized. For statistical analyses, normality was assessed using the Shapiro–Wilk test. For small sample sizes (<10), the data were assumed to be normally distributed, although this was not formally tested. No statistical method was used to predetermine sample size.

### RNA-seq analysis

RNA-seq samples were mapped using the 'align' function of the Subread package (v2.0.6). Subread command 'featureCounts' (with options '-p --countReadPairs -s 2 -t exon'), and the feature file UCSC RefSeq GTF file for mm10 were used to generate count tables that were then used as inputs for DESeq2 (v1.42.1)[87] to perform differential analysis (Supplementary Tables 1–4). GO analysis was performed using the 'enrichGO' function from the clusterProfiler (v4.10.1) package[88] (Supplementary Table 5). Volcano plots and scatterplots were produced in R using the EnhancedVolcano (v1.20.0) and ggplot2 (v3.5.1) libraries, respectively. Motif enrichment analysis at differentially expressed genes was carried out using the 'findMotifs.pl' function of the HOMER (v4.10.0) Motif Discovery and Analysis tool[89] (Supplementary Table 6).

### ChIP–seq and ATAC–seq analysis

ChIP–seq and ATAC–seq samples were mapped using bowtie2 (v2.4.4)[90] with command 'bowtie2 -p 12 --no-mixed --no-discordant' against the mm10 and hg19 genomes. Then, samtools (v1.9)[91] was used to filter out low-quality reads (command 'samtools view -b -q 30') and sambamba (v1.0)[92] was used to sort (command 'sambamba sort'), deduplicate and index BAM files ('sambamba markdup --remove-duplicates') with default parameters. Following, samtools was used to count both human and mouse reads (command 'samtools view -c') to calculate the downsampling factor (dF) for spike-in normalization as described in ref. 39. Next, BAM files were downscaled accordingly using samtools (command 'samtools view -b -s dF') and bigwig files were produced using the deepTools package[93] with command 'bamCoverage --normalizeUsing none --ignoreDuplicates -e 0 -bs 10'. Finally, ChIP–seq tracks were visualized using IGV (v.2.16.1)[94] or HiGlass (v1.11.8)[95]. ATAC–seq and ChIP–seq peaks were called on each replicate using MACS3 (v3.0.3) with a *q*-value cutoff of 0.05, and for histone marks with the additional parameters '--broad --broad-cutoff 0.1' (ref. 96). Finally, peaks detected from both replicates were filtered and all downstream analyses were carried out using this consensus peak set. For differential peak calling the diffBind (v3.12.0)[97], R package was used with normalization 'normalize = DBA_NORM_LIB, spikein = TRUE', analysis method 'method = DBA_DESEQ2', and false discovery rate of <0.05 cutoff (Supplementary Table 7). Heatmaps and metaplots were produced using the 'computeMatrix' function of the deepTools (v3.5.6) package, and plotted using the 'plotHeatmap' and 'plotProfile' functions. ChIP–seq box plots were also created by deepTools using the 'multiBigwigSummary' function and were plotted by ggplot2 (v3.5.1) in R. Chromosome-wide H3K27ac read density plots were generated using a custom R script published in ref. 39. ChIP–seq peak distribution and annotation were carried out with ChIPseeker's[98] (v1.38.0) 'plotPeakProf' and 'annotatePeak' functions,

respectively. GO analysis of annotated ChIP–seq peaks was performed using the 'enrichGO' function from the clusterProfiler (v4.10.1) package[88] (Supplementary Table 5). Differential ATAC–seq peaks were analyzed with the i-cisTarget online tool[99,100], using v.6.0 of the position weight matrix database filtered for hits in the HOMER database (Supplementary Table 6). For cumulative histograms, enhancer distance from TSSs was calculated using bedtools (v2.31.1) 'closest' function and was plotted by ggplot2 (v3.5.1) in R. Expression-matched control gene set was derived using code from the AdelmanLab github repository (https://github.com/AdelmanLab/Expression-Matching). Myc ChIP–seq in ESC and H3K27me3 in NPCs were published previously[101,102].

### Micro-C data analysis

**Generation of contact matrices and standard analyses.** Micro-C data were mapped using the HiC–Pro (v3.1.0) pipeline[103]. FASTQ reads were trimmed to 50 bp using TrimGalore (v0.6.10; '--hardtrim5 50'; https://github.com/FelixKrueger/TrimGalore) and aligned to the mm10 reference genome using bowtie2 (ref. 90; v2.4.4; '--very-sensitive --L 30 --score-min L, -0.6, -0.2 --end-to-end --reorder'), removing singleton, multihit and duplicated reads. Minimum *cis*-distance was set at 200 bp. The total numbers of valid read pairs per sample are reported in Supplementary Table 8. Contact matrices in the .cool file format were generated using cooler[104] (v.0.10.2) at 100-bp resolution (command 'cooler cload pairs -c1 2 -p1 3 -c2 5 -p2 6 ./scripts/chrom_sizes.txt:100'). Similarities between replicates (five replicates for DMSO and TSA; two replicates for 24-h recovery) were measured applying 'HiCRep' (v1.12; https://github.com/TaoYang-dev/hicrep)[105] on chromosomes 2, 9, 13 and 19 using the 'get.scc' function with parameters resol = 20 kb and (lbr,ubr,h) = ((0, 100 kb,1), (100 kb, 500 kb,1), (500 kb, 2 Mb,2), (2 Mb, 10 Mb,4)). *H* values were previously trained using the 'htrain()' on two replicates of the DMSO condition. Using 1.0-SCC, as a measure of the similarity (0 = similar and 1 = dissimilar) between replicates and hierarchical clustering analysis using 'hclust()' function in R with Ward.D2 method on the chromosome-averaged similarities, allowed to distinguish and group together the replicates of the different conditions, motivating to merge the valid-pairs of different replicates in a unique dataset for each condition. Multiresolution '.mcool' files were obtained and normalized through the Iterative Correction and Eigenvector decomposition algorithm (ICE) with default parameters (command 'cooler zoomify -r resolutions file.cool -o file.mcool --balance')[106] and were uploaded onto a local HiGlass (v1.11.8) server for visualization[95]. For comparison of architectural features among different conditions, contact maps were matched to contain approximately the same number of *cis*-contacts (Supplementary Table 8). All genomic snapshots of Micro-C maps were generated using HiGlass (v.1.11.8). Standard analyses (*cis*-decay curves, eigenvector decomposition, saddle plots) were performed using the cooltools (v0.5.4) package.

**Loop analyses.** Loops were called using mustache (v1.0)[107] with default parameters ('--pThreshold 0.1 --sparsityThreshold 0.88 --octaves 2') on ICE-balanced maps at 1-kb and 4-kb resolutions. Redundant loops among different resolutions were filtered in 20-kb windows, and coordinates were retained at the finer resolution. All aggregate plots were created with the coolpuppy (v1.1.0) package[108] and were normalized using expected maps generated by cooltools (v0.5.4). For differential looping, contacts that overlapped with the corresponding loop anchor bin were summed for each loop and were summarized into a count table—genome-wide count tables were created for each replicate at each resolution (command 'cooler dump --join -t pixels'), then filtered against loop using bedtools (v2.31.1) 'pairtopair' function[109]. The count tables from different conditions were used for differential analysis with DESeq2 (v1.42.1; Supplementary Tables 9 and 10). The thresholds $P_{adj} < 0.05$, $|\log_2$ fold change (FC)$| > 0.5$ and baseMean $\geq 10$ were used to filter for substantial changes in looping between conditions. Volcano plots were produced in R using the EnhancedVolcano (v1.20.0) library.

Loop subclasses were defined based on the presence of ChIP–seq peaks at loop anchors (repressive—overlapping with H3K9me3, H3K27me3 or H2AK119ub peaks; active—overlapping with H3K4me1, H3K4me3 or H3K27ac peaks; de novo H3K9me3 loops—loops only present in TSA overlapping with H3K9me3 peak; CTCF—loop anchors within ±1 kb of CTCF peaks; non-CTCF—no CTCF peak within ±2.5 kb of loop anchor) or the presence of TSSs within 2 kb of either loop anchor (Supplementary Table 9). E–P contacts for recovery versus nonrecovery genes were taken from ref. 20. Loop quantification box plots represent the observed/expected value of the central 3 × 3 pixels of aggregate plots that was extracted from coolpuppy matrices using an in-house Python script. Loop anchors were annotated using the 'annotatePeak' function of the ChIPseeker (v1.38.0) R package, and annotated anchors within <10 kb from TSSs were used for GO enrichment with the 'enrichGO' function of clusterProfiler (v4.10.1) library (Supplementary Table 5).

## Biophysical modeling
Biophysical modeling was performed as described in the Supplementary Methods.

## Reporting summary
Further information on research design is available in the Nature Portfolio Reporting Summary linked to this article.

## Data availability
All raw data were submitted to the National Library of Medicine's (NCBI) Sequence Read Archive and processed files were submitted to Gene Expression Omnibus (GEO). All data can be retrieved under the GEO series GSE281151. Myc ChIP–seq dataset was published in ref. 101 and was downloaded from the GEO repository GSE90895. NPC H3K27me3 ChIP–seq was published in ref. 102 and was downloaded from the GEO repository GSE262551. Source data are provided with this paper.

## Code availability
Custom scripts used in this article can be accessed at the Cavalli laboratory GitHub page at https://github.com/cavallifly/Paldi_et_al_NatGenet_2025 and under the Zenodo repository at https://doi.org/10.5281/zenodo.17608120 (ref. 110).

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

## Acknowledgements

The authors thank I. Jerkovic for comments on the manuscript. The authors are grateful to Dovetail Genomics (Cantata Bio) for their contribution to Micro-C reagent costs. The authors are grateful to the Genotoul bioinformatics platform Toulouse Occitanie (BioinfoGenotoul, https://doi.org/10.15454/1.5572369328961167E12), the national infrastructure France-BioImaging supported by the French National Research Agency (ANR-24-INBS-0005 FBI (BIOGEN)) and the Pôle Scientifique de Modélisation Numérique of the ENS de Lyon for computational resources. The authors also thank the Montpellier Resources Imagerie and P. Romé for their help and support with microscopy and cytometry. F.P. was supported by HFSP long-term fellowship (LT000111/2021-L) and EMBO long-term fellowship (ATLF 716-2020). M.-F.S. was supported by a Marie Skłodowska-Curie Innovative Training Network (grant 813327 'ChromDesign') and Fondation de Recherche Medicale. H.R. was supported by La Ligue Contre le Cancer. This work in the Cavalli laboratory was supported through grants from the European Research Council (advanced grants 3DEpi and WaddingtonMemory), the 'Agence Nationale pour la Recherche' (Cell-ID grant from the France 2030 program with references (ANR-24-EXCI-0002 and ANR-24-EXCI-0004); LIVCHROM grant ANR-21-CE45-0011), the European E-RARE NEURO DISEASES grant 'IMPACT', the Fondation ARC (EpiMM3D), the Fondation pour la Recherche Médicale (EQU202303016280), the MSD Avenir Foundation (project EpiMuM-3D) and the French National Cancer Institute (INCa, PIT-MM grant INCA-PLBIO18-362).

## Author contributions

G.C. and F.P. conceived the study. G.C. supervised the project and F.P. designed and carried out wet-lab experiments and analyzed all data. M.-F.S. optimized gastruloid culture, performed NPC differentiation and helped to carry out FACS, ChIP–seq and ATAC–seq experiments, as well as Micro-C data analysis. S.D. helped to handle cell culture, carry out ChIP–qPCR, ChIP–seq, RNA-seq, Micro-C, western blot and FACS experiments on ESCs, and performed and analyzed immunofluorescence in ESCs. M.D.S. and D.J. designed biophysical modeling. M.D.S. conducted polymer simulations and analyses. H.R. performed YY1 ChIP–seq and participated in setting hyperacetylation conditions. F.B. supervised H.R. F.P. wrote the manuscript with input from all authors.

## Competing interests

All authors declare no competing interests.

## Additional information

**Extended data** is available for this paper at https://doi.org/10.1038/s41588-025-02489-4.

**Correspondence and requests for materials** should be addressed to Giacomo Cavalli.

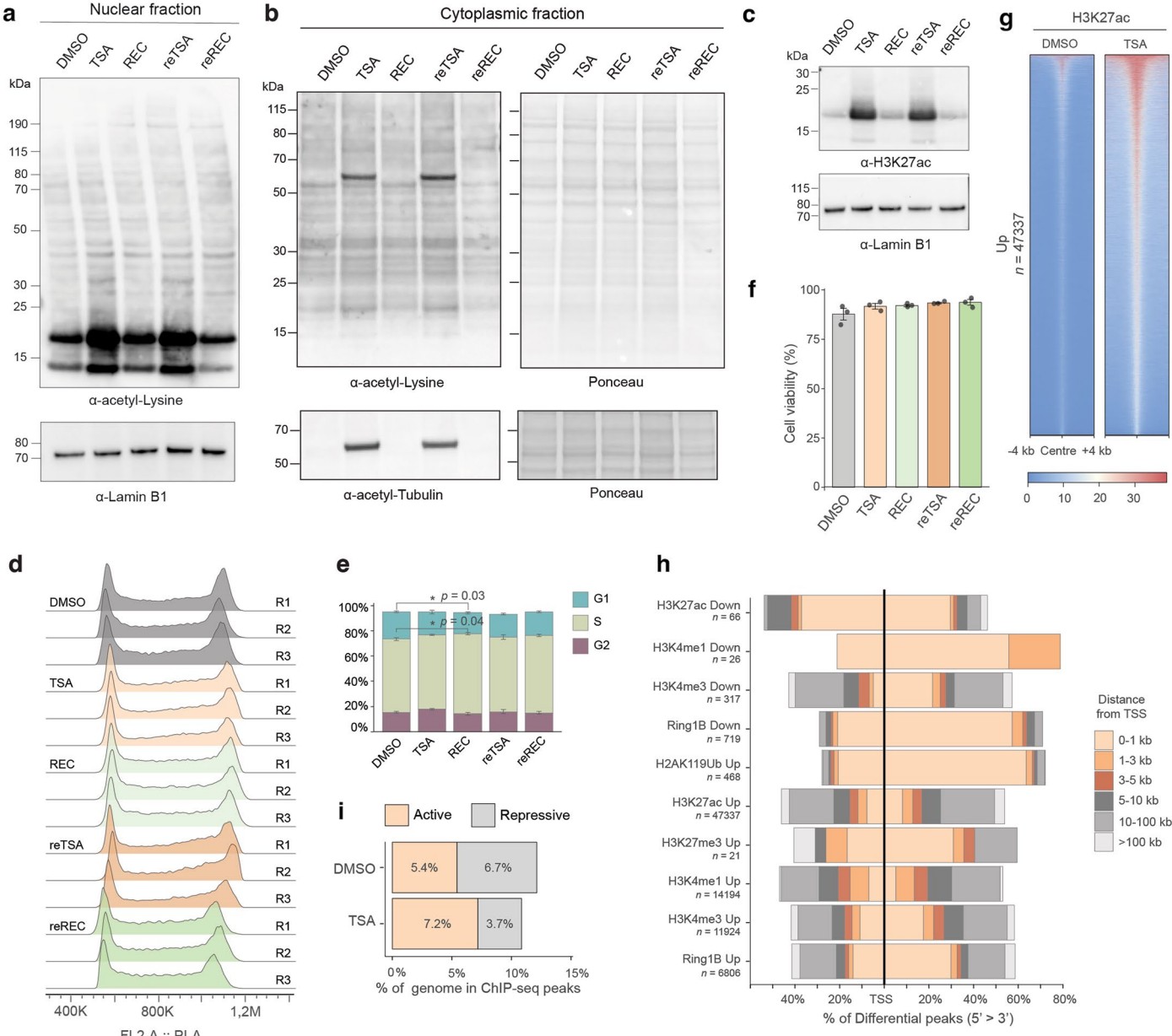

**Extended Data Fig. 1 | Characterization of the effect of TSA treatment on histone modifications and cell cycle in ESCs. a**, Western blots showing the levels of lysine acetylation in nuclear extracts in the following conditions: DMSO control, TSA treatment, 24-h recovery following TSA washout (REC), sequential TSA treatment (reTSA) and 24-h recovery from the second TSA treatment (reREC). Lamin B1 is shown as loading control. **b**, Western blots showing the levels of lysine acetylation (top panel) and levels of acetylated tubulin (bottom panel) in cytoplasmic protein extracts in conditions as in **a**. Ponceau staining is shown as loading control on the right. **c**, Western blot showing the levels of H3K27 acetylation in nuclear extracts in conditions as in **a**. Lamin B1 is used as loading control. **d**, Cell cycle analysis by flow cytometry following propidium iodide staining in conditions as in **a**. **e**, Fraction of cells in G1, S and G2/M at each time point of the TSA–recovery treatment course. *P*-values correspond to two-tailed unpaired t-test, *n* = 3 biological replicates, error bars show standard deviation. **f**, Cell cycle viability counts in conditions as in **a**. Error bars show ±s.e.m., *n* = 3 biological replicates. **g**, Heatmaps showing H3K27ac ChIP-seq signal at differential peaks in TSA. **h**, Bar plots showing the distance of differential ChIP–seq peaks in TSA (H3K27ac, H3K4me3, H3K4me1, H3K27me3, H2AK119ub, H3K9me3) from transcription start sites (TSS). **i**, Bar plots showing percentage of the genome in ChIP–seq peaks intervals of active and repressive histone marks.

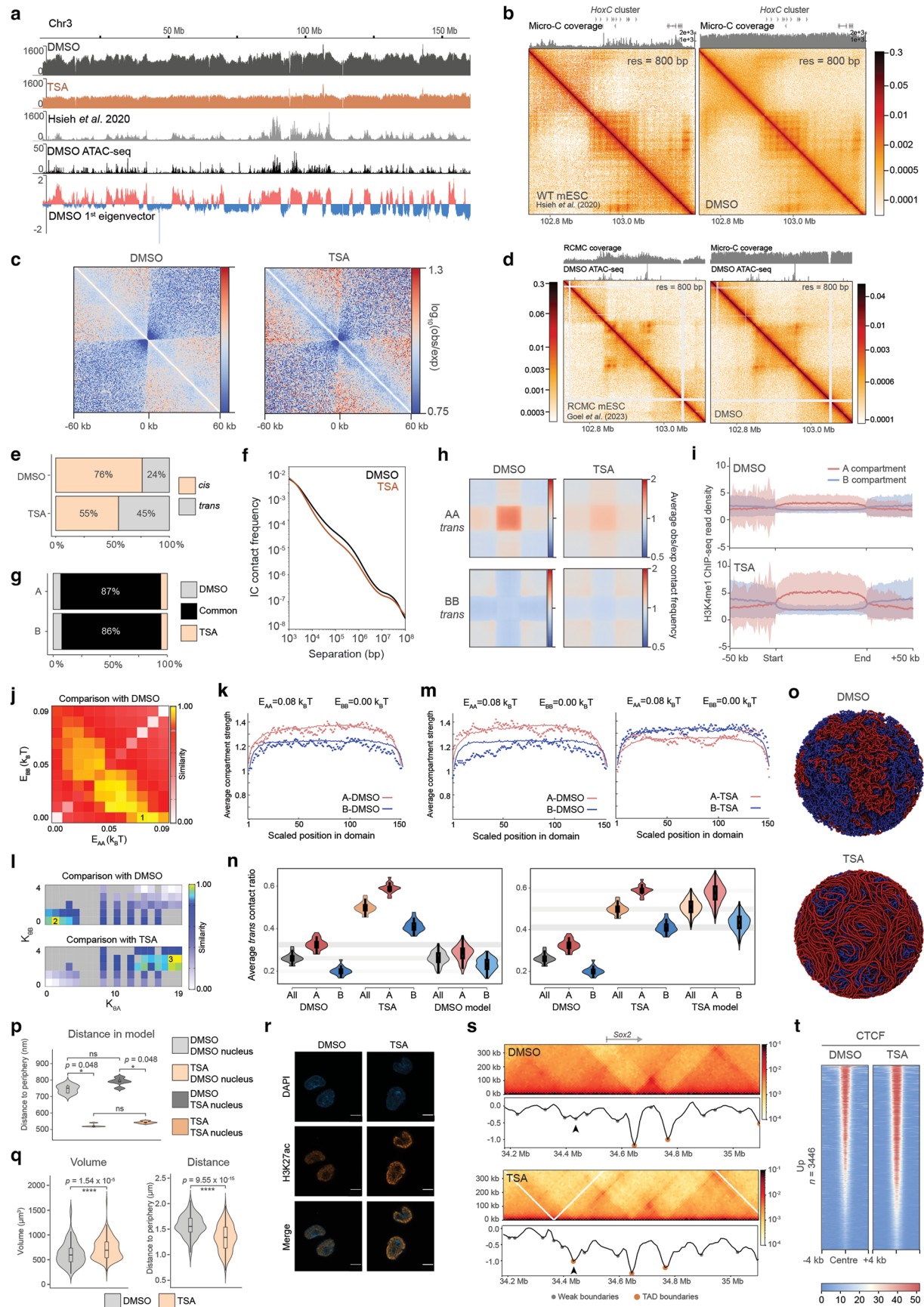

**Extended Data Fig. 2 | See next page for caption.**

**Extended Data Fig. 2 | The effect of HDAC inhibition on genome folding. a**, Total Micro-C coverage in DMSO, TSA and data from ref. [17] on chromosome 3, along with ATAC–seq signal and eigenvector tracks in DMSO. **b**, Micro-C maps showing contacts over the *HoxC* cluster in WT mESCs[17] and in DMSO. Corresponding Micro-C coverage tracks are displayed above the maps. **c**, On-diagonal pile-ups centered at microcompartment loop anchors identified in ref. [19] at the *Klf1, Sox2* and *Nanog* loci (resolution = 600 bp). **d**, Region capture Micro-C (RCMC) map (left) and Micro-C map (right) from this study showing contacts over the *Sox2* locus in WT mESCs[19] and in DMSO. RCMC data have been downsampled to match depth of the Micro-C map in the capture region. Corresponding RCMC or Micro-C coverage tracks are displayed above the maps. **e**, Ratio of *cis* versus *trans* contacts in Micro-C datasets. **f**, Micro-C contact frequency plotted against genomic separation. **g**, Proportion of unique versus common genomic regions assigned to A and B compartments in DMSO and TSA. **h**, Aggregate plots of homotypic interactions between and A and B compartments in *trans*. **i**, Metaplots showing H3K4me1 ChIP–seq read density over A and B compartments (bin size = 1 kb). Shading corresponds to standard deviation of the mean. **j**, Heatmap of the similarity scores between compartment-strength profiles in DMSO Micro-C and models for different parameter sets ($E_{AA}$, $E_{BB}$). **k**, A- and B-specific compartment-strength profiles from DMSO Micro-C data (lines) and single-chain simulations with optimized attraction energies (points). **l**, Heatmap of similarity scores between the median of chromosome-averaged *trans*-contact ratio in DMSO and TSA Micro-C datasets and the models for different parameter sets ($K_{\theta A}$, $K_{\theta B}$). Compartment-specific attraction energies ($E_{AA}$, $E_{BB}$) were maintained equal to the

single-chain optimized values. Gray entries indicate untested parameter sets. **m**, Compartment strength profiles for *trans*-ratio optimized models for DMSO (left) and TSA (right). **n**, Distribution of *trans*-contact ratio per chromosome in DMSO and TSA Micro-C data ($n = 19$), and the optimal models for DMSO (left) and TSA (right) ($n = 600$ corresponding to 20 simulated chains in 30 replicates). Box plots show median (central line), the 25th and 75th percentiles (box limits) and 1.5× IQR (whiskers). Outliers are not shown. **o**, Example configurations of modeled nuclei corresponding to DMSO (top) and TSA (bottom) conditions. Red and blue beads represent A and B chromatins, respectively. **p**, Distance to the periphery for all particles in the A-compartment during the last quarter of the trajectory in simulations (between 3 and 4 h). Plotted values are average per replicate ($n = 5$). TSA nucleus represents larger nuclei of 49.04 sigma (2657.81 nm radius) to take into account the 5% increase in nuclear volume observed in TSA. *P*-values were derived from unpaired two-sided Wilcoxon test. Box plots elements are as in **n**. **q**, Violin plots showing nuclear volume (left) and mean distance of H3K27ac spots to periphery (right) in DMSO ($n = 154$) and TSA nuclei ($n = 169$). Combined result of two biological replicates is shown (unpaired two-tailed Wilcoxon test; ****$P < 0.0001$). Box plot elements are as in **n**. **r**, Representative images of H3K27ac immunofluorescence in DMSO and TSA nuclei (scale bar = 5 µm). **s**, Micro-C maps and insulation curves showing a new topologically associating domain (TAD) boundary forming at the *Sox2* locus upon TSA treatment (resolution = 10 kb). **t**, Heatmaps showing CTCF ChIP–seq signal at differential TSA peaks.

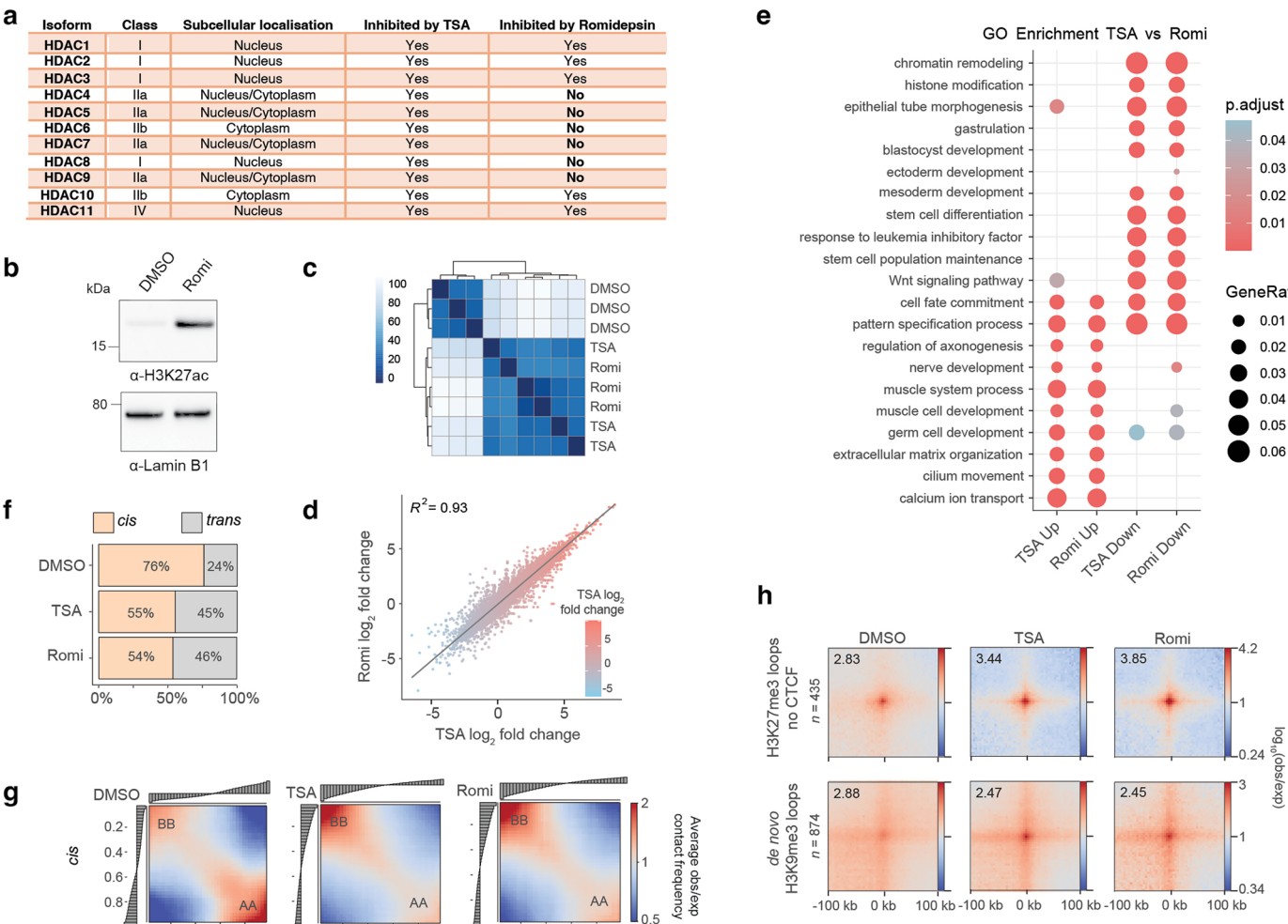

**Extended Data Fig. 3 | Comparison of the effects of TSA and Romidepsin on gene expression and genome architecture in mESCs. a**, HDACs inhibited by TSA and Romidepsin. Data from ref. 26. **b**, Western blot showing Romidepsin-induced (Romi) H3K27 hyperacetylation in nuclear extracts. Lamin B1 is shown as loading control. **c**, Sample distance matrix based on RNA-seq data showing high similarity of transcriptional response to TSA versus Romidepsin. **d**, Scatterplot showing correlation between transcriptomic changes induced by TSA and Romidepsin. Shading represents log$_2$ fold change in TSA. Fitted line shows linear regression. **e**, GO enrichment among up- and downregulated genes in TSA and Romidepsin. *P*-values correspond to one-sided Fisher's exact test corrected for FDR with the BH method. **f**, Ratio of *cis* versus *trans* contacts in DMSO, TSA and Romidepsin Micro-C datasets. **g**, Saddle plots of compartment interactions in *cis*. **h**, Aggregate plots of Micro-C signal in DMSO, TSA and Romidepsin at non-CTCF Polycomb loops (top) and at de novo H3K9me3 loops (bottom) (resolution = 4 kb).

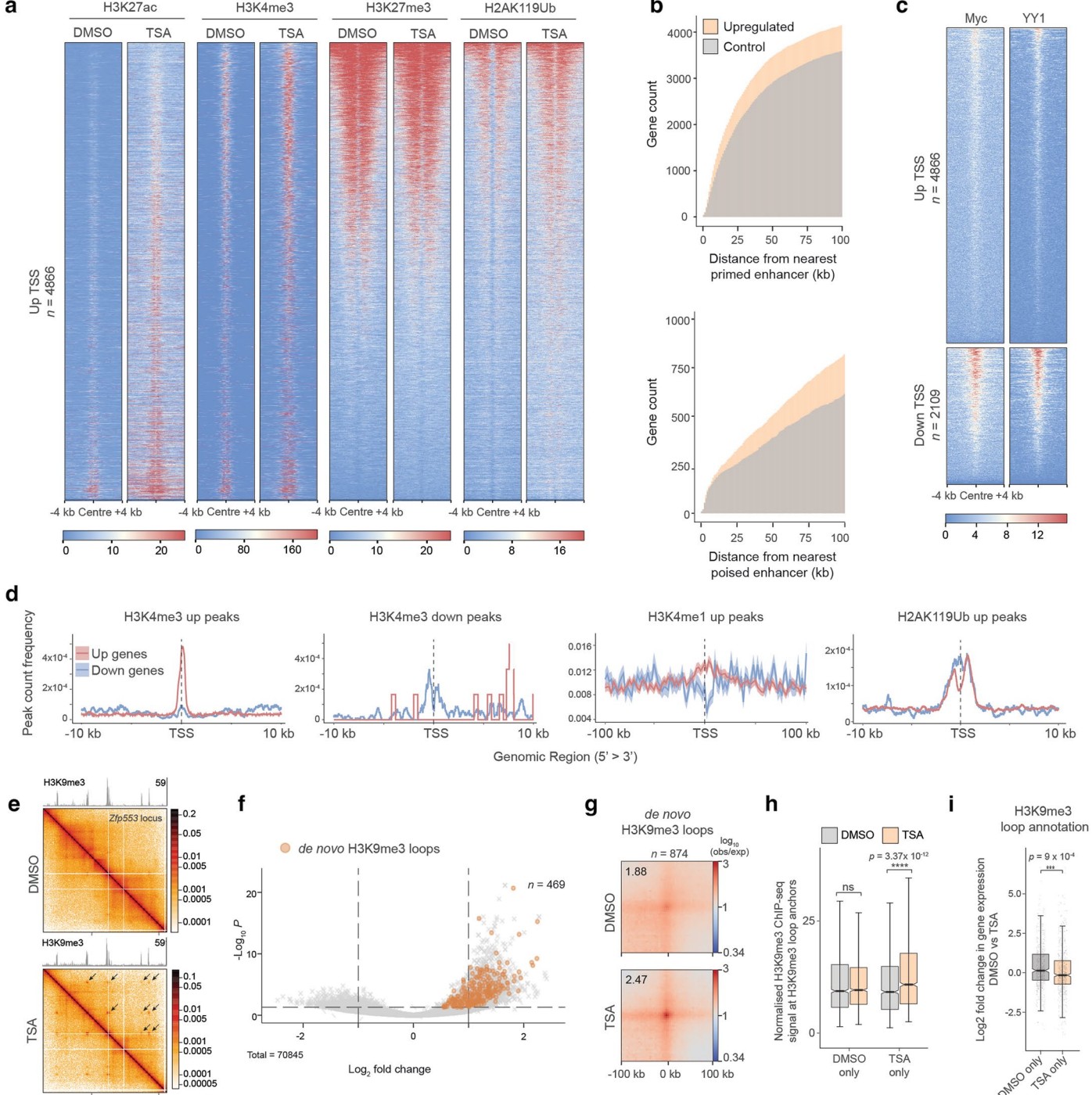

**Extended Data Fig. 4 | Chromatin changes in TSA near deregulated TSSs.**
**a**, Heatmaps showing H3K27ac, H3K4me3, H3K27me3 and H2AK119ub signal in DMSO and TSA around transcription start sites (TSS) of upregulated genes. **b**, Cumulative histogram showing genomic distance between upregulated gene promoters and the nearest primed (top) or poised (bottom) enhancer. Control genes represent an expression-matched gene set that does not increase in expression. **c**, Heatmaps showing Myc and YY1 ChIP–seq signal around up- and downregulated TSSs. **d**, Differential ChIP–seq peak count frequency distribution relative to TSSs of up- and downregulated genes. **e**, Micro-C contact maps showing extensive H3K9me3-associated differential looping at the *Zfp553* locus in TSA. Corresponding H3K9me3 ChIP–seq signal is displayed above. **f**, Volcano plot of differential loops between DMSO and TSA with de novo

H3K9me3 TSA loops highlighted in orange. Positive log₂ fold change indicates stronger interaction in TSA. *P*-values were derived from Wald test and were corrected for multiple testing with BH method. **g**, Pile-up of Micro-C signal (resolution = 4 kb) around de novo H3K9me3 loops that form in TSA. **h**, Box plot showing the normalized H3K9me3 signal at anchors of de novo H3K9me3 TSA loops. *P*-values correspond to unpaired two-tailed Wilcoxon test, *n* = 1526 for DMSO-only and *n* = 1575 for TSA-only loop anchors. Data shown are the median, with hinges corresponding to IQR and whiskers extending to the lowest and highest values within 1.5× IQR. **i**, Log₂ fold change in gene expression (TSA versus DMSO) of genes that are nearest DMSO-only (*n* = 527 genes) or TSA-only (*n* = 604 genes) H3K9me3 loop anchors. *P*-values correspond to unpaired two-tailed t-test. Box plot elements are as in **h**.

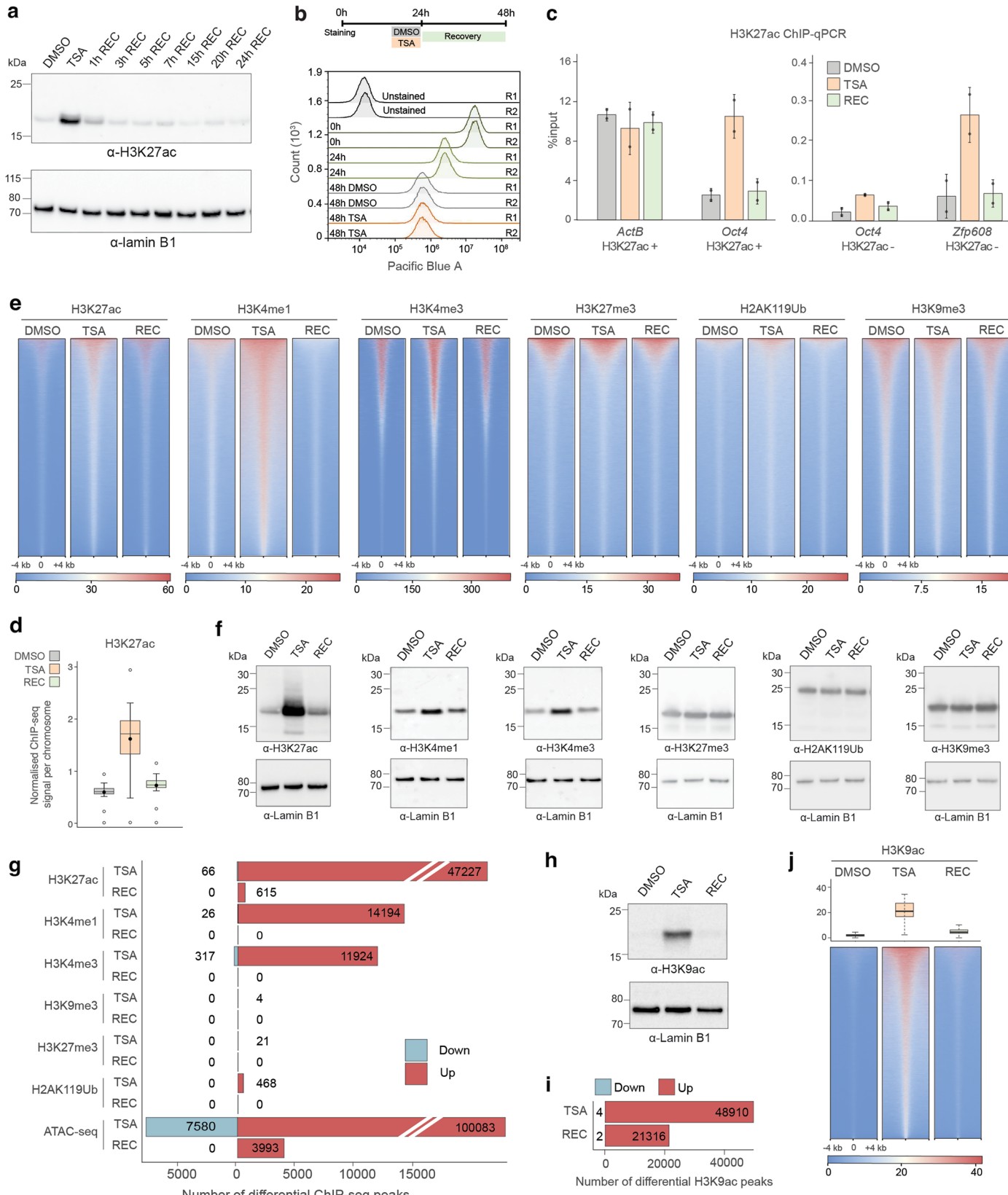

**Extended Data Fig. 5 | See next page for caption.**

**Extended Data Fig. 5 | Epigenetic landscape recovery. a**, Western blot showing H3K27 acetylation in DMSO, TSA and the indicated time intervals following TSA washout. Lamin B1 is shown as loading control. **b**, Cell generation tracing using flow cytometry following DMSO and TSA treatment. Two biological replicates are shown. **c**, ChIP–qPCR showing H3K27ac signal in DMSO, TSA and 24-h recovery (REC) conditions at the *ActB* and *Oct4* promoters that have high levels of acetylation (left) and at the *Zfp608* promoter and an *Oct4* upstream region largely devoid of acetylation (right). Bar charts are mean of two biological replicates ±s.e.m., scatterplot shows individual replicates. **d**, Spike-in normalized H3K27ac ChIP–seq signal per chromosome (*n* = 19) in replicate 3. Data shown are the median, with hinges corresponding to IQR and whiskers extending to the lowest and highest values within 1.5× IQR. **e**, Heatmaps showing scaled ChIP–seq signal per epitope in merged replicates at all enrichment sites in DMSO, TSA and REC conditions. **f**, Western blots showing total level of histone modifications in nuclear extracts. Lamin B is shown as loading control. **g**, Diverging bar chart showing the number of differential ChIP–seq and ATAC–seq peaks identified by DESeq2 analysis in TSA versus DMSO (TSA) and 24-h recovery versus DMSO (REC) conditions. **h**, Western blot showing TSA-induced H3K9 hyperacetylation and recovery in nuclear extracts. Lamin B1 is shown as loading control. **i**, Diverging bar chart showing the number of differential H3K9ac ChIP–seq peaks identified by DESeq2 analysis. **j**, Heatmaps showing scaled H3K9ac ChIP–seq signal in merged replicates at all enrichment sites in DMSO, TSA and REC conditions. Box plots (elements as in **d**) of normalized ChIP–seq signal at all H3K9ac sites (*n* = 60618) are shown above heatmaps.

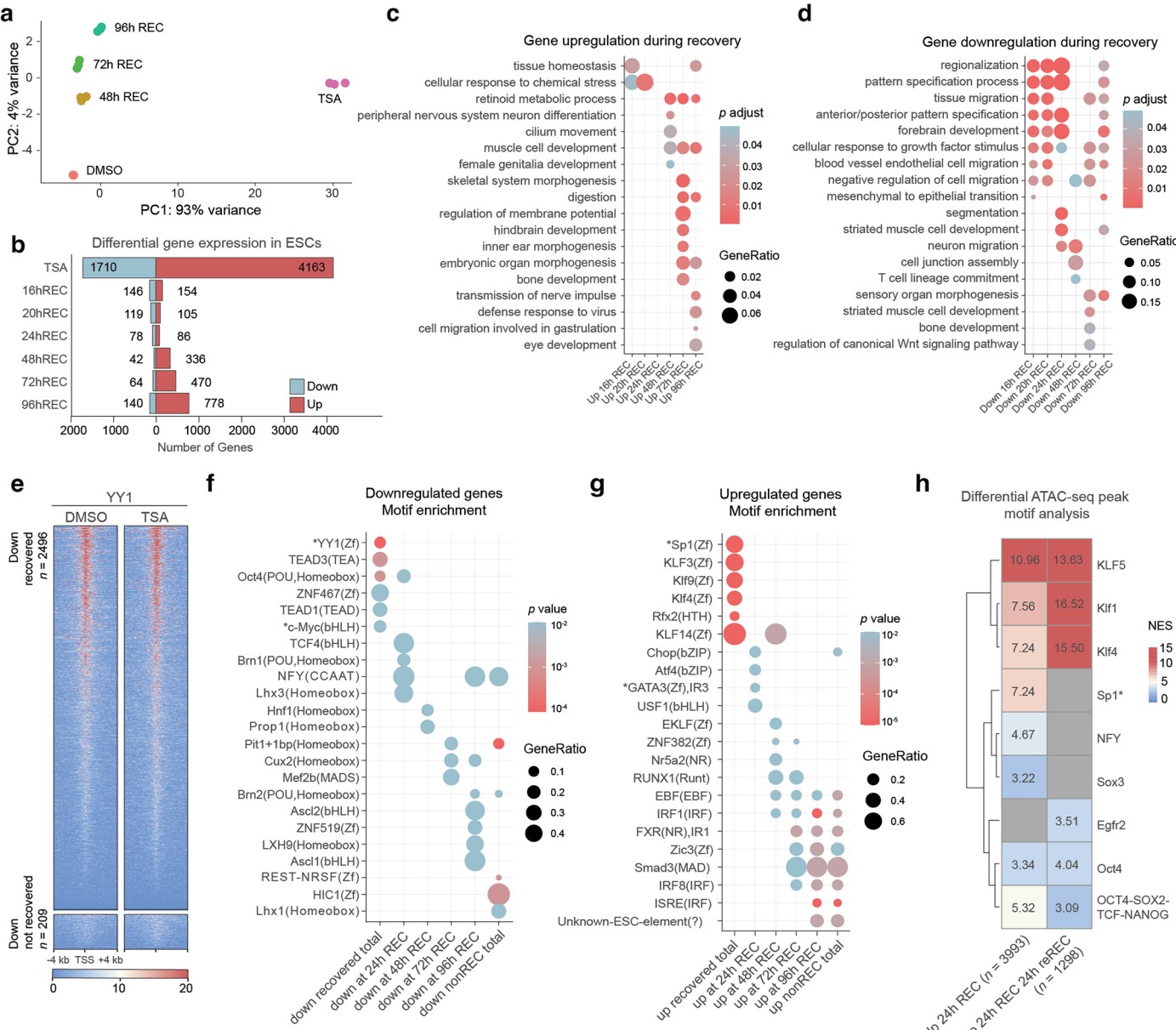

**Extended Data Fig. 6 | Long-term and non-histone effects of TSA treatment.** **a**, Principal component analysis (PCA) analysis showing increasing distance between DMSO and recovery (REC) samples with increasing recovery time. **b**, Diverging bar plot showing the number of differentially expressed genes at the indicated recovery times. **c**,**d**, Bubble plots showing GO (biological process) enrichment among up- (**c**) and downregulated (**d**) genes at different recovery intervals. *P*-values correspond to one-sided Fisher's exact test corrected for FDR with the BH method. **e**, Normalized YY1 ChIP-seq signal at downregulated TSSs that recover (top) and do not recover (bottom) in DMSO and TSA. **f**,**g**, Bubble plots showing HOMER motif enrichment analysis at down- (**f**) and upregulated (**g**) gene promoters of genes that recover from and genes that remain deregulated following TSA treatment. Direct class I, II or IV HDAC targets that have detectable expression levels in mouse ESCs are marked with asterisk (*). *P*-values were computed using uncorrected one-sided binomial test. **h**, Heatmap showing HOMER motif enrichment analysis (NES = normalized enrichment score) at differential ATAC-seq peaks at 24-h recovery following the first and second TSA treatments.

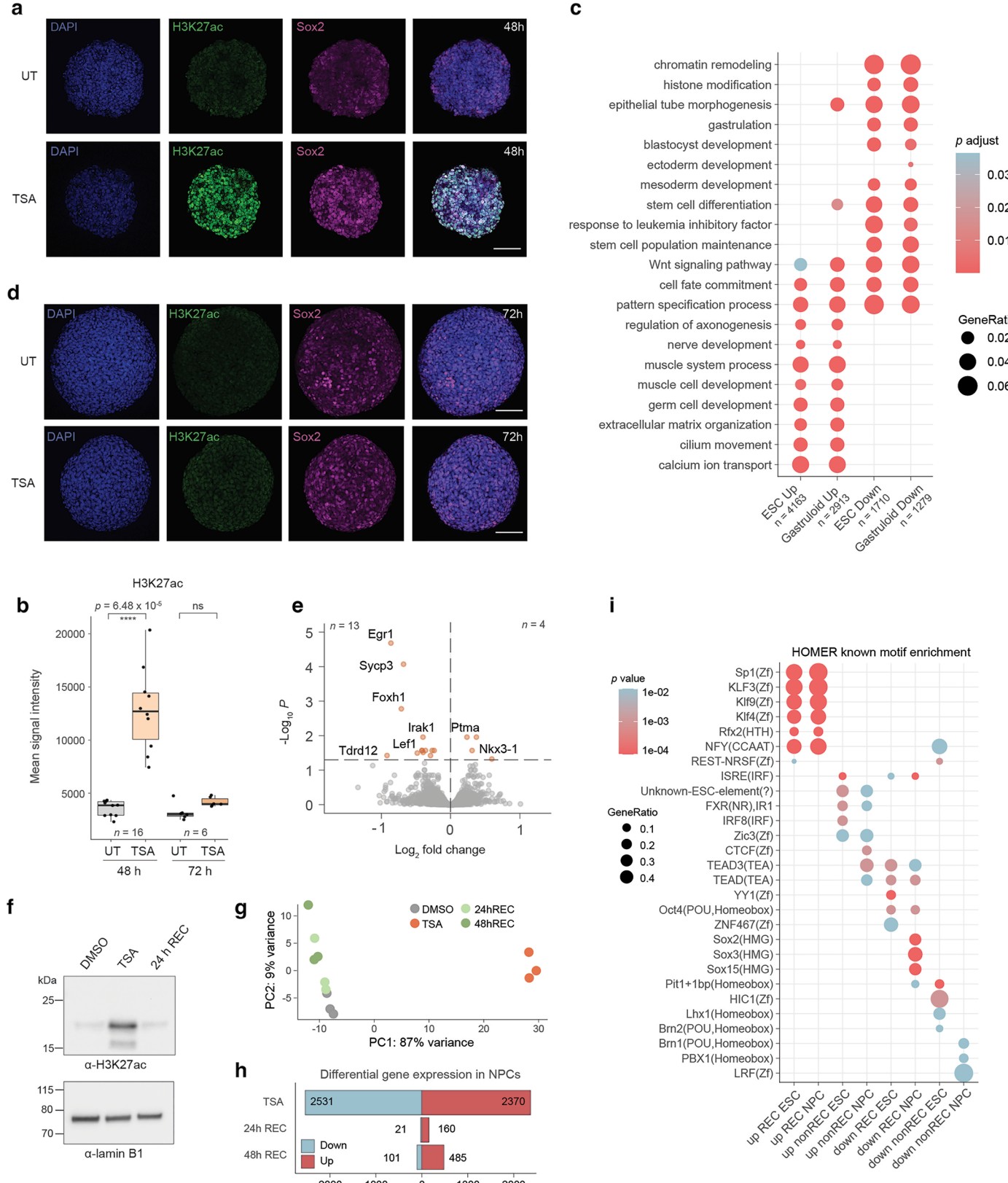

**Extended Data Fig. 7 | See next page for caption.**

**Extended Data Fig. 7 | HDAC inhibition-induced chromatin and gene expression changes in gastruloids and neural progenitor cell (NPCs).**
**a**, Representative images of H3K27ac and Sox2 immunostaining in 48-h gastruloids with (bottom) or without (top) TSA treatment (scale bar = 100 µm). **b**, Quantification of H3K27ac signal intensity in early (48 h, 72 h) gastruloids with and without TSA treatment ($n$ = number of gastruloids measured). Data shown are the median, with hinges corresponding to IQR and whiskers extending to the lowest and highest values within 1.5× IQR. $P$-values were derived from unpaired two-tailed Wilcoxon test (NS>0.05, ****$P$<0.0001). **c**, GO enrichment of terms related to development among differentially expressed genes in mESCs and 48-h gastruloids after 4 h of TSA treatment. $P$-values correspond to one-sided Fisher's exact test corrected for FDR with the BH method. **d**, Representative images of H3K27ac and Sox2 immunostaining in 72-h gastruloids with (bottom) or without

(top) TSA treatment (scale bar = 100 µm). **e**, Volcano plot showing differentially expressed genes in 120-h gastruloids with versus without TSA treatment (significance cutoff: adjusted $p$-value (derived from Wald test and corrected for multiple testing with BH method) < 0.05). **f**, Western blot showing TSA-induced H3K27 hyperacetylation and recovery in nuclear extracts from NPCs. Lamin B1 is shown as loading control. **g**, PCA showing increasing distance between DMSO and recovery (REC) samples with increasing recovery time in NPCs. **h**, Diverging bar plot showing the number of differentially expressed genes in NPCs at the indicated conditions. **i**, Bubble plots showing HOMER motif enrichment analysis at promoters of genes that recover from and genes that remain deregulated following TSA treatment, in ESCs and NPCs. $P$-values were computed using uncorrected one-sided binomial test.

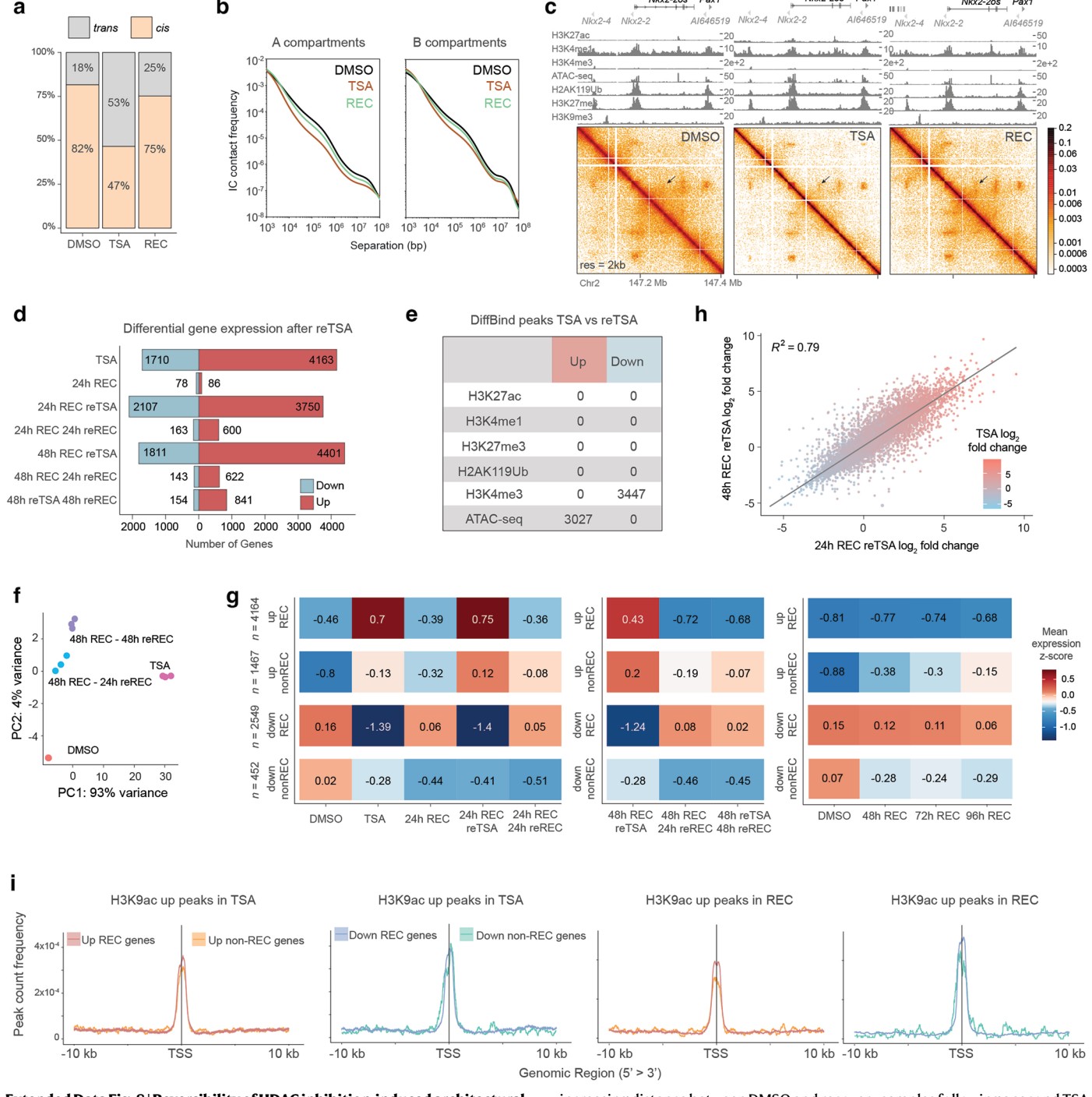

**Extended Data Fig. 8 | Reversibility of HDAC inhibition-induced architectural changes and response to sequential TSA treatment. a**, Ratio of *cis* versus *trans* contacts in Micro-C datasets. **b**, Contact frequency in Micro-C plotted against genomic separation by compartment. **c**, Micro-C maps at the *Nkx2-2* locus showing incomplete architectural recovery. ChIP–seq tracks of the corresponding condition are shown above. **d**, Diverging bar plot showing the number of differentially expressed genes at the indicated conditions. **e**, Differential ChIP–seq peaks identified by DESeq2 between the first (TSA) and second TSA treatments (reTSA), with 24 h or recovery in between. **f**, PCA showing

increasing distance between DMSO and recovery samples following a second TSA treatment with increasing recovery time. **g**, Heatmaps showing mean expression z-scores of recovered and not recovered genes through the TSA–recovery treatment course. **h**, Scatterplot showing correlation between transcriptomic changes induced by sequential TSA treatment following 24-h and 48-h recovery. Fitted line represents linear regression. **i**, Differential H3K9ac ChIP–seq peak frequency distribution relative to TSSs of up- (left) and downregulated (right) genes that recover or do not recover.

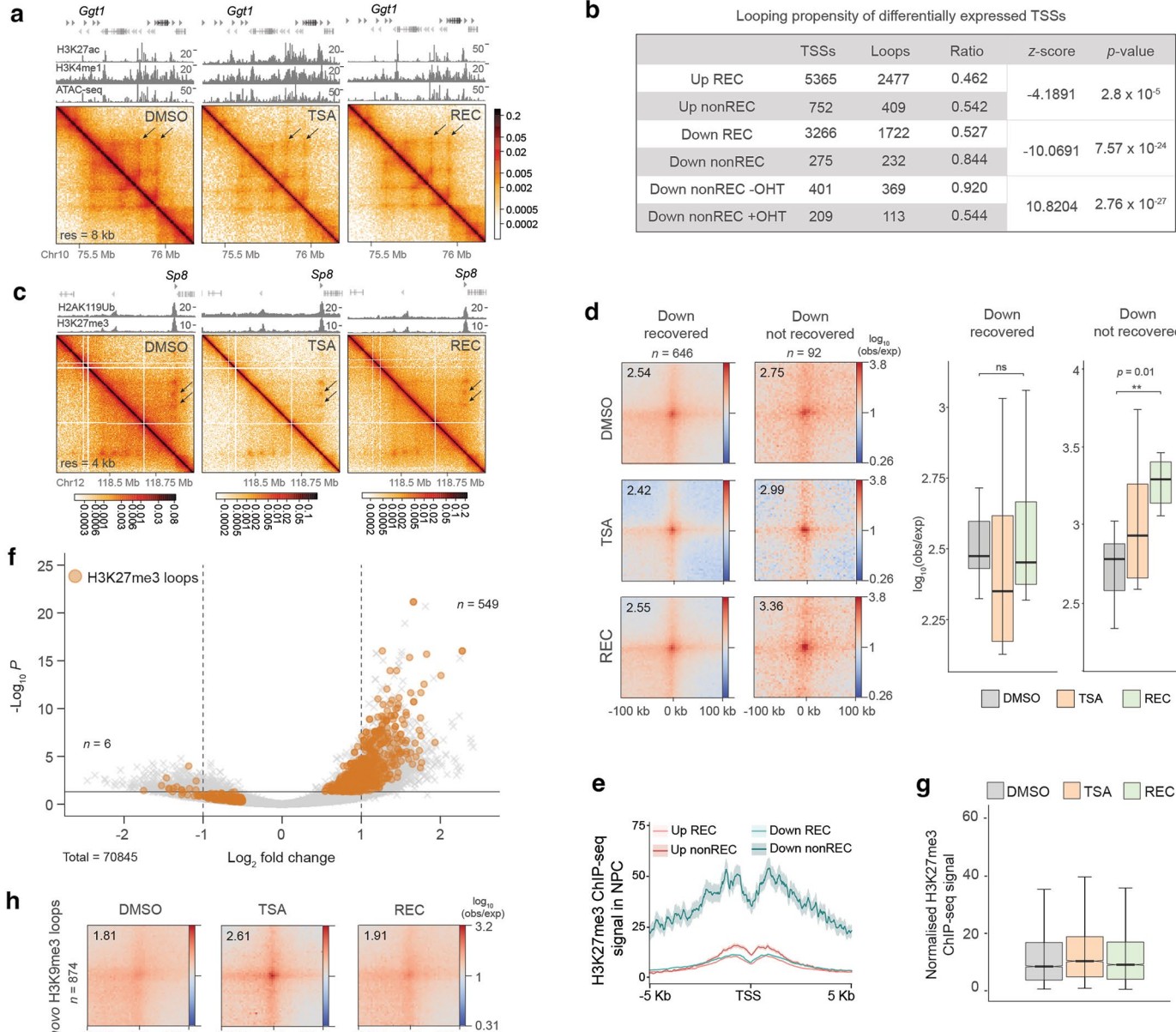

**Extended Data Fig. 9 | A second TSA treatment triggers memory in mESCs.**
**a**, Micro-C contacts at the *Ggt1* locus that shows sustained transcriptional upregulation. Corresponding H3K27ac, H3K4me1 and ATAC–seq signal are displayed above. **b**, Number of DMSO loops within 5 kb vicinity of different TSS groups. Z-scores and *p*-values were calculated using two-tailed two-proportions z-test. **c**, Micro-C maps of Polycomb-mediated contacts at the *Sp8* locus that shows sustained transcriptional downregulation. Corresponding H2AK119ub and H3K27me3 signal are displayed above. **d**, Heatmaps showing aggregate plots of Micro-C signal (left) and corresponding box plots (right) at loops whose anchors overlap with downregulated TSSs that recover or do not recover (resolution = 4 kb). Quantification of piled-up loop signal (*n* = 9 corresponding to central 3 × 3 pixels) is shown on the right (paired two-tailed t-test; ns > 0.05,

**p < 0.01). Data shown are the median, with hinges corresponding to IQR and whiskers extending to the lowest and highest values within 1.5× IQR. **e**, Metaplot showing mean H3K27me3 ChIP–seq signal in NPCs at TSSs of upregulated genes that recover and do not recover, and downregulated genes that recover and do not recover. Shading represents ±s.e.m. **f**, Volcano plot of differential loops between DMSO and TSA. H3K27me3 loops are highlighted in orange. Positive log₂ fold change indicates stronger interaction in TSA. *P*-values were derived from Wald test and were corrected for multiple testing with the BH approach. **g**, Box plot showing normalized H3K27me3 signal at anchors of H3K27me3 loops in DMSO, TSA and recovery. Box plot elements are as in **d**. **h**, Pile-up of Micro-C signal (resolution = 4 kb) around de novo H3K9me3 loops that form in TSA.

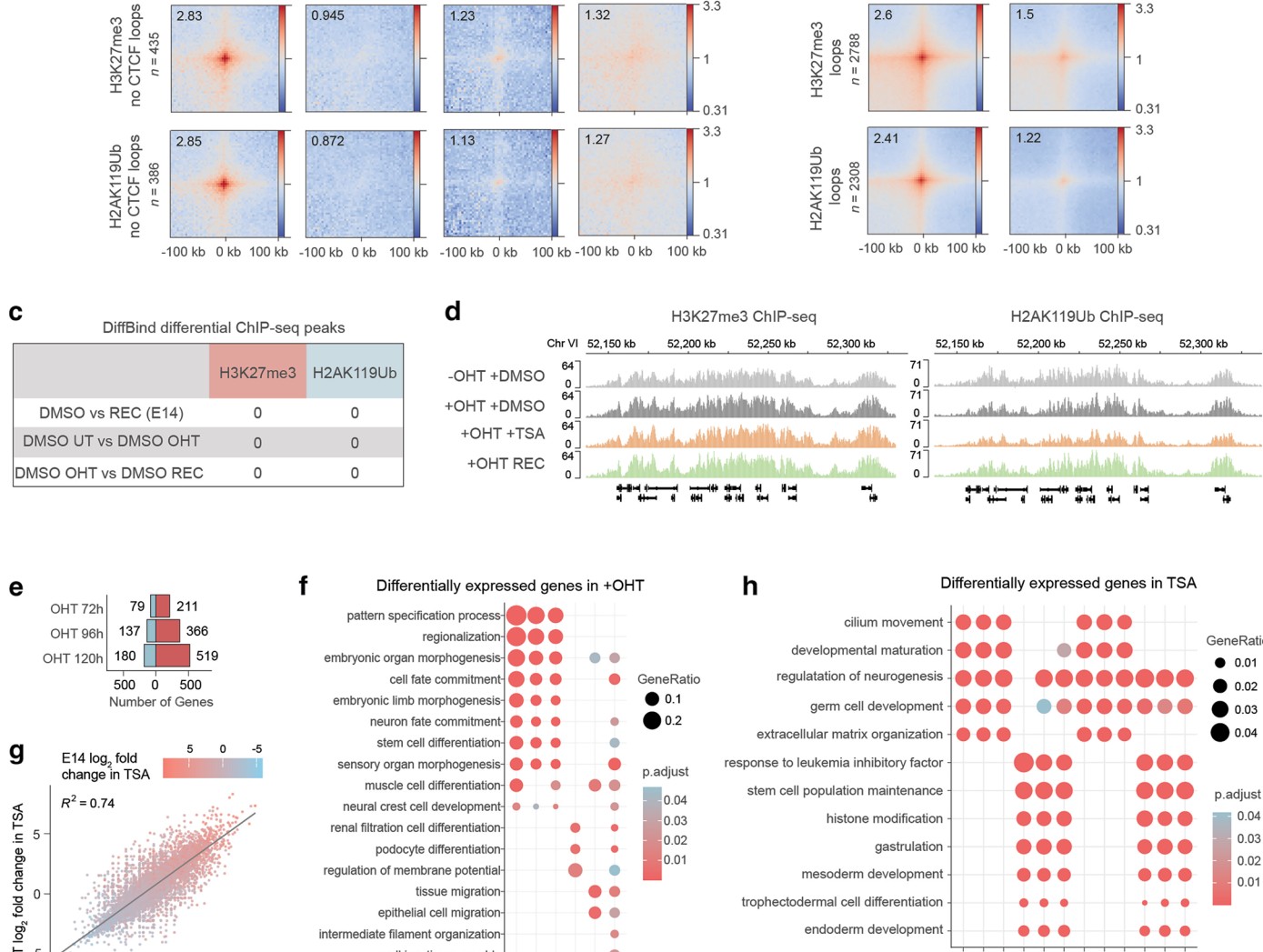

**Extended Data Fig. 10 | Changes in the transcriptome and genome conformation upon PCGF2 depletion. a,b,** Aggregate plots of Micro-C signal at non-CTCF (**a**) and all (**b**) H3K27me3 loops (top panels) and H2AK119ub loops (bottom panels) (resolution = 4 kb) in *Pcgf2^fl/fl^* cells. **c,** Differential ChIP–seq peaks identified by DESeq2 between different treatment conditions in *Pcgf4^−/−^*; *Pcgf2^fl/fl^* mESC cells. **d,** Genomic snapshot of normalized H3K27me3 and H2AK119ub signal over the *HoxA* cluster. **e,** Diverging bar plot showing the number of differentially expressed genes upon prolonged tamoxifen (OHT) treatment in *Pcgf4^−/−^*; *Pcgf2^fl/fl^* mESC cells. **f,** GO enrichment among deregulated

genes following prolonged OHT treatment. *P*-values correspond to one-sided Fisher's exact test corrected for FDR with the BH method. **g,** Scatterplot showing correlation between transcriptomic changes induced by TSA with or without OHT treatment in *Pcgf2^fl/fl^* cells. Shading represents log₂ fold change upon TSA treatment in wild type E14 mESCs. Fitted line shows linear regression. **h,** GO enrichment among differentially expressed genes in the first and second TSA treatments, in E14, *Pcgf2^fl/fl^* −OHT and *Pcgf2^fl/fl^* +OHT cells. *P*-values correspond to one-sided Fisher's exact test corrected for FDR with the BH method.

# Reporting Summary

## Statistics

For all statistical analyses, confirm that the following items are present in the figure legend, table legend, main text, or Methods section.

| n/a | Confirmed | |
|---|---|---|
| ☐ | ☒ | The exact sample size (*n*) for each experimental group/condition, given as a discrete number and unit of measurement |
| ☐ | ☒ | A statement on whether measurements were taken from distinct samples or whether the same sample was measured repeatedly |
| ☐ | ☒ | The statistical test(s) used AND whether they are one- or two-sided<br>*Only common tests should be described solely by name; describe more complex techniques in the Methods section.* |
| ☒ | ☐ | A description of all covariates tested |
| ☐ | ☒ | A description of any assumptions or corrections, such as tests of normality and adjustment for multiple comparisons |
| ☐ | ☒ | A full description of the statistical parameters including central tendency (e.g. means) or other basic estimates (e.g. regression coefficient) AND variation (e.g. standard deviation) or associated estimates of uncertainty (e.g. confidence intervals) |
| ☐ | ☒ | For null hypothesis testing, the test statistic (e.g. *F*, *t*, *r*) with confidence intervals, effect sizes, degrees of freedom and *P* value noted<br>*Give P values as exact values whenever suitable.* |
| ☒ | ☐ | For Bayesian analysis, information on the choice of priors and Markov chain Monte Carlo settings |
| ☐ | ☒ | For hierarchical and complex designs, identification of the appropriate level for tests and full reporting of outcomes |
| ☒ | ☐ | Estimates of effect sizes (e.g. Cohen's *d*, Pearson's *r*), indicating how they were calculated |

*Our web collection on statistics for biologists contains articles on many of the points above.*

## Software and code

Policy information about availability of computer code

| Data collection | Zeiss ZEN Blue (v3.8–3.12)<br>CytExpert (v2.4)<br>NovoExpress (v1.6.3)<br>ChemiDoc Imaging Systems (BioRad) |
|---|---|
| Data analysis | Data analysis was performed as described in the Methods of the paper. Code for the novel analyses was deposited at https://github.com/cavallifly/Paldi_et_al_2024 and at https://doi.org/10.5281/zenodo.17608120<br><br>Softwares used in this study:<br>Subread (v2.0.6)<br>DEseq2 (1.42.1)<br>clusterProfiler (v4.10.1)<br>EnhancedVolcano (v1.20.0)<br>ggplot2 (v3.5.1)<br>HOMER (v4.10.0)<br>bowtie2 (v2.4.4)<br>samtools (v1.9)<br>Sambamba (v1.0)<br>IGV (v.2.16.1)<br>HiGlass (v1.11.8)<br>MACS3 (v3.0.3) |

```
diffBind (v3.12.0)
deepTools (v3.5.6)
ChIPseeker (v1.38.0)
TrimGalore (v0.6.10)
HiC-Pro (v3.1.0)
HiCRep (v1.12)
cooltools (v0.5.4)
mustache (v1.0)
bedtools (v2.31.1)
coolpuppy (v1.1.0)
Imaris (v10.1.1)
FlowJo (v10.10)
LightCycler (v1.5.1)
Fiji (v2.14.0)
```

For manuscripts utilizing custom algorithms or software that are central to the research but not yet described in published literature, software must be made available to editors and reviewers. We strongly encourage code deposition in a community repository (e.g. GitHub). See the Nature Portfolio guidelines for submitting code & software for further information.

## Data

Policy information about availability of data

All manuscripts must include a data availability statement. This statement should provide the following information, where applicable:
- Accession codes, unique identifiers, or web links for publicly available datasets
- A description of any restrictions on data availability
- For clinical datasets or third party data, please ensure that the statement adheres to our policy

All raw data were submitted to the National Library of Medicine's (NCBI) Sequence Read Archive (SRA) and  processed files were submitted to  Gene Expression Omnibus (GEO). All data can be retrieved under the GEO series GSE281151. Myc ChIP-seq dataset was published was downloaded from the GEO repository GSE90895. NPC H3K27me3 ChIP-seq was  downloaded from the GEO repository GSE262551.

## Research involving human participants, their data, or biological material

Policy information about studies with human participants or human data. See also policy information about sex, gender (identity/presentation), and sexual orientation and race, ethnicity and racism.

| | |
|---|---|
| Reporting on sex and gender | N/A |
| Reporting on race, ethnicity, or other socially relevant groupings | N/A |
| Population characteristics | N/A |
| Recruitment | N/A |
| Ethics oversight | N/A |

Note that full information on the approval of the study protocol must also be provided in the manuscript.

# Field-specific reporting

Please select the one below that is the best fit for your research. If you are not sure, read the appropriate sections before making your selection.

☒ Life sciences    ☐ Behavioural & social sciences    ☐ Ecological, evolutionary & environmental sciences

For a reference copy of the document with all sections, see nature.com/documents/nr-reporting-summary-flat.pdf

# Life sciences study design

All studies must disclose on these points even when the disclosure is negative.

| | |
|---|---|
| Sample size | No statistical methods were used to determine sample sizes. Instead, they were defined in compliance with gold standards in the field. Micro-C experiments were performed in 1 million cells. RNA-seq samples were isolated from a number of cells varying between 200 000 and 1 million. ChIP-seq and ChIP-qPCR were performed on approximately 4 million cells for TFs and 2 million cells for histone marks. Immunostainings were performed on ~10 gastruloids per replicate. ~100 cells  per condition were imaged for ESC immunofluorescence. ATAC-seq were performed on 100 000 cells. Western blots were produced with protein extracts from ~10^7 cells. ~20000 cells per condition were assayed by FACS. |
| Data exclusions | For RNA-seq analysis any gene with less than 10 reads was excluded from analysis because of insufficient statistical power in the downstream |

| Data exclusions | analyses. For immunofluorescence, gastruloids with abnormal symmetry were not chosen for imaging. |
|---|---|
| Replication | Micro-C experiments were performed in 5 (DMSO, TSA), 2 (24h recovery, Romidepsin) or 1 (PCGF2 -OHT +DMSO, +OHT +DMSO, +OHT +TSA, +OHT +REC) biological replicates. RNA-seq and gastruloid experiments were performed in biological triplicates. ChIP-seq, ChIP-qPCR and ATAC-seq were performed in biological duplicates except for H3K27ac ChIP-seq (3 biological replicates). ESC immunofluorescence was performed in biological duplicates. For sequencing-based experiments PCA and Spearman correlation was routinely performed to assess reproducibility. Reproducibility of Micro-C experiments was assessed using the Stratum-adjusted Correlation Coefficient from the HiCRep package. |
| Randomization | This study does not require randomization protocols. |
| Blinding | Blinding is not compatible with this study as the identity of control samples must be known for genomic data analyses. Data reported here are based on unbiased analysis. |

# Reporting for specific materials, systems and methods

We require information from authors about some types of materials, experimental systems and methods used in many studies. Here, indicate whether each material, system or method listed is relevant to your study. If you are not sure if a list item applies to your research, read the appropriate section before selecting a response.

## Materials & experimental systems

| n/a | Involved in the study |
|---|---|
| ☐ | ☒ Antibodies |
| ☐ | ☒ Eukaryotic cell lines |
| ☒ | ☐ Palaeontology and archaeology |
| ☒ | ☐ Animals and other organisms |
| ☒ | ☐ Clinical data |
| ☒ | ☐ Dual use research of concern |
| ☒ | ☐ Plants |

## Methods

| n/a | Involved in the study |
|---|---|
| ☐ | ☒ ChIP-seq |
| ☐ | ☒ Flow cytometry |
| ☒ | ☐ MRI-based neuroimaging |

## Antibodies

| Antibodies used | The following antibodies were used (from multiple lots over the course of the study):<br>- H3K4me1 (ActiveMotif #39297) - Wb: 1:5000, ChIP: 3μl<br>- H3K4me3 (Millipore #04-745) - Wb: 1:1000, ChIP: 3μl<br>- H3K9me3 (abcam #8898) - Wb: 1:2000, ChIP: 3μl<br>- H3K27me3 (ActiveMotif #39155) - Wb: 1:2500, ChIP: 3μl<br>- H3K27Ac (ActiveMotif #39133) - Wb: 1:7500, ChIP: 3μl<br>- H2AK119Ub (Cell Signalling #8240S) - Wb: 1:2000, ChIP: 3μl<br>- CTCF (Active Motif #61311) - ChIP: 5μl<br>- YY1 (abcam #109237) - ChIP: 8μl<br>- Pan-Acetyl-Lysine (Proteintech #66289-1-IG) - Wb: 1:1000,<br>- H3K9Ac (Millipore #07-352) - Wb: 1:7500, ChIP: 3μl<br>- Ring1B (Cell Signalling #5694) - ChIP: 5μl<br>- Acetyl-Tubulin (Sigma-Aldrich #T7451) - Wb: 1:2000<br>- Lamin B1 (abcam #ab16048) - Wb: 1:10000 |
|---|---|
| Validation | The antibodies were validated by the manufacturer as follows:<br>- H3K4me1 (ActiveMotif #39297) "Applications Validated by Active Motif: ChIP: 5 - 10 μg per ChIP WB*: 0.2 - 2 μg/ml dilution DB: 1 μg/ml dilution." All other information can be found at https://www.activemotif.com/catalog/details/61781/histone-h3k4me1-antibody-pab-4.<br>- H3K4me3 (Millipore #04-745) "Anti-trimethyl-Histone H3 (Lys4) Antibody, clone MC315 is a rabbit monoclonal antibody for detection of trimethyl-Histone H3 (Lys4) also known as H3K4me3, Histone H3 (tri methyl K4) & has been validated in WB, ChIP, DB, Mplex, ChIP-seq." All other information can be found at https://www.merckmillipore.com/FR/fr/product/Anti-trimethyl-Histone-H3-Lys4-Antibody-clone-MC315-rabbit-monoclonal,MM_NF-04-745.<br>- H3K9me3 (abcam #8898) "Every new batch of ab8898 is tested in house in ChIP." All other information can be found at https://www.abcam.com/en-us/products/primary-antibodies/histone-h3-tri-methyl-k9-antibody-chip-grade-ab8898.<br>- H3K27me3 (ActiveMotif #39155) "Applications Validated by Active Motif: ChIP: 5 - 10 μg per ChIP ChIP-Seq: 5 μg each ICC/IF: 2 μg/ml dilution IHC(FFPE): 2 μg/ml dilution WB*: 0.5 - 2 μg/ml dilution CUT&Tag: 1 μg per 50 μl reaction* CUT&RUN: 1 μg per 50 μl reaction" All other information can be found at https://www.activemotif.com/catalog/details/39155.<br>- H3K27Ac (ActiveMotif #39133) "Validated by ActiveMotif for: ChIP: 10 μg per ChIP, ChIP-Seq: 5 μg each, ICC/IF: 1 - 5 μg/ml dilution, WB*: 0.1 - 1 μg/ml dilution, CUT&Tag: 1 μg per 50 μl reaction." All other information can be found at https://www.activemotif.com/catalog/details/39133/histone-h3-acetyl-lys27-antibody-pab.<br>- H2AK119Ub (Cell Signalling #8240S) "This antibody has been validated using SimpleChIP® Enzymatic Chromatin IP Kits." All other information can be found at https://www.cellsignal.com/products/primary-antibodies/ubiquityl-histone-h2a-lys119-d27c4-rabbit-monoclonal-antibody/8240. |

-CTCF (Active Motif #61311) "Validated for: ChIP: 2 - 8 μl per ChIP,ChIP-Seq: 4 μg per ChIP, ICC/IF: 1:2,000 dilution, WB: 1:500-1:2,000 dilution, IHC(FFPE): 1:1000 dilution, CUT&Tag* 1 μl per 50 μl reaction, CUT&RUN: 1 μl per 50 μl reaction" All other information can found at https://www.activemotif.jp/documents/tds/61311.pdf
-YY1 (abcam #109237) Previously used in Dong et al. (2022) doi: 10.1093/nar/gkac230. All other information can be found at https://www.abcam.com/en-us/products/primary-antibodies/yy1-antibody-epr4652-nuclear-loading-control-ab109237.
-Pan-Acetyl-Lysine (Proteintech #66289-1-IG) "Tested Applications: WB, IF/ICC, ELISA; RecommendedDilutions: WB 1:500-1:3000 IF/ICC 1:50-1:500" All other information can be found at https://www.ptglab.com/fr/products/Pan-Acetylation-Antibody-66289-1-Ig.htm.
-H3K9Ac (Millipore #07-352) "Anti-acetyl-Histone H3 (Lys9) Antibody is a Rabbit Polyclonal Antibody for detection of acetyl-Histone H3 (Lys9) also known as H3K9Ac, Histone H3 (acetyl K9) and has been published and validated in ChIP, WB, Mplex." All other information can be found at https://www.merckmillipore.com/FR/fr/product/Anti-acetyl-Histone-H3-Lys9-Antibody,MM_NF-07-352.
-Ring1B (Cell Signalling #5694) "This antibody has been validated using SimpleChIP® Enzymatic Chromatin IP Kits." All other information can be found at https://www.cellsignal.com/products/primary-antibodies/ring1b-d22f2-rabbit-monoclonal-antibody/5694.
-Acetyl-Tubulin (Sigma-Aldrich #T7451) "Monoclonal Anti-Acetylated Tubulin antibody produced in mouse has been used in: quantitative dot blot, immunofluorescence, Western blot, immunocytochemistry, enzyme-linked immunosorbent assay (ELISA), solid-phase radioimmunoassay (RIA), electron microscopy" All other information can be found at https://www.sigmaaldrich.com/FR/en/product/sigma/t7451.
-Lamin B1 (abcam #ab16048) "KO validated for confirmed specificity." All other information can be found at: https://www.abcam.com/en-us/products/primary-antibodies/lamin-b1-antibody-nuclear-envelope-marker-ab16048.

In addition to external validations, we manually verified the profiles obtained in this study by comparing them with previous profiles obtained in our lab as well as published profiles of each of these marks, that were previously published using ES cells. All of the results were qualitatively comparable, both on loci that are known to be actively expressed in ES cells, on known Polycomb target genes such as Hox clusters, Wnt genes or Pax loci, as well as on known CTCF target sites or TAD boundary sites.

## Eukaryotic cell lines

Policy information about cell lines and Sex and Gender in Research

| Cell line source(s) | E14Tg2a.4 - purchased at MMRRC, donor is BayGenomics, BayGenomics Consortium, strain-129P2/OlaHsd.<br>CTCF-AID-eGFP E14Tga2 (ATCC, CRL-1821) published in Nora et al. (2017) DOI: 10.1016/j.molcel.2019.08.015 - gift from Ricardo Saldaña-Meyer.<br>PCGF4-/- PCGF2fl/fl Rosa26::CreERT2 ESCs published in Fursova et al. (2019) DOI: 10.1016/j.molcel.2019.03.024 - gift from Rob Klose. |
|---|---|
| Authentication | CTCF-AID-eGFP expression was confirmed by anti-GFP immunofluorescence. PCGF4 deletion and PCGF2 excision in response to tamoxifen (OHT) were verified by genotyping PCR. |
| Mycoplasma contamination | Cells were negative to mycoplasma spp. by PCR analysis. |
| Commonly misidentified lines (See ICLAC register) | This study does not use any commonly misidentified lines. |

## Plants

| Seed stocks | N/A |
|---|---|
| Novel plant genotypes | N/A |
| Authentication | N/A |

## ChIP-seq

### Data deposition

☒ Confirm that both raw and final processed data have been deposited in a public database such as GEO.

☒ Confirm that you have deposited or provided access to graph files (e.g. BED files) for the called peaks.

| Data access links<br>*May remain private before publication.* | Analysed and raw data are available under the GEO SuperSeries number GSE281151. ChIP-seq data files can be found under the GEO Series GSE280487. |
|---|---|
| Files in database submission | Bigwig files:<br>GSE280487_CTCF_mDMSO_dS.bam.bw<br>GSE280487_CTCF_mTSA.bam.bw |

GSE280487_H3K9me3_m24hREC_dS.bam.bw
GSE280487_H3K9me3_mDMSO_dS.bam.bw
GSE280487_H3K9me3_mTSA_dS.bam.bw
GSE280487_K119Ub_mDMSO_scaled.bw
GSE280487_K119Ub_mREC_scaled.bw
GSE280487_K119Ub_mTSA_scaled.bw
GSE280487_K119Ub_mreREC_scaled.bw
GSE280487_K119Ub_mreTSA_scaled.bw
GSE280487_K27ac_mDMSO_scaled.bw
GSE280487_K27ac_mREC_scaled.bw
GSE280487_K27ac_mTSA_scaled.bw
GSE280487_K27ac_mreREC_scaled.bw
GSE280487_K27ac_mreTSA_scaled.bw
GSE280487_K27me3_mDMSO_scaled.bw
GSE280487_K27me3_mREC_scaled.bw
GSE280487_K27me3_mTSA_scaled.bw
GSE280487_K27me3_mreREC_scaled.bw
GSE280487_K27me3_mreTSA_scaled.bw
GSE280487_K4me1_mDMSO_scaled.bw
GSE280487_K4me1_mREC_scaled.bw
GSE280487_K4me1_mTSA_scaled.bw
GSE280487_K4me1_mreREC_scaled.bw
GSE280487_K4me1_mreTSA_scaled.bw
GSE280487_K4me3_mDMSO_scaled.bw
GSE280487_K4me3_mreREC_scaled.bw
GSE280487_K4me3_mreTSA_scaled.bw
GSE280487_K9ac_mDMSO_scaled.bw
GSE280487_K9ac_mREC_scaled.bw
GSE280487_K9ac_mTSA_scaled.bw
GSE280487_PCGF2_H2AK119Ub_mOHT_DMSO_scaled.bw
GSE280487_PCGF2_H2AK119Ub_mOHT_REC_scaled.bw
GSE280487_PCGF2_H2AK119Ub_mOHT_TSA_scaled.bw
GSE280487_PCGF2_H2AK119Ub_mUT_DMSO_scaled.bw
GSE280487_PCGF2_H3K27me3_mOHT_DMSO_scaled.bw
GSE280487_PCGF2_H3K27me3_mOHT_REC_scaled.bw
GSE280487_PCGF2_H3K27me3_mOHT_TSA_scaled.bw
GSE280487_PCGF2_H3K27me3_mUT_DMSO_scaled.bw
GSE280487_Ring1B_mDMSO_scaled.bw
GSE280487_Ring1B_mREC_scaled.bw
GSE280487_Ring1B_mTSA_scaled.bw
Peak files:
GSE280487_CTCF_DMSO_intersect.narrowPeak.gz
GSE280487_CTCF_TSA_intersect.narrowPeak.gz
GSE280487_H2AK119Ub_24hREC_intersect.broadPeak.gz
GSE280487_H2AK119Ub_DMSO_intersect.broadPeak.gz
GSE280487_H2AK119Ub_TSA_intersect.broadPeak.gz
GSE280487_H3K27ac_24hREC_intersect.broadPeak.gz
GSE280487_H3K27ac_DMSO_intersect.broadPeak.gz
GSE280487_H3K27ac_TSA_intersect.broadPeak.gz
GSE280487_H3K27me3_24hREC_intersect.broadPeak.gz
GSE280487_H3K27me3_DMSO_intersect.broadPeak.gz
GSE280487_H3K27me3_TSA_intersect.broadPeak.gz
GSE280487_H3K4me1_24hREC_intersect.broadPeak.gz
GSE280487_H3K4me1_TSA_intersect.broadPeak.gz
GSE280487_H3K4me3_24hREC_intersect.broadPeak.gz
GSE280487_H3K4me3_DMSO_intersect.broadPeak.gz
GSE280487_H3K4me3_TSA_intersect.broadPeak.gz
GSE280487_H3K9me3_24hREC_intersect.broadPeak.gz
GSE280487_H3K9me3_DMSO_intersect.broadPeak.gz
GSE280487_H3K9me3_TSA_intersect.broadPeak.gz
GSE280487_K119Ub_reREC_intersect.broadPeak.gz
GSE280487_K119Ub_reTSA_intersect.broadPeak.gz
GSE280487_K119ub_OHT_DMSO_intersect.broadPeak.gz
GSE280487_K119ub_OHT_REC_intersect.broadPeak.gz
GSE280487_K119ub_OHT_TSA_intersect.broadPeak.gz
GSE280487_K119ub_UT_DMSO_intersect.broadPeak.gz
GSE280487_K27ac_reREC_intersect.broadPeak.gz
GSE280487_K27ac_reTSA_intersect.broadPeak.gz
GSE280487_K27me3_OHT_DMSO_intersect.broadPeak.gz
GSE280487_K27me3_OHT_REC_intersect.broadPeak.gz
GSE280487_K27me3_OHT_TSA_intersect.broadPeak.gz
GSE280487_K27me3_UT_DMSO_intersect.broadPeak.gz
GSE280487_K27me3_mreTSA_scaled.bw
GSE280487_K27me3_reREC_intersect.broadPeak.gz
GSE280487_K27me3_reTSA_intersect.broadPeak.gz
GSE280487_K4me1_mreTSA_scaled.bw

GSE280487_K4me1_reREC_intersect.broadPeak.gz
GSE280487_K4me3_mreTSA_scaled.bw
GSE280487_K4me3_reREC_intersect.broadPeak.gz
GSE280487_K4me3_reTSA_intersect.broadPeak.gz
GSE280487_K9ac_DMSO_intersect.broadPeak.gz
GSE280487_K9ac_REC_intersect.broadPeak.gz
GSE280487_RAW.tar
GSE280487_Ring1B_DMSO_intersect.broadPeak.gz
GSE280487_Ring1B_REC_intersect.broadPeak.gz
Raw data files:
H3K4me1_DMSO_rep1_1.fq.gz
H3K4me1_DMSO_rep2_1.fq.gz
H3K4me1_TSA_rep1_1.fq.gz
H3K4me1_TSA_rep2_1.fq.gz
H3K4me1_24hREC_rep1_1.fq.gz
H3K4me1_24hREC_rep2_1.fq.gz
H3K4me3_DMSO_rep1_1.fq.gz
H3K4me3_DMSO_rep2_1.fq.gz
H3K4me3_TSA_rep1_1.fq.gz
H3K4me3_TSA_rep2_1.fq.gz
H3K4me3_24hREC_rep1_1.fq.gz
H3K4me3_24hREC_rep2_1.fq.gz
H3K9me3_DMSO_rep1_1.fq.gz
H3K9me3_DMSO_rep2_1.fq.gz
H3K9me3_TSA_rep1_1.fq.gz
H3K9me3_TSA_rep2_1.fq.gz
H3K9me3_24hREC_rep1_1.fq.gz
H3K9me3_24hREC_rep2_1.fq.gz
H3K27me3_DMSO_rep1_1.fq.gz
H3K27me3_DMSO_rep2_1.fq.gz
H3K27me3_TSA_rep1_1.fq.gz
H3K27me3_TSA_rep2_1.fq.gz
H3K27me3_24hREC_rep1_1.fq.gz
H3K27me3_24hREC_rep2_1.fq.gz
H3K27ac_DMSO_rep1_1.fq.gz
H3K27ac_DMSO_rep2_1.fq.gz
H3K27ac_TSA_rep1_1.fq.gz
H3K27ac_TSA_rep2_1.fq.gz
H3K27ac_24hREC_rep1_1.fq.gz
H3K27ac_24hREC_rep2_1.fq.gz
H2AK119Ub_DMSO_rep1_1.fq.gz
H2AK119Ub_DMSO_rep2_1.fq.gz
H2AK119Ub_TSA_rep1_1.fq.gz
H2AK119Ub_TSA_rep2_1.fq.gz
H2AK119Ub_24hREC_rep1_1.fq.gz
H2AK119Ub_24hREC_rep2_1.fq.gz
input_DMSO_rep1_1.fq.gz
input_DMSO_rep2_1.fq.gz
input_TSA_1.fq.gz
input_24hREC_1.fq.gz
H3K4me1_DMSO_rep1_2.fq.gz
H3K4me1_DMSO_rep2_2.fq.gz
H3K4me1_TSA_rep1_2.fq.gz
H3K4me1_TSA_rep2_2.fq.gz
H3K4me1_24hREC_rep1_2.fq.gz
H3K4me1_24hREC_rep2_2.fq.gz
H3K4me3_DMSO_rep1_2.fq.gz
H3K4me3_DMSO_rep2_2.fq.gz
H3K4me3_TSA_rep1_2.fq.gz
H3K4me3_TSA_rep2_2.fq.gz
H3K4me3_24hREC_rep1_2.fq.gz
H3K4me3_24hREC_rep2_2.fq.gz
H3K9me3_DMSO_rep1_2.fq.gz
H3K9me3_DMSO_rep2_2.fq.gz
H3K9me3_TSA_rep1_2.fq.gz
H3K9me3_TSA_rep2_2.fq.gz
H3K9me3_24hREC_rep1_2.fq.gz
H3K9me3_24hREC_rep2_2.fq.gz
H3K27me3_DMSO_rep1_2.fq.gz
H3K27me3_DMSO_rep2_2.fq.gz
H3K27me3_TSA_rep1_2.fq.gz
H3K27me3_TSA_rep2_2.fq.gz
H3K27me3_24hREC_rep1_2.fq.gz
H3K27me3_24hREC_rep2_2.fq.gz
H3K27ac_DMSO_rep1_2.fq.gz
H3K27ac_DMSO_rep2_2.fq.gz

```
H3K27ac_TSA_rep1_2.fq.gz
H3K27ac_TSA_rep2_2.fq.gz
H3K27ac_24hREC_rep1_2.fq.gz
H3K27ac_24hREC_rep2_2.fq.gz
H2AK119Ub_DMSO_rep1_2.fq.gz
H2AK119Ub_DMSO_rep2_2.fq.gz
H2AK119Ub_TSA_rep1_2.fq.gz
H2AK119Ub_TSA_rep2_2.fq.gz
H2AK119Ub_24hREC_rep1_2.fq.gz
H2AK119Ub_24hREC_rep2_2.fq.gz
input_DMSO_rep1_2.fq.gz
input_DMSO_rep2_2.fq.gz
input_TSA_2.fq.gz
input_24hREC_2.fq.gz

H3K27ac_reTSA_rep1_1.fq.gz
H3K27ac_reTSA_rep2_1.fq.gz
H3K4me1_reTSA_rep1_1.fq.gz
H3K4me1_reTSA_rep2_1.fq.gz
H3K4me3_reTSA_rep1_1.fq.gz
H3K4me3_reTSA_rep2_1.fq.gz
H3K27me3_reTSA_rep1_1.fq.gz
H3K2ume3_reTSA_rep2_1.fq.gz
H2AK119Ub_reTSA_rep1_1.fq.gz
H2AK119Ub_reTSA_rep2_1.fq.gz
input_reTSA_rep1_1.fq.gz
input_reTSA_rep2_1.fq.gz
H3K27ac_reREC_rep1_1.fq.gz
H3K27ac_reREC_rep2_1.fq.gz
H3K4me1_reREC_rep1_1.fq.gz
H3K4me1_reREC_rep2_1.fq.gz
H3K4me3_reREC_rep1_1.fq.gz
H3K4me3_reREC_rep2_1.fq.gz
H3K27me3_reREC_rep1_1.fq.gz
H3K2ume3_reREC_rep2_1.fq.gz
H2AK119Ub_reREC_rep1_1.fq.gz
H2AK119Ub_reREC_rep2_1.fq.gz
input_reREC_rep1_1.fq.gz
input_reREC_rep2_1.fq.gz
H3K9ac_DMSO_rep3_1.fq.gz
H3K9ac_DMSO_rep4_1.fq.gz
Ring1B_DMSO_rep3_1.fq.gz
Ring1B_DMSO_rep4_1.fq.gz
H3K27ac_DMSO_rep3_1.fq.gz
H3K4me1_DMSO_rep4_1.fq.gz
input_DMSO_rep3_1.fq.gz
input_DMSO_rep4_1.fq.gz
H3K9ac_TSA_rep3_1.fq.gz
H3K9ac_TSA_rep4_1.fq.gz
Ring1B_TSA_rep3_1.fq.gz
Ring1B_TSA_rep4_1.fq.gz
H3K27ac_TSA_rep3_1.fq.gz
input_TSA_rep3_1.fq.gz
input_TSA_rep4_1.fq.gz
H3K9ac_REC_rep3_1.fq.gz
H3K9ac_REC_rep4_1.fq.gz
Ring1B_REC_rep3_1.fq.gz
Ring1B_REC_rep4_1.fq.gz
H3K27ac_REC_rep3_1.fq.gz
input_REC_rep3_1.fq.gz
input_REC_rep4_1.fq.gz
TSA_YY1_rep1_1.fq.gz
TSA_YY1_L1_rep2_1.fq.gz
TSA_input_rep1_1.fq.gz
H3K27me3_PCGF2_UT_DMSO_rep1_1.fq.gz
H3K27me3_PCGF2_UT_DMSO_rep2_1.fq.gz
H2AK119Ub_PCGF2_UT_DMSO_rep1_1.fq.gz
H2AK119Ub_PCGF2_UT_DMSO_rep2_1.fq.gz
input_PCGF2_UT_DMSO_1.fq.gz
H3K27me3_PCGF2_OHT_DMSO_rep1_1.fq.gz
H3K27me3_PCGF2_OHT_DMSO_rep2_1.fq.gz
H2AK119Ub_PCGF2_OHT_DMSO_rep1_1.fq.gz
H2AK119Ub_PCGF2_OHT_DMSO_rep2_1.fq.gz
input_PCGF2_OHT_DMSO_1.fq.gz
H3K27me3_PCGF2_OHT_TSA_rep1_1.fq.gz
H3K27me3_PCGF2_OHT_TSA_rep2_1.fq.gz
```

```
H2AK119Ub_PCGF2_OHT_TSA_rep1_1.fq.gz
H2AK119Ub_PCGF2_OHT_TSA_rep2_1.fq.gz
input_PCGF2_OHT_TSA_1.fq.gz
H3K27me3_PCGF2_OHT_REC_rep1_1.fq.gz
H3K27me3_PCGF2_OHT_REC_rep2_1.fq.gz
H2AK119Ub_PCGF2_OHT_REC_rep1_1.fq.gz
H2AK119Ub_PCGF2_OHT_REC_rep2_1.fq.gz
input_PCGF2_OHT_REC_1.fq.gz
H3K27ac_reTSA_rep1_2.fq.gz
H3K27ac_reTSA_rep2_2.fq.gz
H3K4me1_reTSA_rep1_2.fq.gz
H3K4me1_reTSA_rep2_2.fq.gz
H3K4me3_reTSA_rep1_2.fq.gz
H3K4me3_reTSA_rep2_2.fq.gz
H3K27me3_reTSA_rep1_2.fq.gz
H3K2ume3_reTSA_rep2_2.fq.gz
H2AK119Ub_reTSA_rep1_2.fq.gz
H2AK119Ub_reTSA_rep2_2.fq.gz
input_reTSA_rep1_2.fq.gz
input_reTSA_rep2_2.fq.gz
H3K27ac_reREC_rep1_2.fq.gz
H3K27ac_reREC_rep2_2.fq.gz
H3K4me1_reREC_rep1_2.fq.gz
H3K4me1_reREC_rep2_2.fq.gz
H3K4me3_reREC_rep1_2.fq.gz
H3K4me3_reREC_rep2_2.fq.gz
H3K27me3_reREC_rep1_2.fq.gz
H3K2ume3_reREC_rep2_2.fq.gz
H2AK119Ub_reREC_rep1_2.fq.gz
H2AK119Ub_reREC_rep2_2.fq.gz
input_reREC_rep1_2.fq.gz
input_reREC_rep2_2.fq.gz
H3K9ac_DMSO_rep3_2.fq.gz
H3K9ac_DMSO_rep4_2.fq.gz
Ring1B_DMSO_rep3_2.fq.gz
Ring1B_DMSO_rep4_2.fq.gz
H3K27ac_DMSO_rep3_2.fq.gz
H3K4me1_DMSO_rep4_2.fq.gz
input_DMSO_rep3_2.fq.gz
input_DMSO_rep4_2.fq.gz
H3K9ac_TSA_rep3_2.fq.gz
H3K9ac_TSA_rep4_2.fq.gz
Ring1B_TSA_rep3_2.fq.gz
Ring1B_TSA_rep4_2.fq.gz
H3K27ac_TSA_rep3_2.fq.gz
input_TSA_rep3_2.fq.gz
input_TSA_rep4_2.fq.gz
H3K9ac_REC_rep3_2.fq.gz
H3K9ac_REC_rep4_2.fq.gz
Ring1B_REC_rep3_2.fq.gz
Ring1B_REC_rep4_2.fq.gz
H3K27ac_REC_rep3_2.fq.gz
input_REC_rep3_2.fq.gz
input_REC_rep4_2.fq.gz
TSA_YY1_rep1_2.fq.gz
TSA_YY1_L1_rep2_2.fq.gz
TSA_input_rep1_2.fq.gz
H3K27me3_PCGF2_UT_DMSO_rep1_2.fq.gz
H3K27me3_PCGF2_UT_DMSO_rep2_2.fq.gz
H2AK119Ub_PCGF2_UT_DMSO_rep1_2.fq.gz
H2AK119Ub_PCGF2_UT_DMSO_rep2_2.fq.gz
input_PCGF2_UT_DMSO_2.fq.gz
H3K27me3_PCGF2_OHT_DMSO_rep1_2.fq.gz
H3K27me3_PCGF2_OHT_DMSO_rep2_2.fq.gz
H2AK119Ub_PCGF2_OHT_DMSO_rep1_2.fq.gz
H2AK119Ub_PCGF2_OHT_DMSO_rep2_2.fq.gz
input_PCGF2_OHT_DMSO_2.fq.gz
H3K27me3_PCGF2_OHT_TSA_rep1_2.fq.gz
H3K27me3_PCGF2_OHT_TSA_rep2_2.fq.gz
H2AK119Ub_PCGF2_OHT_TSA_rep1_2.fq.gz
H2AK119Ub_PCGF2_OHT_TSA_rep2_2.fq.gz
input_PCGF2_OHT_TSA_2.fq.gz
H3K27me3_PCGF2_OHT_REC_rep1_2.fq.gz
H3K27me3_PCGF2_OHT_REC_rep2_2.fq.gz
H2AK119Ub_PCGF2_OHT_REC_rep1_2.fq.gz
```

H2AK119Ub_PCGF2_OHT_REC_rep2_2.fq.gz
input_PCGF2_OHT_REC_2.fq.gz

**Genome browser session**
(e.g. UCSC)

IGV

## Methodology

**Replicates**

All ChIP-seq experiments were done in biological duplicates, H3K27ac ChIP-seq was done in biological triplicates.

**Sequencing depth**

Epitope Condition Replicate Uniquely mapped reads
H3K4me1 TSA 1 54818314
H3K4me3 TSA 1 68009818
H3K9me3 TSA 1 33008412
H3K27me3 TSA 1 57480252
H3K27ac TSA 1 78013842
H2AK119ub TSA 1 61046204
CTCF TSA 1 54263736
YY1 TSA 1 26802614
YY1 DMSO 1 19484622
input TSA 1 19133166
H3K4me1 TSA 2 61371056
H3K4me3 TSA 2 65808876
H3K9me3 TSA 2 48287118
H3K27me3 TSA 2 55112334
H3K27ac TSA 2 65446160
H2AK119ub TSA 2 48330640
Rad21 TSA 2 54363204
CTCF TSA 2 60778948
YY1 TSA 2 31665080
YY1 DMSO 2 35199046
input DMSO 1 16692324
H3K4me1 DMSO 1 45412782
H3K4me3 DMSO 1 43890218
H3K9me3 DMSO 1 28781034
H3K27me3 DMSO 1 46382548
H3K27ac DMSO 1 44150428
H2AK119ub DMSO 1 37625772
Rad21 DMSO 1 47885902
CTCF DMSO 1 44213344
H3K4me1 24hREC 1 71991732
H3K4me1 24hREC 1 73380982
H3K9me3 24hREC 1 22442310
H3K27me3 24hREC 1 35979550
H3K27ac 24hREC 1 85291528
H2AK119ub 24hREC 1 28678806
H3K4me1 24hREC 2 65286048
H3K4me3 24hREC 2 64448520
H3K9me3 24hREC 2 34203612
H3K27me3 24hREC 2 28833974
H3K27ac 24hREC 2 75452030
H2AK119ub 24hREC 2 53702010
H3K4me1 DMSO 2 60862026
H3K4me3 DMSO 2 66707678
H3K9me3 DMSO 2 24801870
H3K27me3 DMSO 2 35094396
H3K27ac DMSO 2 72168012
H2AK119ub DMSO 2 22036164
Rad21 DMSO 2 69194288
CTCF DMSO 2 64134764
input 24hREC 1 25297596
input DMSO 2 21755478
H3K27ac reTSA 1 68176590
H3K4me1 reTSA 1 46840426
H3K4me3 reTSA 1 55506436
H3K27me3 reTSA 1 69242556
H2AK119Ub reTSA 1 57296340
H3K27ac reTSA 2 71278248
H3K4me1 reTSA 2 54732654
H3K4me3 reTSA 2 61073612
H3K27me3 reTSA 2 60488472
H2AK119Ub reTSA 2 49177130
input reTSA 1 25688120
input reTSA 2 26598012
H3K27ac reREC 1 54721142
H3K4me1 reREC 1 49971722

H3K4me3 reREC 1 59984504
H3K27me3 reREC 1 59279822
H2AK119Ub reREC 1 60537406
H3K27ac reREC 2 61175514
H3K4me1 reREC 2 51603532
H3K4me3 reREC 2 58975348
H3K27me3 reREC 2 57988216
H2AK119Ub reREC 2 69468460
input reREC 1 20212252
input reREC 2 23208486
H3K27ac DMSO 3 53906164
H3K9ac DMSO 3 38367614
Ring1B DMSO 3 49224398
H3K9ac DMSO 4 38150380
Ring1B DMSO 4 50305330
H3K4me1 DMSO 4 46965044
H3K27ac TSA 3 66816874
H3K9ac TSA 3 58712350
Ring1B TSA 3 58902498
H3K9ac TSA 4 54175406
Ring1B TSA 4 59741032
H3K27ac REC 3 49171736
H3K9ac REC 3 52634916
Ring1B REC 3 57112740
H3K9ac REC 4 30452666
Ring1B REC 4 50993230
input DMSO 3 34464872
input DMSO 4 33952542
input TSA 3 32588284
input TSA 4 35452042
input REC 3 33429800
input REC 4 21573896
H3K27me3 PCGF2 -OHT +DMSO 1 46259940
H3K27me3 PCGF2 -OHT +DMSO 2 50642716
H2AK119Ub PCGF2 -OHT +DMSO 1 44492246
H2AK119Ub PCGF2 -OHT +DMSO 2 50212746
input PCGF2 -OHT +DMSO 1 33166120
H3K27me3 PCGF2 +OHT +DMSO 1 38362120
H3K27me3 PCGF2 +OHT +DMSO 2 47164752
H2AK119Ub PCGF2 +OHT +DMSO 1 51489588
H2AK119Ub PCGF2 +OHT +DMSO 2 44400782
input PCGF2 +OHT +DMSO 1 36482660
H3K27me3 PCGF2 +OHT +TSA 1 56949378
H3K27me3 PCGF2 +OHT +TSA 2 53889352
H2AK119Ub PCGF2 +OHT +TSA 1 47727028
H2AK119Ub PCGF2 +OHT +TSA 2 46566516
input PCGF2 +OHT +TSA 1 22281632
H3K27me3 PCGF2 +OHT +REC 1 54817788
H3K27me3 PCGF2 +OHT +REC 2 57616746
H2AK119Ub PCGF2 +OHT +REC 1 54138000
H2AK119Ub PCGF2 +OHT +REC 2 53981734
input PCGF2 +OHT +REC 1 36046280

| | |
|---|---|
| Antibodies | The following antibodies were used:<br>- H3K4me1 (ActiveMotif #39297)<br>- H3K4me3 (Millipore #04-745)<br>- H3K9me3 (abcam #8898)<br>- H3K27me3 (ActiveMotif #39155)<br>- H3K27Ac (ActiveMotif #39133)<br>- H2AK119Ub (Cell Signalling #8240S)<br>- CTCF (Active Motif #61311)<br>- YY1 (abcam #109237)<br>- H3K9Ac (Millipore #07-352)<br>- Ring1B (Cell Signalling #5694) |
| Peak calling parameters | We called peaks with MACS3 (https://hbctraining.github.io/Intro-to-ChIPseq-flipped/lessons/06_peak_calling_macs.html) with default setting; for histone marks with the --broad option specified. More detailed in the Methods. |
| Data quality | Assessed by reproducibility of published data. |
| Software | As described in the Methods ChIP-seq samples were mapped using bowtie2 v.2.3.5.1 (https://bowtie-bio.sourceforge.net/bowtie2/index.shtml) with command "bowtie2 -p 12 --no-mixed --no-discordant" 65. Then, we used samtools v.1.9 (https://www.htslib.org/doc/samtools-view.html) to filter out low-quality reads (command "samtools view -b -q 30"). Finally, we used Sambamba v1.0 |

(https://github.com/biod/sambamba) to sort the bam files (command "sambamba sort"), deduplicate, and index them ("sambamba markdup --remove-duplicates") with default parameters.

# Flow Cytometry

## Plots

Confirm that:

☒ The axis labels state the marker and fluorochrome used (e.g. CD4-FITC).

☒ The axis scales are clearly visible. Include numbers along axes only for bottom left plot of group (a 'group' is an analysis of identical markers).

☒ All plots are contour plots with outliers or pseudocolor plots.

☒ A numerical value for number of cells or percentage (with statistics) is provided.

## Methodology

| | |
|---|---|
| Sample preparation | 1-3x106 mESCs were dissociated with TrypLE, pelleted, and resuspended in PBS. For cell cycle analysis, dissociated mESCs were washed once in PBS and pelleted and fixed in cold 70% ethanol for 30 min at 4°C. Cells were stained with the Propidium Iodide Flow Cytometry Kit (Abcam #ab139418) according to manufacturer's instruction. Flow cytometry was performed on a CytoFlex instrument using CytExpert (v2.4), and analysis was performed using the FlowJo (v10.10) software. For cell proliferation tracing, dissociated mESCs were stained with 1 μM CellTrace Violet staining solution (Invitrogen #C34571) according to manufacturer's instructions, and were plated on gelatine-coated cell culture dishes. After 24 hours, TSA treatment and washes were performed as described before and cells were harvested following a further 24-hour incubation period. Collected cells were fixed in 4% PFA for 10 minutes at room temperature, washed with PBS and preserved at 4°C until further use. Flow cytometry was performed on a Novocyte Quanteon instrument, and analysis was performed using the NovoExpress (v1.6.3) software. |
| Instrument | CytoFlex (Beckman) or Novocyte Quanteon (Agilent) |
| Software | CytExpert (v2.4)<br>NovoExpress (v1.6.3)<br>FlowJo (v10.10) |
| Cell population abundance | 20000-30000 cells were assayed for each sample. |
| Gating strategy | Gating was based on the pattern of FSC-A/SSC-A. Singlets were gated based on the pattern of FSC-H/FSC-A. |

☒ Tick this box to confirm that a figure exemplifying the gating strategy is provided in the Supplementary Information.

