## [Peer Review File · Nature Genetics]

Transient histone deacetylase inhibition induces cellular memory of gene expression and three-dimensional genome folding

Corresponding Author: Dr Giacomo Cavalli

Version 0:

Decision Letter:

7th Jan 2025

Dear Dr Cavalli,

apologize for the delay.

Your Article, "Transient histone deacetylase inhibition induces cellular memory of gene expression and three-dimensional genome folding" has now been seen by 2 referees. You will see from their comments copied below that while they find your work of considerable potential interest, they have raised quite substantial concerns that must be addressed. In light of these comments, we cannot accept the manuscript for publication, but would be very interested in considering a revised version that addresses these serious concerns.

We hope you will find the referees' comments useful as you decide how to proceed. If you wish to submit a substantially revised manuscript, please bear in mind that we will be reluctant to approach the referees again in the absence of major revisions.

To guide the scope of the revisions, the editors discuss the referee reports in detail within the team, including with the chief editor, with a view to identifying key priorities that should be addressed in revision and sometimes overruling referee requests that are deemed beyond the scope of the current study. In this case, we ask you to address reviewers' comments in full. Please do not hesitate to get in touch if you would like to discuss these issues further.

If you choose to revise your manuscript taking into account all reviewer and editor comments, please highlight all changes in the manuscript text file. At this stage we will need you to upload a copy of the manuscript in MS Word .docx or similar editable format.

*2) If you have not done so already please begin to revise your manuscript so that it conforms to our Article format instructions, available [here](http://www.nature.com/ng/authors/article_types/index.html). Refer also to any guidelines provided in this letter.

*3) Include a revised version of any required Reporting Summary: <https://www.nature.com/documents/nr-reporting-summary.pdf>

Please be aware of our [guidelines](https://www.nature.com/nature-research/editorial-policies/image-integrity) on digital image standards.

EXTENDED DATA FIGURES

Link Redacted

If you wish to submit a suitably revised manuscript we would hope to receive it within 6 months. If you cannot send it within this time, please let us know. We will be happy to consider your revision so long as nothing similar has been accepted for publication at Nature Genetics or published elsewhere. Should your manuscript be substantially delayed without notifying us in advance and your article is eventually published, the received date would be that of the revised, not the original, version.

Nature Genetics is committed to improving transparency in authorship. As part of our efforts in this direction, we are now requesting that all authors identified as 'corresponding author' on published papers create and link their Open Researcher and Contributor Identifier (ORCID) with their account on the Manuscript Tracking System (MTS), prior to acceptance. ORCID helps the scientific community achieve unambiguous attribution of all scholarly contributions. You can create and link your ORCID from the home page of the MTS by clicking on 'Modify my Springer Nature account'. For more information please visit please visit www.springernature.com/orcid.

Thank you for the opportunity to review your work.

Sincerely,
Chiara

Chiara Anania, PhD
Associate Editor
Nature Genetics
<https://orcid.org/0000-0003-1549-4157>

Referee expertise:

Referee #1: chromatin, epigenetics

Referee #2: epigenetic inheritance

Referee #3:

Reviewers' Comments:

Reviewer #1 (Remarks to the Author):

In this manuscript, Paldi and colleagues investigate important and still unresolved questions around epigenetic/cellular memory. The overarching approach is to apply a pervasive but transient stimulus in mouse embryonic stem cells and measure the transcriptional, epigenomic and 3D-chromatin state responses at multiple time points during and after perturbation. More specifically, the authors apply the deacetylase inhibitor TSA for 4hrs at a dosage that leads to major changes preferentially in histone acetylation levels. They then release cells and assay across omics modalities. The study can be considered in two broad parts. Firstly, is a very careful and comprehensive characterisation of how perturbed (elevated) global acetylation impacts molecular layers of regulation in ESC, using ultra-deep Micro-C integrated with RNA-seq and ChIP-seq. In this survey, the authors note a number of interesting responses, including preferential interactions between BB compartments in cis (and associated model), widespread histone modifications changes (including some unexpected) and a profoundly altered transcriptome (>4,000). Second, the authors test the ability of these 4hr TSA-treated cells to recover their transcriptional, epigenomic, and 3D-structural state after 24 hours from TSA washout. Here, they observe that transcriptional and epigenomic features are mostly reversed to the pre-TSA state (78 and 86 down- and up-

regulated genes persist, respectively), while genome conformation retained some features of the TSA-induced state, in particular at the level of cis-interactions, and at the sub-megabase scale for specific instances of gene loops. This leads to the central conclusion that chromatin looping carries a memory of prior perturbations. Beyond this, the authors also ask the interesting question of how a second round of stimulus/release affects responses, finding that a subset of genes exhibit a seemingly enhanced second response – further implying an epigenetic memory.

The study is communicated in a clear and well-structured manner. Moreover, it addresses a question that is of wide general interest – both from a genome regulation perspective and from the broader question of transmitting ‘epigenetic’ memory. In my view the study could make important contributions and should be considered by Nature Genetics in principle. However, there are a number of interpretations and confounders that make strong conclusions very difficult in present form, and at least two major areas of design/conceptual concern. Firstly, based on the data presented the perturbation affects many molecular layers and is still persisting at 24hrs washout, meaning that rather than measuring a memory of looping, the observed changes could reflect an intermittent timepoint in a dynamic – but incomplete – recovery of many factors. Second and related, the argument that mechanistically it is the “genome architecture [that] carries a memory of its TSA-induced effects” is not supported by direct (rather than correlative) evidence and other possibilities are not ruled out. Given this is central for the novel insight of the study, this could be strengthened. Below I go into further detail on these points and highlight some other concerns as well as the strengths of the study.

Key comments & concerns (in order of occurrence in manuscript)

1.) Specificity of the perturbation - It is evident from literature and the author data that TSA is not just a nuclear histone perturbation – despite careful titration of dosing - but likely pleiotropic. For example, in Fig S1A an entirely new band appears upon (re)TSA exposure in the cytoplasmic fraction. More generally, the overall intensity of the cytoplasmic acetylation fraction also increases upon TSA, supporting non-histone changes in protein acetylation. Finally, there appears to be a subtly different cell cycle profile upon TSA (in Fig S1C – increase in transition population). Taken together, it appears the TSA treatment is not just a chromatin perturbation, which has ripple effects for interpreting the remaining data. The statement that “the bulk of the acetylome, cell cycle progression and cell viability remain unchanged” might be too strong.

2. Epigenetic landscape analysis. Mapping the chromatin modification changes in response to TSA both establishes an interesting dataset and provides insights into response relationships. It has clearly been performed carefully and in a calibrated manner so, like the micro-C, it is a powerful resource. However, some of the core observations are strange and hint at confounders – or should be further followed up.

(i) Almost all chromatin marks increase upon TSA, including the repressive H2Ab119ub (Fig3A-B) at all target genes. This could be interesting but could also be an artefact related to increased antibody access to histone epitopes upon global acetylation. In other words does ChIP just pull down more of everything upon TSA treatment due to technical reasons. If not how do you explain almost all marks going up – particularly repressive ones.

(ii) Related to above, almost all chromatin modifications exhibit the same the same degree of change at both UP and DOWN-regulated genesets (Fig 3b-c). That is, the intensity of both activating and repressing marks increases irrespective of the direction of transcription changes. This implies they are not functionally linked and could further hint at artefacts in performing ChIP with global acetylation increases, such as universal accessibility for antibodies. There only appears to be one exception, which is H3K4me3 – this increases specifically in upregulated genes but does NOT increase in downregulated genes. This could imply H3K4me3 has a functional role in response to HDAC inhibitors, which is testable.

(iii) Further related to the increased modifications, the authors conclude “Altogether, we find that gene upregulation as well as downregulation occur with a gain of activating chromatin marks” but no mention is made of the increase in repressive marks at active genes. Instead it is concluded that “while gene upregulation is potentially associated with enhancer over-activation without gain in E-P contacts, downregulation is linked to repressive chromatin looping. Its not clear to me how these collective conclusions are made. There are changes in many epigenomic layers so how can the authors claim these are the relevant mechanisms.

(iv)) The role of looping in gene repression is possibly under-explored and potentially very interesting. Why is there increased looping at polycomb targets but no increased polycomb?

3.) Principle of the recovery – is the initial response to TSA really recovered at 24hrs?

(i) While there is clearly a significant reversion of gene expression and acetylation, it appears incomplete to me. For example, H3K27ac marks are clearly not fully recovered at 24hrs in Fig 4B. Authors should provide a FC change plot of DMSO vs REC to support the argument that “original H3K27ac levels and the rest of the acetylome appeared restored at the protein level. Given that Fig4B suggests increased K27ac (DMSO vs REC) – and that H3K4me1/ATAC also appears different between DMSO and REC – it is difficult to make this conclusion. Moreover, if the chromatin state has not recovered then not only are the conclusions that favour looping less clear-cut, but it calls into question whether it is ‘memory’ being measured or a stage of an ongoing dynamic recovery.

(ii) similarly to histone modifications, it appears that gene expression is largely recovered - but not completely (Fig 4d). When comparing DMSO with REC, many hundreds of genes with LFC>2 remain (not all are significant). Thus, there is still at least some significant levels of altered transcriptional networks at 24hrs and presumably the proteome is even slower in recovering, given the half-life of many proteins, and thus in an even more perturbed state. As a note, why is the volcano plot in Fig 4D scaled to 300 on the y-axis when the highest data point is 25. The effect is to compress the data unnecessarily, giving the impression of low significance by eye.

Together, the gene expression and chromatin states (and looping) in REC suggests that the reversion is incomplete. This makes drawing conclusions on memory challenging. To get around this issue, I believe it is important to measure responses at longer timepoints (i.e. 48hrs, 96hrs etc) and also think about more regular and intermediate timepoints to record the rates of change between modalities. At present it is difficult to (i) confirm which regulatory layers have or have not recovered relative to each other (i.e. which is providing the memory) and (ii) whether it is memory or a freeze-frame during a dynamic recovery.

4.) TF targets - beyond histone and gene expression (incomplete) recovery, there are also questions around protein acetylation. Specifically, the authors mention that TSA-induced HDAC inhibition can have effects beyond histone acetylation (231-244), including an increase of acetylation of the transcription factor YY1 and MYC. Both TFs are known to be crucial transcriptional regulators, targeting thousands of genes. It could thus be that a portion of the transcriptional effects measured upon TSA treatment and recovery are a consequence of differential TF function rather than epigenomic and chromatin conformation changes. The authors show in Fig.E4c that genes downregulated upon TSA treatment are enriched for YY1 and Myc binding sites. To clarify the importance of differential TF activity upon TSA treatments and recovery phases, it could be valuable (albeit optionally) to perform ChIP-seq for the two factors in all conditions measured in the MS (i.e. DMSO, TSA, recovery, 2nd TSA and 2nd recovery). It could in fact be the case that acetylation of these TFs has a slower recovery dynamic as compared to, for example, nucleosome acetylation. In this case, some of the transcriptional “memory” features that the authors attribute to chromatin 3D-structural changes could be instead caused by TF activity.

5.) Second recovery – if one accepts that there is not complete recovery after 24hrs then the second exposure could be additive on top of an ongoing perturbation rather than evoking a memorized response in genome architecture. This would be important to discuss or consider testing experimentally – for example checking if there is a second perturbation boost at 48hrs (or longer) after the 1st exposure.

6.) Gastruloids as a model - while I agree with the HDAC-pulse experimental design in a steady state cell type such as ESC, I think there are some fundamental issues with application to gastruloids. Specifically, a transient change in a dynamic model will change the trajectory of differentiation and thus affect proportions of cell lineages in a gastruloid. Therefore observing increased Sox2 expression after washout does not “indicate that a memory of the TSA pulse might be ingrained.”, but more likely that a bias on lineage allocation was made at the time of exposure. Testing memory and dynamic cell states at the same time makes it difficult to uncouple one thing from the other – or understand if it is an earlier-induced change in cell trajectory rather than an molecular memory.

6.) Cell type. Many studies have shown that ESC possess a unique chromatin state and response patterns. Therefore, whether the observations have relevance beyond in vitro pluripotent cells remains open. The authors attempt to address this with gastruloids, with some success in terms of similar genesets being responsive. However, given comments above, it would be useful to discuss further the caveats to using ESC and their relevance to in vivo or developed tissues. Of course, it would be ideal to test the principle of ‘memory’ in another steady state cell type but this is may be a big task – especially if further timepoints are added to the ESC model

7.) Perturbation controls – it is surprising that only a single HDAC inhibitor was deployed. In order to begin to rule out off-target or indirect effects additional orthogonal perturbations (or genetic approaches such as degrons) could be used. This would add considerable strength to the conclusions if the effects were comparable.

8.) Mechanisms - Ultimately, it is concluded that looping plays a central role in the memory process, which is largely based on associations. Given the pleiotropy and incomplete recovery it is necessary to understand if memory is really encoded in looping. It would be ideal to artificially stimulate looping to test its ability to confer memory, for example by using a dCas9 to target activation or repression to candidate sites or via tethering assays. Such targeted manipulations would provide much more confidence in the conclusions.

Minor

I particularly appreciated the employment of biophysical modelling. In my view, mathematical models in Biology are useful when they allow to understand complex phenomena in a simplified manner, and when they produce predictions that can be tested and confirmed. In your model, as mentioned in line 155, there is the prediction of a physical migration of A-domains toward the periphery of the nucleus. This can be tested by microscopy, using DNA or RNA-FISH (the latter for an intron of a transcribed gene), for loci known to be part of A-compartments. At present the model largely reinforces experimental observations rather than providing new testable insights

210: A possible typo; "at" least?

Related to 304-308: It is not clear to me what is the extent of the TSA perturbation memorised in gastruloids. In Fig.E5e there is a Volcano plot showing some dysregulated genes (please add n for up- and down-regulated genes). I don't see among them Sox2, despite the authors show by staining an extension of the Sox2-marked neuroectodermal region (Fig.4h). Why is Sox2 not shown as upregulated in Fig.E5e?

Reviewer #2 (Remarks to the Author):

NG-A67123

In this study the authors aim to establish a link between 3D genome organization -a key regulator of genome function- and cellular memory. Specifically, they investigate whether transient changes in histone acetylation can induce durable and heritable alterations in 3D genome organization and transcriptional program in mouse embryonic stem cells (mESCs). Using a combination of ChIP-seq, ATAC-seq, RNA-seq, and ultra-deep Micro-C, the authors demonstrate that an acute (transient) increase in histone acetylation can trigger changes in genome folding and gene expression that can be retained, even in the absence of maintaining histone acetylation state. They show that histone deacetylase (HDAC) inhibition by TSA leads to modifications in histone acetylation (H3K27ac) and methylation (H3K4me1/3, H3K27me3, H3K9me3) and alterations in both global and local genome 3D folding.

Upon TSA washout, the epigenetic state (and particularly histone modification H3K27ac) rapidly reverts, although 3D conformational changes and deregulated gene expression persist as cells divide.

This "memory" becomes more pronounced upon a second exposure to TSA.

Overall, this is an extensive and well-described study. It provides evidence of a TSA-induced cellular memory of gene transcription and 3D genome that persists after transient epigenetic change. Although potentially interesting, there are two major issues that need to be addressed/resolved .

1. Firstly, the mechanism by which cellular memory operates in response to TSA treatment is unclear and is currently not sufficiently addressed. For example, it is not yet clear whether the 3D structure itself plays a direct role in conveying epigenetic memory, or whether this is achieved by other factors targeted by TSA. While the enhanced Micro-C technique used in this study offers a powerful tool to investigate the role of 3D genome interactions in regulating gene expression and transcriptional memory, the results remain correlative rather than demonstrating a clear causality.
2. Secondly, to claim an effect on cellular memory (as is indicated in the manuscript title) the authors need clonal assessments of TSA impact, rather than studying the effects of TSA on heterogenous cell cultures. This would distinguish allow them to show that single and repeat TSA treatments change the epigenetic status of cells (at the level of individual ESCs), rather than simply reading out changes within a population.

Specific Points:

- First, the authors demonstrate that TSA, a broad HDAC inhibitor, induces a global increase in H3K27ac, primarily at cis-regulatory elements along gene bodies. TSA also has secondary effects on histone modifications, including a loss of repressive marks (PRC1/2-mediated histone modification) and a gain of active marks (such as H3K4me1), suggesting increased chromatin relaxation and transcriptional activity. It would be important to understand whether other primary changes in histone acetylation, such as H3K9/14ac or H4K5/16ac, could also occur under TSA treatment at global or at specific loci of deregulated genes. How do the authors rule out increased acetylation of non-histone proteins ?
- Figure 1b shows H3K27ac peak density on chromosome 18, and Figure 1d show ChIP-seq data for various histone modifications on chromosome 7. Are there any specific acetylation patterns unique to these two chromosomes compared to others? Additionally, Figure 1d depicts ATAC-seq data following TSA treatment in a highly acetylated region. Chromatin accessibility does not appear to change significantly or correlate with increased acetylation change, the authors should describe and interpret these findings.
- Extended Data Figure 1: The authors should include data for H3K4me1 downregulation and H2AK119ub downregulation, as these are currently missing.
- The authors develop ultra-deep Micro-C contact maps for both control and TSA-treated mESCs, generating genome-wide interaction maps with unprecedented resolution. TSA treatment leads to an increase in inter-chromosomal contacts and a decrease in A-A compartment interactions. These 3D changes, along with histone modification alterations, partially correlate with gene expression dysregulation. Transcriptomic changes are associated with increased H3K27ac and concomitant H3K4me1 gain in surrounding regions, with no apparent loss of repressive marks (Figure 3). However, no promoter contact changes were observed at upregulated transcription start sites (TSSs). Beyond this correlative evidence, could the authors provide an example or the list of upregulated genes that exhibit increased interactions (measured by Micro-C, precising which type), H3K27ac, and H3K4me1/3 modifications. Are these genes involved in ESC commitment?

- 24 hours after TSA washout, H3K27ac and H3K4me1 levels return rapidly to normal level, which could be expected for very dynamic histone modifications. However, some gene expression changes, associated with 3D conformational alterations induced by TSA, persist much longer than the acetylation state including after cell division. This raises several questions about the underlying mechanisms.

In these experiments the authors should clarify/discuss how long these changes persist in culture and approximately how many cell divisions are required for this memory to dissipate.

- To support the claim that cellular memory is changed by TSA repeat treatments the authors must verify the properties of clonally-derived ESCs after the first treatment, and subsequently after the second (If the property is inherited/stable as claimed, then cells can be expanded at each step and the analysed). This is necessary to confirm, at the level of individual cells, that epigenetic memory has been revised.

- The authors should confirm and extend their view that local 3D genome changes at deregulated genes are sustained when cells divide. For example, by using a cell labelling method (such as CFSE) where cells at successive rounds of proliferation can be examined. Providing such evidence would significantly strengthen the claims of this manuscript.

Version 1:

Decision Letter:

Our ref: NG-A67123R

19th Sep 2025

Dear Dr. Cavalli,

Thank you for submitting your revised manuscript "Transient histone deacetylase inhibition induces cellular memory of gene expression and three-dimensional genome folding" (NG-A67123R). It has now been seen by the original referees and their comments are below. The reviewers find that the paper has improved in revision, and therefore we'll be happy in principle to publish it in Nature Genetics, pending minor revisions to satisfy the referees' final requests and to comply with our editorial and formatting guidelines.

Congratulations!

Sincerely,
Chiara

Chiara Anania, PhD
Associate Editor
Nature Genetics
<https://orcid.org/0000-0003-1549-4157>

Reviewer #1 (Remarks to the Author):

The revised manuscript by Paldi and colleagues is greatly strengthened. The authors have clearly gone to considerable lengths to address the feedback and concerns, adding a wide array of new experiments and analyses. The inclusion of extensive RNA-seq, ChIP-seq, ATAC-seq, and Micro-C datasets, as well as controls with a selective HDAC inhibitor and analyses in neural progenitor cells and gastruloids, demonstrates commendable rigor. I want to emphasise that it is not just the scale of new data – which is impressive - but the thoughtful design and implementation that greatly advances conclusions. The additions clarify several key points: first, that TSA-induced changes largely recover at the level of histone acetylation, with quantitative ChIP-seq and Western blotting showing rapid restoration within hours; second, that Polycomb domain interactions underlie architecture-based memory, supported by Ring1B ChIP-seq and PCGF2/PCGF4 depletion; and third, that transcriptional and architectural features of memory persist through confirmed cell division. Together with other clarifications, these represent a significant addition, providing mechanistic insight into how transient hyperacetylation pulses can leave a lasting mark on genome architecture and gene regulation.

Overall, the study is a conceptually important advance into how gene regulation is modulated, and its dynamic responses over time. Importantly, the revised version is better controlled and the conclusions more broadly supported. I therefore believe this study provides valuable insight for the community and should be considered a strong candidate for publication.

At the same time, some issues remain that the authors may wish to comment on or address. For example, although Polycomb looping emerges as a central player, the contribution of other factors such as transcription factor perturbation or non-histone acetylation cannot be fully excluded. This remains a relatively unclear possibility in the manuscript in current format. Moreover, the authors have explained several incongruent results through H2AK119ub potentially acting to promote gene activation (rather than canonical repression), yet none of this is cited or discussed in the manuscript. Perhaps more importantly, the new observation of increasing gene dysregulation over recovery time is to me unexpected and therefore interesting. However, it is not clear how that fits in with the original hypothesis/conclusions, nor that the explanation of a 'psuedo-developmental cascade' fits the paradigm. If there are increasing baseline changes over recovery time, then it follows that the response to a second perturbation may not be 'memory' but an exaggeration of the 'cascade;' that is ongoing. This feels quite important to clarify. Finally, the functional consequences of this memory mechanism for cell fate and physiology, beyond the molecular and architectural signatures described, remain only partially explored.

In sum, the revisions markedly enhance the manuscript and addresses key concerns. The study offers a new angle on the regulation of epigenetic memory, while leaving open interesting questions about mechanistic exclusivity and biological relevance that could inspire future work.

Reviewer #2 (Remarks to the Author):

Nature Genetics manuscript NG-A67123

In this revised manuscript, the authors performed new experiments that reinforce their initial finding of the persistence of a cellular memory induced by transient inhibition of histone deacetylase. These additional experiments provide deeper insight into the molecular changes involved, including analyses of other acetylation targets and examination of longer recovery times following the initial pulse of the epigenetic perturbation. Notably, in this revised version, the authors have addressed the 3D genome maintenance through cell divisions and discussed the clonal transmission of cellular memory.

The authors have meticulously addressed most of the concerns raised in the first submission. They have included significant new set of data and analyses, along with improved clarification of new results and provided a more precise definition of cellular memory.

This study offers new insights into the mechanistic contributions of different epigenetic layers (namely H3K27ac, H3K4me2, and H3K27me3) and genome folding in regulating transcription in ESCs, and how this impact transcriptional memory.

Reviewer #2 (Remarks on code availability):

these look OK

Nature Genetics manuscript NG-A67123 Reviewer response document

Summary of the changes introduced in the revised version of the manuscript

The two reviewers gave a positive assessment of our submitted manuscript version but also raised important points and suggested experiments in order to improve it. We summarize below the main areas in which we have improved the manuscript.

1 – Insight into the molecular mechanism for architecture-based memory

We show that the disruption of Polycomb domain interactions eliminates the transcriptional memory effect at genes that we linked to enhanced Polycomb-mediated chromatin looping.

2 – Showing that 3D genome changes are sustained when cells divide

Through cell generation tracing we demonstrate that similarly to control cells, the entire TSA-treated cell population divides during the 24-hour recovery period. This indicates that the 3D genome can carry information on its past state even through mitosis.

3 – Assessing the dynamics of the recovery from the TSA pulse

We show that the transcriptional identity of cells keeps shifting further from the pre-TSA state with increasing recovery time, arguing against the possibility that the 24-hour recovery would be an intermittent timepoint of an ongoing recovery.

4 – Improving the quantitative analysis of the epigenetic landscape

We rigorously re-analysed our ChIP-seq datasets to systematically and quantitatively identify genomic sites that are differentially enriched between conditions. We complemented this with Western blot and ChIP-qPCR experiments to show that the epigenetic landscape undergoes a virtually complete recovery from the pervasive TSA pulse.

5 – Strengthening the relevance of findings: cell line and perturbation controls

We tested transcriptional and chromatin architecture changes in response to a selective HDAC inhibitor that showed high consistency with our TSA results. We conducted transcriptome analysis of TSA treated and recovered neural progenitor cells (NPCs) that confirmed principles of the recovery, as well as an enrichment of Polycomb targets among continually downregulated genes.

Together, these data converge to show that mouse embryonic stem cells – as well as more differentiated cells – retain a memory of a transient perturbation of their histone acetylation state. In stem cells this memory is stored in architectural changes of the 3D genome and results in regulatory changes at hundreds of genes. Cells can record a hyperacetylation pulse as changes in the physical state of the 3D genome that correspond to gene expression changes that are maintained and even amplified through cell division. This cellular memory results in a more robust gene expression change response upon a further hyperacetylation stimulus.

New datasets and main figure changes in order to respond to the reviewers' comments:

New datasets:

- I. **RNA-seq - 81 datasets (27 conditions, in triplicates) including:** longer (48h, 72h, 96h) and shorter (16h, 20h) recovery times from TSA treatment; increased recovery time between the first and second TSA treatments (48hREC-reTSA, 48hREC-24hREC, 48hREC-48hREC); romidepsin treatment; TSA treatment and recovery (24h, 48h) in NPCs; complete TSA-recovery double treatment time-course in PCGF4^{-/-} PCGF2^{fl/fl} cell line with and without PCGF2 depletion (-OHT/+OHT) and appropriate depletion controls (+OHT 96h, +OHT 120h)
- II. **ChIP-seq, ATAC-seq – 66 datasets (27 conditions, in duplicates) including:** Ring1B and H3K9ac ChIP-seq in DMSO, TSA and REC; YY1 ChIP-seq in TSA; H3K4me1, H3K4me3, H2AK119Ub in reTSA; ATAC-seq, H3K27ac, H3K27me3 in reTSA and reREC; H3K27me3 and H2AK119Ub in PCGF4^{-/-} PCGF2^{fl/fl} cell in -OHT, +OHT +DMSO, +OHT +TSA, +OHT +REC and appropriate inputs
- III. **Micro-C – 6 datasets (5 conditions, one of them in duplicates):** Romidepsin; PCGF4^{-/-} PCGF2^{fl/fl} cell line in -OHT, +OHT +DMSO, +OHT +TSA, +OHT +REC

Figure 1: Panel **d** has been updated with differential ChIP-seq peaks from our revised quantitative analysis.

Figure 2: Panel **e** and **f** have been updated with new parameters to model the TSA condition.

Figure 3: Panel **a** and **d** have been updated with differential ChIP-seq peaks from our revised quantitative analysis. Panels **b** and **c** have been updated to show ChIP-seq signal following normalisation against new datasets.

Figure 4: Panel **b** has been updated to show ChIP-seq signal following normalisation against new datasets. Plot axes have been adjusted in panel **d**. Panel **e** has been updated with differential ChIP-seq peaks from our revised quantitative analysis.

Figure 6: Old Extended Data Fig. 7b became panel **b**. Revised panels **c-i** correspond to old panels b-h. Plot axes have been adjusted in panel **c**. Panels **f, h** now include ChIP-seq (f and h) data on the second TSA and recovery (reTSA, reREC).

Figure 7: Entirely new figure that introduces novel key findings on the architecture-based memory at Polycomb target genes.

Extended Data Figure 1: Western blot showing cytoplasmic lysine acetylation signal has been replaced by a new one and is shown in panel **b**, along with cytoplasmic tubulin acetylation signal. Revised panels **c, f-i** correspond to old panels b, d-g. New cell cycle profiles by propidium iodide staining and its quantification are shown in panels **d-e**. Panel **g** and **h** have been updated with differential ChIP-seq peaks from our revised quantitative analysis.

Extended Data Figure 2: Old Extended Data Figure 2 and 3 have been combined, with old Extended Data Figure 3 panels a-m now shown in revised Extended Data Figure 2 **e-o, s-t**. Panels **j-m** have been updated with new parameters to model the TSA condition. New results

showing changes in radial distribution of H3K27ac signal upon TSA treatment are shown in panels **p-r**.

Extended Data Figure 3: Entirely new figure that shows transcriptional and genome folding changes induced by the selective HDAC inhibitor romidepsin.

Extended Data Figure 4: Panel **a** has been updated to show ChIP-seq signal following normalisation against new datasets. Panel **b** has been updated with differential ChIP-seq peaks from our updated quantitative analysis. Panel **d** has been added to show differential ChIP-seq peak distribution relative to differentially expressed gene promoters. Revised panels **e-i** correspond to old panels d-h.

Extended Data Figure 5: Entirely new figure that demonstrates recovery of the epigenetic landscape. Cell generation tracing experiment in panel **b** shows that the entire cell population divides during the recovery period.

Extended Data Figure 6: Entirely new figure that shows transcriptional changes with increasing recovery times (panels **a-d**) and unlinks these effects from HDAC action on non-histone targets through motif analysis (panels **e-h**).

Extended Data Figure 7: Revised panels **a-e** correspond to old Extended Data Figure 5 panels a-e. Panels **f-i** include new results on the response to - and recovery from - TSA pulse in neural progenitor cells (NPCs).

Extended Data Figure 8: Revised panels **a-c** correspond to old Extended Data Figure 6 panels a-c. Panels **d-h** show new data on second TSA pulse following a 48-hour initial recovery period. Panel **i** shows H3K9ac peaks that do not recover relative to gene promoters that do or do not remain deregulated.

Extended Data Figure 9: Revised panels **a**, **c-d** and **f-g** correspond to old Extended Data Figure 7 panels c-e and f-h. Panel **d** shows looping propensity of recovery vs non-recovery TSSs. Panel **e** shows H3K27me3 enrichment at gene promoters in NPCs that recover or do not recover from the TSA pulse.

Extended Data Figure 10: Entirely new figure that shows supplemental information related to the disruption of Polycomb interaction domains via PCGF2/PCGF4 depletion.

Point-by-point response to the comments of each reviewer

This revision required relatively complex new experimental work and data analyses, which resulted in several changes in figures and text. To facilitate the navigation through the document, **we wrote our responses in blue**. For each point, in addition to the response text, **we pasted the corresponding figure panels and figure legends that are used in the revised manuscript**, such that reviewers do not need to go to the manuscript to track the changes. Furthermore, **key sentences that we introduced in the manuscript to address reviewer's issues are transcribed in green** in our response text. **Additionally, we introduced some Reviewer Figures**, which present additional data that are not introduced in the manuscript in order to avoid unnecessary complexity for the readers, but which can help reviewers in their assessment. Finally, we insert below a "**Table of contents**", reporting each reviewer questions as subheadings that are **hyperlinked to the corresponding pages**. We hope that this will help the reviewers reading through our responses.

Table of Contents

Summary of the changes introduced in the revised version of the manuscript	1
New datasets and main figure changes in order to respond to the reviewers' comments:	2
Point-by-point response to the comments of each reviewer	3
Reviewer #1 (Remarks to the Author):	5
Key comments & concerns (in order of occurrence in manuscript)	6
___ Reviewer#1 major comment 1: Specificity of the perturbation	6
___ Reviewer#1 major comment 2: Epigenetic landscape analysis	7
___ Reviewer#1 major comment 3: Principle of the recovery	13
___ Reviewer#1 major comment 4: TF targets	18
___ Reviewer#1 major comment 5: Second recovery	21
___ Reviewer#1 major comment 6: Gastruloids as a model	23
___ Reviewer#1 major comment 7: Cell type	23
___ Reviewer#1 major comment 8: Perturbation controls	26
___ Reviewer#1 major comment 9: Mechanisms	28
___ Minor comments	32
Reviewer #2 (Remarks to the Author):	35
Specific Points:	42
___ Reviewer#2 major comment 1 – Other primary changes in histone acetylation	42
___ Reviewer#2 major comment 2 – Changes in chromatin accessibility	47
___ Reviewer#2 major comment 3 – Chromatin interactions at upregulated genes	48
___ Reviewer#2 major comment 4 – Longer recovery times	49
___ Reviewer#2 major comment 5 – Clonal transmission of epigenetic memory	54
___ Reviewer#2 major comment 6 – 3D genome through cell division	55
___ Minor comments	56
Reviewers' references	57

Reviewer #1 (Remarks to the Author):

In this manuscript, Paldi and colleagues investigate important and still unresolved questions around epigenetic/cellular memory. The overarching approach is to apply a pervasive but transient stimulus in mouse embryonic stem cells and measure the transcriptional, epigenomic and 3D-chromatin state responses at multiple time points during and after perturbation. More specifically, the authors apply the deacetylase inhibitor TSA for 4hrs at a dosage that leads to major changes preferentially in histone acetylation levels. They then release cells and assay across omics modalities. The study can be considered in two broad parts. Firstly, is a very careful and comprehensive characterisation of how perturbed (elevated) global acetylation impacts molecular layers of regulation in ESC, using ultra-deep Micro-C integrated with RNA-seq and ChIP-seq. In this survey, the authors note a number of interesting responses, including preferential interactions between BB compartments in cis (and associated model), widespread histone modifications changes (including some unexpected) and a profoundly altered transcriptome (>4,000). Second, the authors test the ability of these 4hr TSA-treated cells to recover their transcriptional, epigenomic, and 3D-structural state after 24 hours from TSA washout. Here, they observe that transcriptional and epigenomic features are mostly reversed to the pre-TSA state (78 and 86 down- and up-regulated genes persist, respectively), while genome conformation retained some features of the TSA-induced state, in particular at the level of cis-interactions, and at the sub-megabase scale for specific instances of gene loops. This leads to the central conclusion that chromatin looping carries a memory of prior perturbations. Beyond this, the authors also ask the interesting question of how a second round of stimulus/release affects responses, finding that a subset of genes exhibit a seemingly enhanced second response – further implying an epigenetic memory.

The study is communicated in a clear and well-structured manner. Moreover, it addresses a question that is of wide general interest – both from a genome regulation perspective and from the broader question of transmitting 'epigenetic' memory. In my view the study could make important contributions and should be considered by Nature Genetics in principle. However, there are a number of interpretations and confounders that make strong conclusions very difficult in present form, and at least two major areas of design/conceptual concern. Firstly, based on the data presented the perturbation affects many molecular layers and is still persisting at 24hrs washout, meaning that rather than measuring a memory of looping, the observed changes could reflect an intermittent timepoint in a dynamic – but incomplete – recovery of many factors. Second and related, the argument that mechanistically it is the “genome architecture [that] carries a memory of its TSA-induced effects” is not supported by direct (rather than correlative) evidence and other possibilities are not ruled out. Given this is central for the novel insight of the study, this could be strengthened. Below I go to into further detail on these points and highlight some other concerns as well as the strengths of the study.

We thank the reviewer for the positive comments and hope that the following point-by-point response correctly addresses pending questions.

Key comments & concerns (in order of occurrence in manuscript)

Reviewer#1 major comment 1: Specificity of the perturbation

It is evident from literature and the author data that TSA is not just a nuclear histone perturbation – despite careful titration of dosing - but likely pleiotropic. For example, in Fig S1A an entirely new band appears upon (re)TSA exposure in the cytoplasmic fraction. More generally, the overall intensity of the cytoplasmic acetylation fraction also increases upon TSA, supporting non-histone changes in protein acetylation. Finally, there appears to be a subtly different cell cycle profile upon TSA (in Fig S1C – increase in transition population). Taken together, it appears the TSA treatment is not just a chromatin perturbation, which has ripple effects for interpreting the remaining data. The statement that “the bulk of the acetylome, cell cycle progression and cell viability remain unchanged” might be too strong.

Following the reviewer’s comments, we decided to optimise Western blotting conditions for cytoplasmic lysine acetylation. These new blots show that bulk acetylation levels remain virtually unchanged in TSA condition (Extended Data Fig. 1b, top panel). Pan-lysine acetylation may appear slightly stronger following the second TSA pulse (reTSA), however, careful examination of the Ponceau staining suggests slightly higher loading in the gel lane corresponding to reTSA. A few specific bands indeed gain signal in TSA, but they recover nevertheless, which is the central aspect to our study. The most prominent band pointed out by the reviewer is tubulin (Extended Data Fig. 1b, bottom panel), which is indeed a target of HDAC6¹. Similarly to other proteins, tubulin acetylation levels are restored following TSA washout, excluding long-term off-target effects of HDAC inhibition on cytoplasmic targets.

Extended Data Figure 1b - Western blots showing the levels of lysine acetylation (top panels) and levels of acetylated tubulin (bottom panels) in cytoplasmic protein extracts in the following conditions: DMSO control, TSA treatment, 24-hour recovery following TSA washout, sequential TSA treatment and 24-hour recovery from the second TSA treatment. Ponceau staining is shown as loading control on the right.

As tubulin acetylation modulates microtubule dynamics, there was a possibility that the effect of TSA on tubulin would interfere with chromosome segregation. Therefore, we performed cell cycle profiling by propidium iodide staining followed by flow cytometry (Extended Data Fig. 1d). Following the reviewer’s concern, we repeated this experiment in biological triplicates and

performed cell cycle modelling during our TSA-recovery double treatment time course. Statistical analysis indicated that there is no significant change in the proportion of G1 vs S vs G2/M cells, except for a mild increase in the S phase population (on the expense of G1) in the first recovery (Extended Data Fig. 1e). As this effect is specific to the first recovery and does not worsen throughout the time course, we think that it is unlikely that it would interfere with downstream interpretations of our results.

Finally, according to the reviewer’s suggestion we rephrased the manuscript text in lines 69-71 as follows: “To minimise pleiotropic effects, we optimised treatment conditions so that histone hyperacetylation is strongly induced but the bulk of the acetylome, cell cycle progression and cell viability remain minimally affected (Extended Data Fig. 1a-f).”

Extended Data Figure 1d, Cell cycle analysis by flow cytometry using propidium iodode staining. **e**, Fraction of cells in G1, S and G2/M at each time point of the TSA-recovery treatment course.

Reviewer#1 major comment 2: Epigenetic landscape analysis

Mapping the chromatin modification changes in response to TSA both establishes an interesting dataset and provides insights into response relationships. It has clearly been performed carefully and in a calibrated manner so, like the micro-C, it is a powerful resource. However, some of the core observations are strange and hint at confounders – or should be further followed up.

We are very happy that the reviewer appreciates that our datasets will constitute a useful resource to the scientific community even beyond the scope of the current study.

(i) Almost all chromatin marks increase upon TSA, including the repressive H2A119ub (Fig3A-B) at all target genes. This could be interesting but could also be an artefact related to increased antibody access to histone epitopes upon global acetylation. In other words, does ChIP just pull down more of everything upon TSA treatment due to technical reasons. If not, how do you explain almost all marks going up – particularly repressive ones.

We thank the reviewer for this remark which prompted us to rigorously re-analyse our ChIP-seq datasets in a meticulous manner. While ChIP-seq is useful to quantitatively compare differences in signal enrichment within the same sample, it is poorly quantitative between different conditions. Although spike-in calibration² greatly improved absolute level comparisons between samples, quantitative comparisons are still often hampered by variability introduced by multi-step sample preparation performed at different times^{3,4}. This scenario is particularly tricky when – like in our case - global changes in signal are expected between conditions. Therefore, optimal choice of data analysis tools is critical for accurate detection of differential binding events in order to draw adequate biological conclusions.

To this end, at the beginning of this study we carefully considered analysis tools and opted for the use of the DiffBind package. DiffBind was systematically identified as one of the highest performing quantitative ChIP-seq analysis tools in multiple biological scenarios, but in particular when global increase or decrease is expected in ChIP signal with broad peak shapes (like histone modifications)⁵. Additionally, DiffBind allows for the integration of spike-in library normalisation with the DESeq2 differential binding algorithm, using multiple biological replicates⁶. Thus, it constitutes the most optimal tool to analyse the epigenetic landscape following TSA-induced hyperacetylation.

Spike-in calibration in conjunction with DESeq2 quantitative analysis revealed that global signal increase in TSA condition is only induced for H3K27ac and ATAC-seq (shown in the new Extended Dat Fig. 5g, which we reproduce below). Activating histone marks H3K4me3 and H3K4me1 show prominent increase (>10,000 sites) while repressive histone marks H3K27me3 and H3K9me3 remain virtually unchanged. We detect a few hundred of increased peaks for the Polycomb Repressive Complex 1 (PRC1) mark H2AK119Ub. Although historically considered as transcriptional repressors, PRC1 complexes – and their deposited histone modification – have been implicated in transcriptional activation, both in mammals^{7,8} and *Drosophila*⁹, in a context-dependent manner.

Extended Data Figure 5g. Diverging bar charts showing the number of differential ChIP-seq and ATAC-seq peaks identified by DESeq2 analysis in TSA vs DMSO (TSA) and 24-hour recovery vs DMSO (REC) conditions.

Importantly, Western blotting supports the results of differential binding analysis, showing TSA-induced increase in H3K27ac, H3K4me1 and H3K4me3 levels, and steady H3K27me3, H2AK119Ub and H3K9me3 levels (Extended Data Fig. 5e-f). Therefore, we conclude that TSA treatment specifically induces an overall gain of activating chromatin marks, while leaving the repressive histone modification landscape unchanged. This argues against TSA-induced technical artefacts due to increased antibody accessibility.

Extended Data Figure 5e, Heatmaps showing scaled ChIP-seq signal per epitope in merged replicates at all enrichment sites in DMSO, TSA and REC conditions. **f**, Western blots showing total level of histone modifications in nuclear extracts. Lamin B is shown as loading control.

(ii) Related to above, almost all chromatin modifications exhibit the same the same degree of change at both UP and DOWN-regulated gene sets (Fig 3b-c). That is, the intensity of both activating and repressing marks increases irrespective of the direction of transcription changes. This implies they are not functionally linked and could further hint at artefacts in performing ChIP with global acetylation increases, such as universal accessibility for antibodies. There only appears to be one exception, which is H3K4me3 – this increases specifically in upregulated genes but does NOT increase in downregulated genes. This could imply H3K4me3 has a functional role in response to HDAC inhibitors, which is testable.

We agree with the reviewer that heatmaps of ChIP-seq signal at transcription start sites (TSSs) could give the impression that changes in the histone landscape and changes in gene expression become uncoupled in the TSA condition. Therefore, we decided to examine the distribution of differential ChIP-seq peaks relative to differentially expressed gene TSSs (Fig. 3a; Extended Data Fig. 4d). This has shown that ‘up’ peaks of activating histone marks (H3K27ac, H3K4me3, H3K4me1) are more often located at upregulated TSSs, while ‘down’ peaks (H3K27ac, H3K4me3) are more frequently localised to downregulated TSSs. H2K119Ub ‘up’ peaks were found to be equally frequent around up- and downregulated TSSs, consistent with the context-dependent role or PRC1 in gene activation and repression.

Figure 3a; Extended Data Figure 4d. Differential ChIP-seq peak count frequency distribution relative to TSSs of up- and down-regulated genes.

Additionally, enhancers are known to be able to positively influence gene expression located large distances away from their promoter targets. Thus, we looked at the proximity of H3K4me1 sites that gain signal strength in TSA relative to up- and downregulated gene promoters. This showed that increased H3K4me1 peaks are more closely located to upregulated TSSs than downregulated ones (Reviewer Fig. 1).

Reviewer Figure 1. Cumulative histogram showing genomic distance between up- or downregulated gene promoters and the nearest increased H3K4me1 peak.

Finally, as presented in the original version of the manuscript, we suspect that gene expression downregulation is partially due to the effects of HDAC inhibition on transcription factor targets, as exemplified by an enrichment of HDAC targets YY1 and Myc at downregulated TSSs relative to upregulated ones (Extended Data Figure 4c). The relevance and consequences of this observation will be further discussed below, related to Reviewer#1 major comment 4.

Overall, our peak frequency distribution analysis established a functional link between the behaviour of activating histone marks and deregulated TSSs. Examination of potential enhancers activated by TSA treatment (as signified by an increase in H3K4me1 signal) revealed a more distal localisation of such sites from downregulated TSSs, highlighting an additional difference between up- and downregulated TSSs. Finally, HDAC action on non-histone targets could also constitute a mechanism of gene downregulation, independently from changes in the histone modification landscape. To reflect on these conclusions more clearly in the manuscript, we updated the text in lines 245-249 as follows:

“However, closer inspection of differential ChIP-seq peak distribution relative to deregulated TSSs revealed that regions that lose signal enrichment for activating histone marks (H3K27ac, H3K4me3) are more frequently located near downregulated TSSs (Fig. 3a, Extended Data Fig. 4d), indicating that functional coupling is preserved between changes in the histone landscape and gene expression changes.”

(iii) Further related to the increased modifications, the authors conclude “Altogether, we find that gene upregulation as well as downregulation occur with a gain of activating chromatin marks” but no mention is made of the increase in repressive marks at active genes. Instead, it is concluded that “while gene upregulation is potentially associated with enhancer over-activation without gain in E-P contacts, downregulation is linked to repressive chromatin looping. It’s not clear to me how these collective conclusions are made. There are changes in many epigenomic layers so how can the authors claim these are the relevant mechanisms.

Following the reviewer’s remark and new analyses presented in point (ii) above, we updated the manuscript text in lines 261-262 as follows:

“Altogether, we find that gene upregulation occurs with a gain of activating signal while gene downregulation happens without a gain of repressive chromatin marks.”

We believe that by “increase in repressive marks at active genes” the reviewer is referring to an apparent increase in H2AK119Ub signal at upregulated TSSs as H3K27me3 signal shows no change. Should this be the case, we would like to refer to our response in point (i) where we cite recent work demonstrating that H2AK119Ub should not be considered strictly as a repressive histone mark.

To the second part of this comment, the conclusion that “gene upregulation is potentially associated with enhancer over-activation without gain in E-P contacts” is made based on the findings that H3K4me1 peaks that increase in signal intensity are located around upregulated genes (Fig. 3d). As gene activation is commonly (but now always) associated with gain in physical contacts between enhancers and their promoter targets¹⁰, we filtered our Micro-C loops sets for loop anchors that coincide with differentially expressed gene TSSs (Fig. 3f). Contrarily to what was expected, we did not find increased looping at upregulated TSSs, implying that changes in the histone modification landscape – rather than 3D chromatin contacts – could be responsible for TSA-induced gene expression upregulation.

Figure 3d, Cumulative histogram showing genomic distance between upregulated gene promoters and the nearest increased H3K4me1 peak. Control genes represent an expression-matched gene set that does not increase in expression. **Figure 3f**, Aggregate plots of Micro-C signal around loops where anchors overlap with down- (left) or up-regulated (middle) TSSs, as well as bivalent (right) TSSs that

undergo upregulation (resolution = 4 kb). Quantification of piled-up loop signal is shown on the right (paired two-tailed t-test; ns > 0.05, ***p < 0.001).

This, however, was not the case for downregulated gene promoters. Although gene downregulation happened without a gain in activating histone marks, we did not observe any gain in repressive histone marks either (see response to point (ii) above). Therefore, we took a similar approach and looked at 3D chromatin contacts that emanate from downregulated TSSs: unlike upregulated TSSs, downregulated TSSs made significantly stronger chromatin contacts in TSA treated cells (Fig. 3f). This led us to the idea that repressive chromatin contacts might constitute a prominent mechanism that can drive gene expression downregulation in a scenario where global chromatin activation is induced. We can of course not exclude the possibility that other histone marks or factors participate in the repression mechanisms, but we believe that our interpretation is a parsimonious scenario that might explain the data.

(iv) The role of looping in gene repression is possibly under-explored and potentially very interesting. Why is there increased looping at Polycomb targets but no increased Polycomb?

We thank the reviewer for this excellent question which prompted us to carefully consider potential mechanisms of chromatin looping-mediated gene repression. Polycomb domain interactions are not necessarily dependent on the catalytic activity of PRC1 and PRC2 complexes, rather, they seem to be linked to the physical presence of canonical PRC1 complexes on chromatin^{11,12}. Thus, we hypothesised that TSA treatment might induce increased PRC1 binding to chromatin which in turn could lead to increased Polycomb domain interactions. To this end, we performed calibrated ChIP-seq against the Ring1B subunit of PRC1, followed by quantitative identification of differential binding sites. Interestingly, we indeed found increased Ring1B occupancy at several thousands of sites in TSA (Reviewer Fig. 2a-b), implying that binding of PRC1 complexes to chromatin – rather than H2K27me3/H2AK119Ub – may be responsible for increased looping.

Reviewer Figure 2. **a**, Heatmaps showing scaled Ring1B ChIP-seq signal per epitope in merged replicates at all enrichment sites in DMSO, TSA and REC conditions. **b**, Diverging bar charts showing the number of differential Ring1B ChIP-seq peaks identified by DESeq2 analysis in TSA vs DMSO (TSA) and 24-hour recovery vs DMSO (REC) conditions.

Reviewer#1 major comment 3: Principle of the recovery

(i) While there is clearly a significant reversion of gene expression and acetylation, it appears incomplete to me. For example, H3K27ac marks are clearly not fully recovered at 24hrs in Fig 4B. Authors should provide a FC change plot of DMSO vs REC to support the argument that “original H3K27ac levels and the rest of the acetylome appeared restored at the protein level. Given that Fig4B suggests increased K27ac (DMSO vs REC) – and that H3K4me1/ATAC also appears different between DMSO and REC – it is difficult to make this conclusion. Moreover, if the chromatin state has not recovered then not only are the conclusions that favour looping less clear-cut, but it calls into question whether it is ‘memory’ being measured or a stage of an ongoing dynamic recovery.

We performed a series of new experiments and analyses to strengthen our conclusions on the recovery of the histone modification landscape - with a particular emphasis on the recovery of H3K27ac levels that constitute the initial causative event.

First, we analysed the recovery of H3K27ac at the protein level in a time-resolved manner. This indicated that H3K27ac is extremely dynamic and its pre-TSA levels are restored within as little as 3 hours following TSA removal (Extended Data Fig. 5a). This reinforces the idea that the 24-hour timepoint constitutes a relatively long-term recovery period compared to our short 4-hour TSA treatment. Thus, by the time cells are re-assayed for their transcriptional, epigenomic and chromatin architecture states, H3K27 acetylation levels have been restored for >20 hours.

Extended Data Figure 5a - Western blots showing the levels of H3K27 acetylation DMSO, TSA and varying time intervals following TSA removal. Lamin B1 is shown as loading control.

Next, we performed H3K27ac ChIP-qPCR – a low throughput but highly quantitative approach to study DNA-protein interactions – at a handful of loci. This indicated that H3K27 acetylation is similar to DMSO levels following the 24-hour recovery period at all tested loci (Extended Dat Fig. 5c). Additionally, we performed a third H3K27ac ChIP-seq replicate in DMSO, TSA and REC conditions to minimise variability introduced by experiments performed at different times. This confirmed that chromosome-wide H3K27ac levels are efficiently restored following the 24-hour recovery period (Extended Dat Fig. 5d).

Extended Data Figure 5c, ChIP-qPCR showing H3K27ac signal in DMSO, TSA and 24-hour recovery (REC) conditions at the ActB and Oct4 promoters that have high levels of acetylation (left panel) and at the Zfp608 promoter and an Oct4 upstream region largely devoid of acetylation (right panel). Data shown are mean of two biological replicates \pm s.e.m is shown (unpaired left-tailed t-test; ns $p > 0.05$). **d**, Spike-in normalised H3K27ac ChIP-seq signal per chromosome in replicate 3 (unpaired two-tailed t-test; ns $p > 0.05$).

Finally, previously presented differential binding site analysis (see again below) identified only ~600 sites with increased H3K27ac enrichment in the recovery condition (Extended Dat Fig. 5g). Western blotting also supports recovery of the epigenetic landscape, with TSA-induced increase in H3K27ac, H3K4me1 and H3K4me3 being restored 24 hours after drug removal (Extended Dat Fig. 5f).

Extended Data Figure 5g, Diverging bar charts showing the number of differential ChIP-seq and ATAC-seq peaks identified by DESeq2 analysis in TSA vs DMSO (TSA) and 24-hour recovery vs DMSO (REC) conditions.

To understand if remaining increase in H3K27 acetylation could be responsible for the transcriptional memory effect at certain genes, we analysed their distribution relative to TSSs that recover and do not recover from TSA treatment. This analysis showed that H3K27ac 'up' peaks in recovery show no particular association with TSSs that do not recover their expression states following TSA treatment (Fig. 4e).

Figure 4e H3K27ac peak count frequency distribution in recovery dataset relative to gene promoters that remain up- and down-regulated (TSS = transcription start site). Shading represents 95% confidence interval.

(ii) similarly to histone modifications, it appears that gene expression is largely recovered - but not completely (Fig 4d). When comparing DMSO with REC, many hundreds of genes with LFC>2 remain (not all are significant). Thus, there is still at least some significant levels of altered transcriptional networks at 24hrs and presumably the proteome is even slower in recovering, given the half-life of many proteins, and thus in an even more perturbed state. As a note, why is the volcano plot in Fig 4D scaled to 300 on the y-axis when the highest data point is 25. The effect is to compress the data unnecessarily, giving the impression of low significance by eye.

Following the reviewer's request, we now adjusted plot axes in Fig. 4d. Careful cutoff selection is essential for defining differentially expressed genes and it improves the reliability of results. In this study, we chose to apply the routinely used adjusted p-value threshold <0.05, and 2-fold magnitude of difference, represented by the absolute log₂ fold change (L2FC) >1, to filter our gene expression datasets¹³. Lowering either threshold would increase the number of genes identified as differentially expressed, but it would also come at the expense of introducing more noise and false positives in the analysis. Therefore, we maintain our view that it is important to filter genes using multiple cutoffs to improve the reproducibility and interpretation of our results.

Figure 4d. Volcano plot showing differential gene expression 24 hours after TSA washout. Labels show developmental marker genes that remain above the significance threshold (adjusted p-value > 0.05, absolute log₂ fold change > 1).

Together, the gene expression and chromatin states (and looping) in REC suggests that the reversion is incomplete. This makes drawing conclusions on memory challenging. To get

around this issue, I believe it is important to measure responses at longer timepoints (i.e. 48hrs, 96hrs etc) and also think about more regular and intermediate timepoints to record the rates of change between modalities. At present it is difficult to (i) confirm which regulatory layers have or have not recovered relative to each other (i.e. which is providing the memory) and (ii) whether it is memory or a freeze-frame during a dynamic recovery.

We thank the reviewer for this excellent suggestion. We agree that it is critical to test longer recovery times, therefore, we performed RNA-seq in 48-, 72- and 96 hours following TSA washout. Interestingly, we found gene expression differences increasing – rather than dissipating – with increasing recovery times. This is exemplified by an increasing distance between DMSO and consecutive recovery time points in principle component analysis (PCA), as well as the sheer number of differentially expressed genes that pass our filtering thresholds (Extended Data Fig. 6a-b). Conversely, we also found higher number of differentially expressed genes at shorter recovery times (16 hours and 20 hours), indicating that the 24-hour time point represents the most recovered state among the tested.

Extended Data Figure 6. a, Principal component analysis (PCA) analysis showing increasing distance between DMSO and recovery (REC) samples with increasing recovery time. **b**, Diverging bar plot showing the number of differentially expressed genes at the indicated recovery times.

When we looked at the behaviour of the gene groups that recover (REC) and do not recover (nonREC) from TSA, we observed that while recovery genes have a stable expression on the long term, non-recovery genes undergo progressive gene expression deregulation (Reviewer Fig. 3a). This provides further support to an ongoing cellular memory – that being a sustained cellular response to a transient stimulus. Importantly, ChEA Transcription Factor Target enrichment among downregulated genes showed that continued gene downregulation on the long term also relies on Polycomb-mediated repression (Reviewer Fig. 3b).

Reviewer Figure 3. a, Heatmaps showing mean expression z-Scores of recovered and not recovered genes through the TSA-recovery treatment course. **b**, Bubble plot showing enrichment of downregulated genes that do not recover in the ChEA Transcription Factor Target database.

We then analysed Gene Ontology (GO) term enrichment among deregulated genes throughout our recovery time series that showed partially overlapping developmental categories. Thus, we suspect that mESC initially make a near-complete transcriptional recovery, following which they undergo a pseudo-developmental cascade (Extended Data Fig. 6c-d).

Extended Data Figure 6c-d, Bubble plots showing Gene Ontology (Biological Process) enrichment among up- (**c**) and downregulated (**d**) genes at different recovery intervals.

To visualise functional enrichment results at the different recovery time points, genes in the major developmental categories were plotted on network diagrams (Reviewer Fig. 4a-b). This highlights complex associations between different days of the recovery, further arguing for an ongoing developmental cascade. Thus, rather than representing an intermittent time point during an ongoing recovery, the 24-hour time point is the closest to the DMSO cellular state.

These new critical results are now summaries in lines 307-313 of the manuscript text as follows:

“In order to exclude the possibility that the near-complete recovery is simply due to an insufficiently long time period, we assayed gene expression changes 48-, 72- and 96-hours following TSA removal. Interestingly, this revealed that the number of differentially expressed genes increases with longer recovery times as cells appear to undergo a seemingly unguided developmental cascade (Extended Data Fig, 6a-d). Conversely, assessing gene expression changes at shorter recovery times (16 hours and 20 hours) showed that gene expression changes are the lowest at 24 hours.”

Reviewer Figure 4. a-b, Network diagrams showing relationships between non-recovery (a – upregulated, b – downregulated) genes at the indicated recovery times.

Reviewer#1 major comment 4: TF targets

Beyond histone and gene expression (incomplete) recovery, there are also questions around protein acetylation. Specifically, the authors mention that TSA-induced HDAC inhibition can have effects beyond histone acetylation (231-244), including an increase of acetylation of the transcription factor YY1 and MYC. Both TFs are known to be crucial transcriptional regulators, targeting thousands of genes. It could thus be that a portion of the transcriptional effects measured upon TSA treatment and recovery are a consequence of differential TF function rather than epigenomic and chromatin conformation changes. The authors show in Fig.E4c that genes downregulated upon TSA treatment are enriched for YY1 and Myc binding sites. To clarify the importance of differential TF activity upon TSA treatments and recovery phases, it could be valuable (albeit optionally) to perform ChIP-seq for the two factors in all conditions measured in the MS (i.e. DMSO, TSA, recovery, 2nd TSA and 2nd recovery). It could in fact be the case that acetylation of these TFs has a slower recovery dynamic as compared to, for example, nucleosome acetylation. In this case, some of the transcriptional “memory” features

that the authors attribute to chromatin 3D-structural changes could be instead caused by TF activity.

We thank the reviewer for encouraging us to clarify this important point. To understand if sustained gene expression deregulation could be due to modulated transcription factor (TF) activity induced by HDAC inhibition, we carried out motif analysis at deregulated gene promoters. Next, we crossed the list of enriched motifs with a list of characterised non-histone proteins that undergo deacetylation¹⁴, and filtered results for proteins that are targets of Class I, Class II or Class IV HDACs (those being inhibited by TSA) expressed in mESCs. Such proteins are marked with asterisk on the bubble plots below (Extended Data Fig. 6f-g).

Extended Data Figure 6f-g. Bubble plots showing HOMER motif enrichment analysis at up- (f) and downregulated (g) gene promoters of genes that recover from and genes that remain deregulated following TSA treatment. Direct Class I, II or IV HDAC targets that have detectable expression levels in mouse ESCs are marked with asterisk (*).

This analysis demonstrated that while some genes that recovered from TSA treatment could be clearly linked to certain non-histone HDAC targets (eg. YY1, c-Myc, Sp1), genes that remained deregulated could not. The sole exception to this is GATA3 whose targets (4 genes in total) are enriched among upregulated genes at 24 hours. This however, is unlikely to happen through the direct action of HDACs on GATA3, given that HDAC inhibition attenuates – rather than activates - GATA3-dependent gene transcription in lymphoma¹⁵.

Despite the fact that these computational analyses do not support the hypothesis that TF acetylation is linked to memory, we decided to test this idea experimentally. As YY1 was the top enriched motifs among downregulated genes that recover from TSA treatment, we performed ChIP-seq on YY1 in TSA condition (Extended Data Fig. 6e). Importantly, this showed that YY1 binding at downregulated TSSs remains unchanged in TSA and binding of YY1 to non-recovered downregulated genes is very weak, both in control and TSA conditions.

This makes it unlikely that persistently altered YY1-chromatin interactions (as a result of modulated YY1 acetylation state¹⁶) would be responsible for long-lasting transcriptional effects.

Transcription factor binding leads to chromatin opening both at promoter sites as well as distal *cis* regulatory elements (enhancers). Thus, instead of restricting our analysis to gene promoters, we decided to carry out motif enrichment analysis at all ATAC-seq sites that remain open following the first or second recovery periods (Extended Data Fig. 6h). Then, like above, we crossed the list of enriched motifs with a list of characterised non-histone proteins that undergo deacetylation¹⁴, and filtered results for proteins that are targets of Class I, II or IV HDACs expressed in mESCs. This revealed that sites of continually increased chromatin accessibility are not enriched in motifs of direct HDAC targets. An exception to this is Sp1, whose motif is enriched among sites that remain open in the first recovery. However, given that 1) Sp1 can function as a transcriptional repressor by recruiting HDAC1-containing complexes¹⁷ and 2) Sp1 itself is downregulated in TSA treatment, it is difficult to envisage plausible mechanisms that might lead to prolonged chromatin opening at Sp1 motifs.

Extended Data Figure 6e. Normalised YY1 ChIP-seq signal at downregulated TSSs that recover (top) and do not recover (bottom) in DMSO and TSA. **h.** Heatmap showing HOMER motif enrichment analysis at differential ATAC-seq peaks at 24-hour recovery following the first and second TSA treatments.

Despite all these data, we of course cannot fully exclude the possibility that some continued gene expression deregulation is due to previously uncharacterised non-histone HDAC target. Nevertheless, these complex analyses strongly support the idea that the transcriptional memory induced by TSA treatment is not due to altered acetylation state of non-histone HDAC targets. Therefore, we updated the manuscript text in lines 313-316 as follows:

“Crucially, we did not find any link between non-histone targets of HDACs and sustained gene expression deregulation, minimising the possibility that long term transcriptional consequences would be due to off-target effects of HDAC inhibition (Extended Data Fig, 6e-h).”

Reviewer#1 major comment 5: Second recovery

If one accepts that there is not complete recovery after 24hrs then the second exposure could be additive on top of an ongoing perturbation rather than evoking a memorized response in genome architecture. This would be important to discuss or consider testing experimentally – for example checking if there is a second perturbation boost at 48hrs (or longer) after the 1st exposure.

Thank you very much to the reviewer for this insightful suggestion which we decided to approach from two different angles. First, we compared transcriptional and chromatin effects of the first and second TSA pulses. This revealed that the transcriptional response (Fig. 6b) to the two pulses is highly similar, and that changes in the chromatin landscape are virtually identical (Extended Data Fig. 8e). We detect ~3000 H3K4me3 peaks where signal in the reTSA condition is decreased compared to the TSA condition, and vice versa, ~3000 peaks where ATAC-seq signal is increased in reTSA compared to TSA. Overall, these argue against an additive effect between the two TSA pulses.

Figure 6b. Scatterplot showing the correlation of transcriptomic changes induced by the first and second TSA treatments. **Extended Data Figure 8e.**, Number of differential binding sites between the first and second TSA treatments.

Next, as the reviewer suggested, we sought to understand if a second perturbation had a similar effect if the first recovery period was increased from 24- to 48 hours. Thus, we carried out RNA-seq in a second TSA pulse following a 48-hour initial recovery period (48hREC-reTSA). We also monitored recovery from this second pulse at 24 and 48 hours (48hREC-24hREC, 48hREC-48hREC). We found that transcriptional response to the second TSA pulse was highly similar whether the first recovery period was 24 hours or 48 hours: this was visible from the number of differentially expressed genes as well as from the scatterplot showing high correlation between the two conditions (Extended Data Fig. 8d,h).

Extended Data Figure 8 d, Diverging bar plot showing the number of differentially expressed genes at the indicated conditions. **f**, Principal component analysis (PCA) analysis showing increasing distance between DMSO and recovery (reREC) samples following a second TSA treatment with increasing recovery time. **g**, Heatmaps showing mean expression z-Scores of recovered and not recovered genes through the TSA-recovery treatment course. **h**, Scatterplot showing correlation between transcriptomic changes induced by sequential TSA treatment following 24-hour and 48-hour recovery.

Moreover, the extent of recovery from the second TSA treatment in both cases (24hREC-24hREC, 48hREC-24hREC) was nearly identical in terms of number of differentially expressed genes (Extended Data Fig. 8d), arguing for the case that continued transcriptional changes induced by repeat exposure to TSA are stable over time. Thus, we included the following statement in lines 425-428 of the manuscript text:

“Crucially, transcriptional response and recovery was similar when the first recovery period was increased from 24- to 48-hours (Extended Data Fig. 8f-h), indicating that the cellular memory effect can persist through multiple cell generations.”

Reviewer Figure 5. a, Venn diagram showing overlap between differentially expressed gene sets between the 96-hour recovery versus 48-hour + 48-hour recovery conditions. b-c, Bubble plots showing unique Gene Ontology terms enriched among up- (b) and downregulated (c) genes.

Similarly to single exposure to TSA, transcriptional differences compared to the pre-TSA condition increased when recovery time was increased following the second exposure (Extended Data Fig. 8f). However, when we compare differentially expressed genes in the 96-hour recovery versus 48-hour + 48-hour recovery condition, we find that ~30% and ~50% of up- and downregulated genes were unique to either condition, respectively (Reviewer Fig. 5a). We then compared gene ontology terms that are uniquely enriched in either condition, and we noted that the second TSA treatment modifies the developmental trajectory of cells (Reviewer Fig. 5b). In sum, our data suggest that in response to transient challenges, cells initiate self-propagating responses that are reproducible, but might partly depend on the duration of the challenge on whether it is unique or repeated. Please note that we chose not to include these reviewer analyses in order to avoid adding unnecessary complexity that might blur, rather than sharpen, the message of the manuscript. However, readers that would like to delve into details will easily be able to perform these and other analyses thanks to the extensive raw data, extended and supplementary information provided along with the manuscript.

Reviewer#1 major comment 6: Gastruloids as a model

While I agree with the HDAC-pulse experimental design in a steady state cell type such as ESC, I think there are some fundamental issues with application to gastruloids. Specifically, a transient change in a dynamic model will change the trajectory of differentiation and thus affect proportions of cell lineages in a gastruloid. Therefore, observing increased Sox2 expression after washout does not “indicate that a memory of the TSA pulse might be ingrained.”, but more likely that a bias on lineage allocation was made at the time of exposure. Testing memory and dynamic cell states at the same time makes it difficult to uncouple one thing from the other – or understand if it is an earlier-induced change in cell trajectory rather than a molecular memory.

We agree with the reviewer on this remark and thus removed the sentence ~~“Nevertheless, the incomplete recovery of the Sox2 expression domain in gastruloids might indicate that a memory of the TSA pulse might be ingrained.”~~ from lines 346-347 of the manuscript. Additionally, we tested recovery from TSA in another steady state cell type – see detailed answer at Reviewer#1 major comment 7 below.

Reviewer#1 major comment 7: Cell type

Many studies have shown that ESC possess a unique chromatin state and response patterns. Therefore, whether the observations have relevance beyond in vitro pluripotent cells remains open. The authors attempt to address this with gastruloids, with some success in terms of similar gene sets being responsive. However, given comments above, it would be useful to discuss further the caveats to using ESC and their relevance to in vivo or developed tissues. Of course, it would be ideal to test the principle of ‘memory’ in another steady state cell type but this is may be a big task – especially if further timepoints are added to the ESC model.

We certainly agree that the unique chromatin state and extreme plasticity of pluripotent cells might give them the ability to dynamically respond to and recover from the TSA pulse. Thus, in order to understand if transcriptional memory could be induced in cells committed to a developmental lineage, we differentiated mESC into neural progenitor cells (NPCs) and subjected them to a single round of TSA-recovery cycle. First, we observed that, similarly to ESCs, TSA treatment induces hyperacetylation at H3K27ac which is then efficiently restored 24 hours after of TSA washout at the protein level (Extended Data Fig. 7f).

Extended Data Figure 7f. Western blot showing TSA-induced H3K27 hyperacetylation and recovery in nuclear extracts from NPCs. Lamin B1 is shown as loading control.

Looking at the transcriptome, we detected large-scale gene expression deregulation following a 4-hour TSA pulse in NPCs, that was very well recovered 24 hours after drug washout. Critically, like in ESCs, the transcriptional identity of TSA-treated cells shifted further from the DMSO state as recovery time was increased from 24- to 48 hours, as seen on PCA plot as well as the number of differentially expressed genes (Extended Data Fig. 7g-h).

Extended Data Figure 7. g. Principal component analysis (PCA) analysis showing increasing distance between DMSO and recovery (REC) samples with increasing recovery time. **h.** Diverging bar plot showing the number of differentially expressed genes at the indicated conditions. **i.** Bubble plots showing HOMER motif enrichment analysis at promoters of genes that recover from and genes that remain deregulated following TSA treatment, in ESCs and NPCs.

To understand if the molecular factors responsible for the TSA-induced transcriptional response are shared between the ESCs and NPCs, we performed motif analysis at deregulated TSSs that recover or do not recover in either cell types. This analysis highlighted a highly similar initial response to TSA, and more divergent transcription factors responsible for continued gene expression deregulation that can be partially attributed to cell type-specific transcription factors (e.g. Sox3) (Extended Data Fig. 7i). To confirm if sustained downregulation could be once again linked to Polycomb activity, we plotted the mean H3K27me3 ChIP-seq signal in NPCs at TSSs of the various gene groups. This revealed that H3K27me3 signal was specifically enriched around promoters of genes that show sustained downregulation following TSA removal, indicating that the Polycomb-mediated transcriptional memory applies in NPCs (Extended Data Fig. 9e).

Extended Data Figure 9e. Metaplot showing mean H3K27me3 ChIP-seq signal in NPCs at TSSs of upregulated genes that recover and do not recovery, and downregulated genes that that recover and do not recover. Shading represents standard error.

Finally, we added our main conclusions regarding the effects of the TSA pulse in NPCs in lines 334-340:

“To reinforce the relevance of our findings beyond pluripotent cells, we tested the ability of neural progenitor cells (NPCs) to recover from a TSA pulse. This revealed that, similarly or ESCs, transcriptional differences are minimal after 24 hours of recovery from TSA and increase at 48 hours (Extended Data Fig. 7f-h). Additionally, while transcriptional response to TSA is seems to be partially linked to similar regulatory pathways in ESCs and NPCs, cell type specific-difference exist (Extended Data Fig. 7i).”

as well as in lines 442-444:

“The same was found to be true in NPCs (Extended Data Fig. 9d), indicating that sustained gene downregulation might be linked to Polycomb activity in multiple cell types.”

Reviewer#1 major comment 8: Perturbation controls

It is surprising that only a single HDAC inhibitor was deployed. In order to begin to rule out off-target or indirect effects additional orthogonal perturbations (or genetic approaches such as degrons) could be used. This would add considerable strength to the conclusions if the effects were comparable.

We agree with the reviewer that effects of an orthogonal perturbation are critical to test. Class I HDACs comprise the main histone modifying enzymes^{18,19}, and have a high degree of functional redundancy but also regulate different aspects of ESC maintenance and differentiation²⁰⁻²³. Thus, a pervasive chromatin perturbation – like that induced by TSA – would require the conditional depletion of several HDACs, making cell line construction extremely challenging. Therefore, we opted for the use of an alternative pharmacological inhibitor, romidepsin, that exhibits high specificity against HDACs with histone targets (HDAC1, 2, 3 and 8)¹⁸ (Extended Data Fig. 3a). We treated ESCs with romidepsin and assayed them for H3K27 acetylation (by Western blotting), transcriptional changes and changes in chromatin architecture.

We found that romidepsin-induced hyperacetylation resulted in a transcriptional response extremely similar to that induced by TSA. This was evident from 1) the distance matrix plot revealing that the TSA and Romidepsin conditions cluster together, 2) the scatter plot showing very high correlation between \log_2 fold changes in gene expression and 3) the Gene Ontology enrichment plot demonstrating the amplification of similar biological processes among up- and downregulated genes (Extended Data Fig. 3b-e).

Extended Data Figure 3 – Comparison of the effects of TSA and Romidepsin on gene expression and genome architecture in mESCs. **a**, HDACs inhibited by TSA and Romidepsin. Data from Ho et al. 2020. **b**, Western blot showing Romidepsin-induced (Rom) H3K27 hyperacetylation and recovery in nuclear extracts. Lamin B1 is shown as loading control. **c**, Sample distance matrix based on RNA-seq data showing high similarity of transcriptional response to TSA vs Romidepsin. **d**, Scatterplot showing correlation between transcriptomic changes induced by TSA and Romidepsin. Shading represents \log_2 fold change in TSA. **e**, Gene Ontology enrichment among up- and downregulated genes in TSA and Romidepsin. **f**, Ratio of cis versus trans contacts in DMSO, TSA and Romidepsin Micro-C datasets. **g**, Saddle plots of compartment interactions in cis. **h**, Aggregate plots of Micro-C signal in DMSO, TSA and Romidepsin at non-CTCF Polycomb loops (top panels) and at de novo H3K9me3 loops (bottom panels) (resolution = 4 kb).

Analysis of the changes in 3D genome conformation also confirmed TSA-induced effects: increase in *trans* interactions, strengthened BB and mitigated AA compartmenting interactions, as well as bolstered chromatin looping at repressive chromatin sites (H3K27me3, H3K9me3) were all prominent features of Romidepsin Micro-C maps (Extended Data Fig. 3f-h). Thus, we conclude that the pan-HDAC inhibitor TSA, and the selective Class I inhibitor Romidepsin induce similar changes in transcription and genome architecture in ESCs. Accordingly, we added this new result to lines 170-174 in the manuscript text:

“To rule out potential off-target effects of the non-selective HDAC inhibitor TSA, we assayed changes in gene expression and chromatin architecture changes induced by romidepsin, a specific inhibitor of nuclear HDACs¹⁸ (Extended Data Fig, 3a). This revealed a nearly identical transcriptional response as well as similar changes in 3D genome folding between the two inhibitors (Extended Data Fig. 3b-h).”

Reviewer#1 major comment 9: Mechanisms

Ultimately, it is concluded that looping plays a central role in the memory process, which is largely based on associations. Given the pleiotropy and incomplete recovery, it is necessary to understand if memory is really encoded in looping. It would be ideal to artificially stimulate looping to test its ability to confer memory, for example by using a dCas9 to target activation or repression to candidate sites or via tethering assays. Such targeted manipulations would provide much more confidence in the conclusions.

We thank the reviewer for this insightful comment. We fully agree that the possible role of looping as a mechanism for memory should be addressed experimentally. As our TSA treatments leads to differential gene expression and differential chromatin looping at thousands of sites – some of which are linked to memory and some of which are not - taking a locus-specific approach would be difficult, in particular in terms of which locus one would choose in order to identify representative effects. Therefore, we decided to test - specifically for genes that appear to show Polycomb-dependent continued gene expression downregulation - the necessity of Polycomb-dependent looping for the transcriptional memory effects. This appeared especially relevant after having found that TSA induced increased Ring1B chromatin occupancy that persisted through the recovery period, without continued increase in H3K27me3 and H2AK119Ub levels (Fig. 7a) (see also at Reviewer#1 major comment 2 point (iv)).

Figure 7a – Boxplots (top) and heatmaps (bottom) of normalised Ring1B, H3K27me3 and H2AK119Ub ChIP-signal in DMSO, TSA and 24-hour recovery (REC) at genomic sites with increased Ring1B binding in Recovery condition.

It has been previously shown that Polycomb domain interactions are provided by canonical PRC1 (cPRC1) complexes, and they specifically depend on the presence of the PCGF2 and PCGF4 subunits^{12,24}. Accordingly, Polycomb-mediated chromatin looping can be disrupted in an ES cell line where PCGF4 is mutated and PCGF2 can be deleted by tamoxifen (OHT) treatment (PCGF4^{-/-} PCGF2^{fl/fl}). Importantly, it has been shown that this leaves the enzymatic activities of PRC1 and PRC2 intact^{12,24} (Extended Data Fig. 10c-d), separating the repressive properties of the histone modification landscape from that of chromatin topology.

First, we tested that OHT treatment indeed disrupted Polycomb-mediated chromatin looping genome wide in the PCGF4^{-/-} PCGF2^{fl/fl} cell line. As evident from the aggregate plot analysis of our Micro-C maps, OHT treatment indeed led to the complete elimination of Polycomb

loops, especially at CTCF-independent sites (Extended Data Fig. 10a-b). Next, as our TSA-recovery double treatment time course required prolonged (120 hours) PCGF2 depletion, we tested gene expression changes that were induced during this time period. According to previous findings that chromatin topology contributes to the repressive function of Polycomb complexes^{11,25–29}, we observed mild, progressively increasing derepression of Polycomb target genes with increasing PCGF2 depletion times. Nevertheless, the transcriptional response to TSA was similar between 1) wild type E14 ESCs, 2) untreated (-OHT) PCGF4^{-/-} PCGF2^{fl/fl} ESCs and 3) tamoxifen-treated (+OHT), PCGF2-depleted PCGF4^{-/-} PCGF2^{fl/fl} ESCs (Extended Data Fig. 10c-f).

Extended Data Figure 10 – Changes in the transcriptome and genome conformation upon PCGF2 depletion. a-b, Aggregate plots of Micro-C signal in at non-CTCF (a) and CTCF (b) H3K27me3 loops (top panels) and at non-CTCF H2AK119Ub loops (bottom panels) (resolution = 10 kb) in *Pcgf2*^{fl/fl} cells. c, Differential ChIP-seq peaks identified by DESeq2 between different treatment conditions in the *Pcgf4*^{-/-}; *Pcgf2*^{fl/fl} mESC cell line. d, Genomic snapshot of spike-in normalised H3K27me3 and H2AK119Ub signal over the *HoxA* cluster. e, Diverging bar plot showing the number of differentially expressed genes upon prolonged tamoxifen (OHT) treatment in a *Pcgf4*^{-/-}; *Pcgf2*^{fl/fl} mESC cell line. f, Gene Ontology enrichment among deregulated genes following prolonged OHT treatment. g, Scatterplot showing correlation between transcriptomic changes induced by TSA in the presence or absence of OHT treatment in *Pcgf2*^{fl/fl} cells. Shading represents log₂ fold change upon TSA treatment

in wild type E14 mESCs. **h**, Gene Ontology enrichment among differentially expressed genes in the first and second TSA treatments, in E14, *Pcgf2^{fl/fl}* -OHT and *Pcgf2^{fl/fl}* +OHT cells.

Next, we looked at the ability of gene expression to recover from the subsequent TSA pulses, in the -OHT versus the +OHT conditions. We noticed that in +OHT cells, the number of genes that remained deregulated following the recovery periods was decreased by more than two-fold (Fig. 7c). This suggests that the absence of Polycomb domain interactions weakens the transcriptional memory response induced by the TSA pulses. We then assessed the behaviour of genes that get irreversibly downregulated only in the -OHT condition (321 genes out of 366). According to our previous observations, this gene set shows progressively worsening gene expression deregulation throughout the time course (Fig 7d). Crucially however, the same gene set shows steady expression levels in the +OHT conditions, indicating that the disruption of Polycomb domain interactions eliminated the transcriptional memory response.

Figure 7c. Diverging bar plot showing the number of differentially expressed genes during the TSA-recovery treatment course in the *Pcgf4^{-/-}; Pcgf2^{fl/fl}* cell line with (+OHT) or without (-OHT) *Pcgf2* depletion. **d**, Gene expression Z-Scores of recovered and not recovered genes through the TSA-recovery treatment course in -OHT (left panel) and +OHT (right panel) conditions. Control condition for -OHT and +OHT are DMSO and DMSO +120h OHT treatment, respectively. Data shown are the median, with hinges corresponding to IQR and whiskers extending to the lowest and highest values within 1.5× IQR. Fitted line represents smoothed mean, shading corresponds to standard error.

We nevertheless identify a smaller gene set that shows transcriptional memory behaviour only in the +OHT condition (152 genes out of 197) (Fig. 7c-d), so thus we sought to understand what chromatin features distinguish the two gene groups that may give rise to their distinct behaviour. First, by looking at H3K27me3, H2K119Ub and Ring1B occupancy at TSSs, we concluded that genes that exhibited sustained downregulation only in +OHT cells (Down nonREC +OHT) were equally Polycomb targets (Fig. 7e) – although we note a slightly lower PRC1 occupancy at these sites compared to Down nonREC -OHT TSSs. Next, as not all Polycomb loci show the same ability to form chromatin loops, we examined the behaviour of loops emanating from TSSs of the two gene groups (Fig. 7f-g). To start with, we noted a decreased proportion of irreversibly downregulated TSSs to make focal chromatin contacts within their 5kb vicinity in +OHT versus -OHT conditions (Extended Data Fig. 9b).

Figure 7e, Metaplot showing mean H3K27me3, H2AK119Ub and Ring1B ChIP-seq signal in wild type E14 mESCs at TSSs of downregulated genes that recover and do not recover in -OHT and +OHT conditions. Shading represents standard error.

b Looping propensity of differentially expressed TSSs

	TSSs	Loops	Ratio	z-score	p-value
Up REC	5365	2477	0.462	-4.1588	<10 ⁻⁵
Up nonREC	752	409	0.542		
Down REC	3266	1722	0.527	-10.1322	<10 ⁻⁵
Down nonREC	275	232	0.844		
Down nonREC -OHT	401	369	0.920	10.9252	<10 ⁻⁵
Down nonREC +OHT	113	209	0.544		

Extended Data Figure 9b, Number of DMSO loops within 5 kb vicinity of different TSS groups. Z-scores and p-values were calculated using two-tailed two-proportions z-test.

Moreover, while chromatin loops at -OHT TSSs increased in strength in TSA as well as in recovery, we did not observe such enhanced looping at +OHT TSSs (Fig. 7f-g). Finally, as looping depends on the presence of cPRC1 complexes on chromatin, we compared the levels of Ring1B at loop anchors. This indicated that while -OHT TSSs underwent increased Ring1B binding in TSA and recovery, +OHT TSSs did not (Fig. 7h). These observations suggest that the transcriptional memory response exhibited by +OHT cells relies on a mechanism different from chromatin looping. Altogether, these results provide unequivocal evidence that disruption of PRC1-mediated chromatin looping rewrites the transcriptional memory response induced by TSA, establishing chromatin topology as a novel molecular mechanism for cellular memory.

Figure 7 – Disruption of PRC1-mediated 3D contacts changes memory response triggered by TSA treatment. *f*, Aggregate plots Micro-C signal in DMSO, TSA and Recovery around loops where anchors overlap with downregulated TSSs that do not recover in -OHT (top panels) or +OHT (bottom panels) conditions (resolution = 4 kb). *g*, Quantification of piled-up loop strength shown in *f* (paired two-tailed t-test; ns > 0.05, **p < 0.01). Data shown are the median, with hinges corresponding to IQR and whiskers 436 extending to the lowest and highest values within 1.5× IQR. *h*, Boxplot showing normalised Ring1B ChIP-seq signal at TSS loop anchors in DMSO, TSA and Recovery, around genes that do not recover in -OHT (left) and +OHT (right) conditions.

These critical results are now described under the “**PRC1-mediated spatial clustering contributes to continued gene expression downregulation**” subheading of the manuscript, in lines 479-513:

“PRC1-mediated spatial clustering contributes to continued gene expression downregulation

As sustained transcriptional downregulation of Polycomb target genes happened without the accumulation of the Polycomb Repressive Complex 2 (PRC2) mark H3K27me3, we sought to understand whether the mechanistic basis of memory relies on the activity of the Polycomb Repressive Complex 1 (PRC1). We performed calibrated ChIP-seq against Ring1B which indeed revealed increased binding of PRC1 at >8,000 sites in the recovery condition (Fig. 7a). Neither H3K27me3 nor H2AK119Ub showed sustained increase at these sites, consistent with our previous finding that the Polycomb histone modification landscape remains largely unchanged. To test whether transcriptional memory relied on PRC1-mediated spatial clustering, we disrupted Polycomb-mediated looping via the depletion of two subunits of a subclass of PRC1 complexes called canonical PRC1 (cPRC1), namely PCGF2 and PCGF4 (Fig. 7b). We have done so by using an ESC line (which we call PCGF2^{fl/fl}) in which PCGF4 is mutated and PCGF2 can be deleted by tamoxifen (OHT) treatment. This line was previously used to demonstrate that PCGF2 and PCGF4 are responsible for creating interactions between Polycomb domains^{35,36} (Extended Data Fig. 10a-b) and that this architectural role can be uncoupled from the Polycomb histone modification landscape (Extended Data Fig. 10c-d). Although sustained PCGF2 depletion led to mild derepression of Polycomb targets, the transcriptional response to the TSA-recovery double treatment course was highly similar between E14, PCGF2^{fl/fl} -OHT and PCGF2^{fl/fl} +OHT cells (Extended Data Fig. 10c-h). PCGF2 depletion, however, decreased the number of genes that did not recover from the TSA pulse by more than two-fold (Fig. 7c), indicating a weaker memory response. Critically, most genes (321 out of 366) that showed sustained downregulation in the -OHT condition, did not exhibit transcriptional memory in the +OHT condition (Fig. 7d), indicating that the disruption of PcG loops rewrites the memory response to TSA. We nevertheless identified a smaller subset of genes that showed sustained downregulation in +OHT cells that were also Polycomb targets (Fig. 7d-e). However, while the gene set that remained downregulated in only -OHT cells showed increased looping at TSSs and increased Ring1B occupancy at TSS loops anchors, the non-recovery genes unique to +OHT cells (152 genes out of 197) did not (Extended Data Fig. 9b, Fig. 7f-h), indicating that their downregulation relies on a mechanism that is independent from PRC1-mediated chromatin looping.

In sum, we find that disruption of Polycomb domain interactions modulates the memory response, demonstrating that PRC1-mediated spatial clustering is responsible for the TSA-induced transcriptional memory at Polycomb target genes.”

Minor comments

I particularly appreciated the employment of biophysical modelling. In my view, mathematical models in Biology are useful when they allow to understand complex phenomena in a simplified manner, and when they produce predictions that can be tested and confirmed. In your model, as mentioned in line 155, there is the prediction of a physical migration of A-domains toward the periphery of the nucleus. This can be tested by microscopy, using DNA or RNA-FISH (the latter for an intron of a transcribed gene), for loci known to be part of A-compartments. At present the model largely reinforces experimental observations rather than providing new testable insights.

We thank the reviewer for their appreciation of our biophysical modelling on the TSA-induced changes in genome folding. Indeed, the model predicted a displacement of A chromatin towards the nuclear periphery which we now tested experimentally. As the bulk of acetylation occurs in A compartments (Fig. 1b, Fig 2b-c), we deemed it was sufficient to stain ESCs for H3K27ac signal and quantify the distance of H3K27ac foci from the nuclear periphery. This showed that H2K27ac foci were more peripheral in TSA-treated cells than control cells (DMSO), and that this decreased distance occurred despite of a mild increase in nuclear volume in the TSA condition (Extended Data Fig. 2q-r).

Extended Data Figure 2 - q, Violin plots showing nuclear volume (left panel) and mean distance of H3K27ac spots to periphery (right panel) in DMSO ($n = 154$) and TSA nuclei ($n = 169$). Combined result of two biological replicates is shown. **r**, Representative images of H3K27ac immunofluorescence in DMSO and TSA nuclei.

The extent of displacement towards the periphery measured in cells was comparable to that observed in further simulations (~ 200 nm). Importantly, this did not change when the 5% increase in nuclear volume observed in TSA was taken into account in simulations, indicating that the change in nuclear radius has no major role in this process (Extended Data Fig. 2p).

Extended Data Figure 2p, Distance to the periphery for all particles in the A-compartment during the last quarter of the trajectory (between 3 and 4 hours). Plotted values are average per replicate ($n = 5$). TSA nucleus represents larger nuclei of 49.04 sigma (2657.81 nm radius) to take into account the 5% increase in nuclear volume observed in TSA.

These new results are now added to lines 159-161 in the manuscript text:

“This was confirmed by H3K27ac immunofluorescence which indicated increased proximity of H3K27ac foci to the nuclear periphery in TSA-treated despite a mild increase in nuclear volume (Extended Data Fig. 2q-r).”

210: A possible typo; “at” least?

We apologise for this mistake, it is now corrected in the manuscript text.

Related to 304-308: It is not clear to me what is the extent of the TSA perturbation memorised in gastruloids. In Fig.E5e there is a Volcano plot showing some dysregulated genes (please add n for up- and down-regulated genes). I don’t see among them Sox2, despite the authors show by staining an extension of the Sox2-marked neuroectodermal region (Fig.4h). Why is Sox2 not shown as upregulated in Fig.E5e?

We thank the reviewer for this insightful comment. Indeed, Sox2 is not present among upregulated genes according to the RNA-seq results. This could be due to multiple reasons. First, it could be due to discrepancy between RNA and protein levels: an earlier transcriptional upregulation could result in prolonged presence of increased amount of protein. Second, as transcriptional variability between our gastruloid differentiations was relatively high, it is possible that this degree of increase is not detectable by bulk RNA-seq technique and would require a single-cell transcriptomics approach. We suspect that the very small number of differentially expressed genes (17 in total) could be due to the same reason. We now indicated these numbers on the revised Extended Data Fig. 7e.

Reviewer #2 (Remarks to the Author):

In this study the authors aim to establish a link between 3D genome organization -a key regulator of genome function- and cellular memory. Specifically, they investigate whether transient changes in histone acetylation can induce durable and heritable alterations in 3D genome organization and transcriptional program in mouse embryonic stem cells (mESCs). Using a combination of ChIP-seq, ATAC-seq, RNA-seq, and ultra-deep Micro-C, the authors demonstrate that an acute (transient) increase in histone acetylation can trigger changes in genome folding and gene expression that can be retained, even in the absence of maintaining histone acetylation state. They show that histone deacetylase (HDAC) inhibition by TSA leads to modifications in histone acetylation (H3K27ac) and methylation (H3K4me1/3, H3K27me3, H3K9me3) and alterations in both global and local genome 3D folding.

Upon TSA washout, the epigenetic state (and particularly histone modification H3K27ac) rapidly reverts, although 3D conformational changes and deregulated gene expression persist as cells divide. This “memory” becomes more pronounced upon a second exposure to TSA.

Overall, this is an extensive and well-described study. It provides evidence of a TSA-induced cellular memory of gene transcription and 3D genome that persists after transient epigenetic change. Although potentially interesting, there are two major issues that need to be addressed/resolved.

We are grateful to the reviewer for their support and their recognition of our study and for raising insightful points that gave us the chance to perform crucial experiments. We hope that they will find our responses convincing.

1. Firstly, the mechanism by which cellular memory operates in response to TSA treatment is unclear and is currently not sufficiently addressed. For example, it is not yet clear whether the 3D structure itself plays a direct role in conveying epigenetic memory, or whether this is achieved by other factors targeted by TSA. While the enhanced Micro-C technique used in this study offers a powerful tool to investigate the role of 3D genome interactions in regulating gene expression and transcriptional memory, the results remain correlative rather than demonstrating a clear causality.

We agree with the reviewer that this was a critical missing point of the study which we now addressed through a complex series of experiments. Specifically, we decided to test the necessity of Polycomb-dependent looping for the transcriptional memory effects at genes that show continued gene expression downregulation. As this concern was equally raised by Reviewer#1, we duplicated our Reviewer#1 major comment 9 response below.

Polycomb domain interactions are not necessarily dependent on the catalytic activity of PRC1 and PRC2 complexes, rather, they seem to be linked to the physical presence of canonical PRC1 complexes on chromatin^{11,12}. Thus, we hypothesised that TSA treatment might induce increased PRC1 binding to chromatin which in turn could lead to increased Polycomb domain interactions. To this end, we performed calibrated ChIP-seq against the Ring1B subunit of PRC1, followed by quantitative identification of differential binding sites. Interestingly, we indeed found increased Ring1B occupancy at several thousands of sites in TSA (Reviewer Fig. 1a-b), implying that binding of PRC1 complexes to chromatin – rather than H3K27me3/H2AK119Ub – may be responsible for increased looping.

Reviewer Figure 1. *a*, Heatmaps showing scaled Ring1B ChIP-seq signal per epitope in merged replicates at all enrichment sites in DMSO, TSA and REC conditions. *b*, Diverging bar charts showing the number of differential Ring1B ChIP-seq peaks identified by DESeq2 analysis in TSA vs DMSO (TSA) and 24-hour recovery vs DMSO (REC) conditions.

Indeed, increased Ring1B chromatin occupancy that persisted through the recovery period occurred without continued increase in H3K27me3 and H2AK119Ub levels (Fig. 7a), consistent with our previous finding that the Polycomb histone modification landscape remains largely unchanged.

Figure 7a – Boxplots (top) and heatmaps (bottom) of normalised Ring1B, H3K27me3 and H2AK119Ub ChIP-signal in DMSO, TSA and 24-hour recovery (REC) at genomic sites with increased Ring1B binding in Recovery condition.

It has been previously shown that Polycomb domain interactions are provided by canonical PRC1 (cPRC1) complexes, and they specifically depend on the presence of the PCGF2 and PCGF4 subunits^{12,24}. Accordingly, Polycomb-mediated chromatin looping can be disrupted in an ES cell line where PCGF4 is mutated and PCGF2 can be deleted by tamoxifen (OHT) treatment (PCGF4^{-/-} PCGF2^{fl/fl}). Importantly, it has been shown that this leaves the enzymatic activities of PRC1 and PRC2 intact^{12,24} (Extended Data Fig. 10c-d), separating the repressive properties of the histone modification landscape from that of chromatin topology.

Extended Data Figure 10 – Changes in the transcriptome and genome conformation upon PCGF2 depletion. **a-b**, Aggregate plots of Micro-C signal in at non-CTCF (**a**) and CTCF (**b**) H3K27me3 loops (top panels) and at non-CTCF H2AK119Ub loops (bottom panels) (resolution = 10 kb) in *Pcgf2^{fl/fl}* cells. **c**, Differential ChIP-seq peaks identified by DESeq2 between different treatment conditions in the *Pcgf4^{-/-}; Pcgf2^{fl/fl}* mESC cell line. **d**, Genomic snapshot of spike-in normalised H3K27me3 and H2AK119Ub signal over the *HoxA* cluster. **e**, Diverging bar plot showing the number of differentially expressed genes upon prolonged tamoxifen (OHT) treatment in a *Pcgf4^{-/-}; Pcgf2^{fl/fl}* mESC cell line. **f**, Gene Ontology enrichment among deregulated genes following prolonged OHT treatment. **g**, Scatterplot showing correlation between transcriptomic changes induced by TSA in the presence or absence of OHT treatment in *Pcgf2^{fl/fl}* cells. Shading represents \log_2 fold change upon TSA treatment in wild type E14 mESCs. **h**, Gene Ontology enrichment among differentially expressed genes in the first and second TSA treatments, in E14, *Pcgf2^{fl/fl}* -OHT and *Pcgf2^{fl/fl}* +OHT cells.

First, we tested that OHT treatment indeed disrupted Polycomb-mediated chromatin looping genome wide in the *PCGF4^{-/-} PCGF2^{fl/fl}* cell line. As evident from the aggregate plot analysis of our Micro-C maps, OHT treatment indeed led to the complete elimination of Polycomb loops, especially at CTCF-independent sites (Extended Data Fig. 10a-b). Next, as our TSA-recovery double treatment time course required prolonged (120 hours) PCGF2 depletion, we tested gene expression changes that were induced during this time period. According to previous findings that chromatin topology contributes to the repressive function of Polycomb complexes^{11,25–29}, we observed mild, progressively increasing derepression of Polycomb

target genes with increasing PGC2 depletion times. Nevertheless, transcriptional response to TSA was similar between 1) wild type E14 ESCs, 2) untreated (-OHT) PCGF4^{-/-} PCGF2^{fl/fl} ESCs and 3) tamoxifen-treated (+OHT), PCGF2-depleted PCGF4^{-/-} PCGF2^{fl/fl} ESCs (Extended Data Fig. 10c-f).

Next, we looked at the ability of gene expression to recover from the subsequent TSA pulses, in the -OHT versus the +OHT conditions. We noticed that in +OHT cells, the number of genes that remained deregulated following the recovery periods was decreased by more than two-fold (Fig. 7c). This suggests that the absence of Polycomb domain interactions weakens the transcriptional memory response induced by the TSA pulses. We then assessed the behaviour of genes that get irreversibly downregulated only in the -OHT condition (321 genes out of 366). According to our previous observations, this gene set shows progressively worsening gene expression deregulation throughout the time course (Fig 7d). Crucially however, the same gene set shows steady expression levels in the +OHT conditions, indicating that the disruption of Polycomb domain interactions eliminated the transcriptional memory response.

Figure 7c. Diverging bar plot showing the number of differentially expressed genes during the TSA-recovery treatment course in the *Pcgf4*^{-/-}; *Pcgf2*^{fl/fl} cell line with (+OHT) or without (-OHT) *Pcgf2* depletion. **d.** Gene expression Z-Scores of recovered and not recovered genes through the TSA-recovery treatment course in -OHT (left panel) and +OHT (right panel) conditions. Control condition for -OHT and +OHT are DMSO and DMSO +120h OHT treatment, respectively. Data shown are the median, with hinges corresponding to IQR and whiskers extending to the lowest and highest values within 1.5× IQR. Fitted line represents smoothed mean, shading corresponds to standard error.

We nevertheless identify a smaller gene set that shows transcriptional memory behaviour only in the +OHT condition (152 genes out of 197) (Fig. 7c-d), so thus we sought to understand what chromatin features distinguish the two gene groups that may give rise to their distinct behaviour. First, by looking at H3K27me3, H2K119Ub and Ring1B occupancy at TSSs, we concluded that genes that exhibited sustained downregulation only in +OHT cells (Down nonREC +OHT) were equally Polycomb targets (Fig. 7e) – although we note a slightly lower PRC1 occupancy at these sites compared to Down nonREC -OHT TSSs. Next, as not all Polycomb loci show the same ability to form chromatin loops, we examined the behaviour of loops emanating from TSSs of the two gene groups (Fig. 7f-g). To start with, we noted a decreased proportion of irreversibly downregulated TSSs to make focal chromatin contacts within their 5kb vicinity in +OHT versus -OHT conditions (Extended Data Fig. 9b).

Figure 7e, Metaplot showing mean H3K27me3, H2AK119Ub and Ring1B ChIP-seq signal in wild type E14 mESCs at TSSs of downregulated genes that recover and do not recover in -OHT and +OHT conditions. Shading represents standard error.

b Looping propensity of differentially expressed TSSs

	TSSs	Loops	Ratio	z-score	p-value
Up REC	5365	2477	0.462	-4.1588	<10 ⁻⁵
Up nonREC	752	409	0.542		
Down REC	3266	1722	0.527	-10.1322	<10 ⁻⁵
Down nonREC	275	232	0.844		
Down nonREC -OHT	401	369	0.920	10.9252	<10 ⁻⁵
Down nonREC +OHT	113	209	0.544		

Extended Data Figure 9b, Number of DMSO loops within 5 kb vicinity of different TSS groups. Z-scores and p-values were calculated using two-tailed two-proportions z-test.

Moreover, while chromatin loops at -OHT TSSs increased in strength in TSA as well as in recovery, we did not observe such enhanced looping at +OHT TSSs. Finally, as looping depends on the presence of cPRC1 complexes on chromatin, we compared the levels of Ring1B at loop anchors. This indicated that while -OHT TSSs underwent increased Ring1B binding in TSA and recovery, +OHT TSSs did not (Fig. 7h). These observations suggest that the transcriptional memory response exhibited by +OHT cells relies on a mechanism different from chromatin looping. Altogether, these results provide unequivocal evidence that disruption of PRC1-mediated chromatin looping rewrites the transcriptional memory response induced by TSA, establishing chromatin topology as a novel molecular mechanism for cellular memory.

Figure 7 – Disruption of PRC1-mediated 3D contacts changes memory response triggered by TSA treatment. *f*, Aggregate plots Micro-C signal in DMSO, TSA and Recovery around loops where anchors overlap with downregulated TSSs that do not recover in -OHT (top panels) or +OHT (bottom panels) conditions (resolution = 4 kb). *g*, Quantification of piled-up loop strength shown in *f* (paired two-tailed *t*-test; ns > 0.05, ***p* < 0.01). Data shown are the median, with hinges corresponding to IQR and whiskers 436 extending to the lowest and highest values within 1.5× IQR. *h*, Boxplot showing normalised Ring1B ChIP-seq signal at TSS loop anchors in DMSO, TSA and Recovery, around genes that do not recover in -OHT (left) and +OHT (right) conditions.

These critical results are now described under the “**PRC1-mediated spatial clustering contributes to continued gene expression downregulation**” subheading of the manuscript, in lines 479-513:

“PRC1-mediated spatial clustering contributes to continued gene expression downregulation

As sustained transcriptional downregulation of Polycomb target genes happened without the accumulation of the Polycomb Repressive Complex 2 (PRC2) mark H3K27me3, we sought to understand whether the mechanistic basis of memory relies on the activity of the Polycomb Repressive Complex 1 (PRC1). We performed calibrated ChIP-seq against Ring1B which indeed revealed increased binding of PRC1 at >8,000 sites in the recovery condition (Fig. 7a). Neither H3K27me3 nor H2AK119Ub showed sustained increase at these sites, consistent with our previous finding that the Polycomb histone modification landscape remains largely unchanged. To test whether transcriptional memory relied on PRC1-mediated spatial clustering, we disrupted Polycomb-mediated looping via the depletion of two subunits of a subclass of PRC1 complexes called canonical PRC1 (cPRC1), namely PCGF2 and PCGF4 (Fig. 7b). We have done so by using an ESC line (which we call PCGF2^{fl/fl}) in which PCGF4 is mutated and PCGF2 can be deleted by tamoxifen (OHT) treatment. This line was previously used to demonstrate that PCGF2 and PCGF4 are responsible for creating interactions between Polycomb domains^{35,36} (Extended Data Fig. 10a-b) and that this architectural role can be uncoupled from the Polycomb histone modification landscape (Extended Data Fig. 10c-d). Although sustained PCGF2 depletion led to mild derepression of Polycomb targets, the transcriptional response to the TSA-recovery double treatment course was highly similar between E14, PCGF2^{fl/fl}-OHT and PCGF2^{fl/fl}+OHT cells (Extended Data Fig. 10c-h). PCGF2 depletion, however, decreased the number of genes that did not recover from the TSA pulse by more than two-fold (Fig. 7c), indicating a weaker memory response. Critically, most genes

(321 out of 366) that showed sustained downregulation in the -OHT condition, did not exhibit transcriptional memory in the +OHT condition (Fig. 7d), indicating that the disruption of PcG loops rewrites the memory response to TSA. We nevertheless identified a smaller subset of genes that showed sustained downregulation in +OHT cells that were also Polycomb targets (Fig. 7d-e). However, while the gene set that remained downregulated in only -OHT cells showed increased looping at TSSs and increased Ring1B occupancy at TSS loops anchors, the non-recovery genes unique to +OHT cells (152 genes out of 197) did not (Extended Data Fig. 9b, Fig. 7f-h), indicating that their downregulation relies on a mechanism that is independent from PRC1-mediated chromatin looping.

In sum, we find that disruption of Polycomb domain interactions modulates the memory response, demonstrating that PRC1-mediated spatial clustering is responsible for the TSA-induced transcriptional memory at Polycomb target genes.”

2. Secondly, to claim an effect on cellular memory (as is indicated in the manuscript title) the authors need clonal assessments of TSA impact, rather than studying the effects of TSA on heterogenous cell cultures. This would allow them to show that single and repeat TSA treatments change the epigenetic status of cells (at the level of individual ESCs), rather than simply reading out changes within a population.

We thank the reviewer for raising this critical point which allows us to discuss some important definitions concerning epigenetics and cellular memory. Alternative gene activity states that make up cellular identity must be established and maintained, which are two distinct - but inherently interlinked - processes. At the heart of this lie epigenetic phenomena that maintain or reprogram cell fates, integrate environmental stimuli to genome function and propagate non-genetic information to the next generation of cells or organisms. However, definitions for epigenetics have been put forward by many since the term was coined in the 1940s and thus it carries different meaning to different people. In this study, we refer to epigenetics as “the study of molecules and mechanisms that can perpetuate alternative gene activity states in the context of the same DNA sequence” as defined by Cavalli & Heard (2019)³⁰. When it comes to the term cellular memory, consensus definitions are even more sparse, and further complicated by the interchangeable use of other terms such as biological memory, epigenetic memory or transcriptional memory. Here, we use the term cellular memory as “sustained cellular response to a transient stimulus”^{31,32} as it remains inclusive for the underlying mechanisms and the time scale at which the phenomenon occurs. Ultimately, population-based measurements do not allow us to assess the responsiveness of individual cells to TSA treatment - which may eventually dilute the memory effect we observe – we nevertheless believe that they are suitable for measuring cellular memory as defined above. In particular, a major worry might arise in case the cell population response during recovery after the TSA pulse was slow and heterogeneous, particularly in terms of recovery of histone acetylation levels, or of putative cell division and cell cycle changes. We therefore decided to test this directly with analysis of histone acetylation time-course and of cell tracer experiments that we describe and discuss in the Reviewer#2 major comment 5 and 6 paragraphs. In short, these experiments clearly show that the cells quickly recover the initial histone acetylation levels, that they do not show any overt induction of heterogeneity, and that they all divide during the recovery from the TSA pulse. Although we agree with the reviewer that demonstrating clonal inheritance of chromatin states would provide further support to TSA-induced memory, we believe that the experiments provided here lend support to the interpretation that the cell

population globally maintains the memory of the TSA stimulus and that such clonal investigation would extend beyond the scope of this study (see further technical reasons discussed at Reviewer#2 major comment 5). Therefore, we have not directly addressed this point experimentally by performing clonal analysis.

We apologise to the reviewer that our definition and interpretation of cellular memory was not clear from the initial version of the manuscript. Therefore, we added our definition of cellular memory in lines 44-45 of the manuscript text:

“An important feature of functional chromatin states is the ability to convert short-lived signals to long-lived changes in gene expression – a concept commonly referred to as cellular memory³³, *that is, a sustained cellular response to a transient stimulus.*”

Specific Points:

Reviewer#2 major comment 1 – Other primary changes in histone acetylation

First, the authors demonstrate that TSA, a broad HDAC inhibitor, induces a global increase in H3K27ac, primarily at cis-regulatory elements along gene bodies. TSA also has secondary effects on histone modifications, including a loss of repressive marks (PRC1/2-mediated histone modification) and a gain of active marks (such as H3K4me1), suggesting increased chromatin relaxation and transcriptional activity. It would be important to understand whether other primary changes in histone acetylation, such as H3K9/14ac or H4K5/16ac, could also occur under TSA treatment at global or at specific loci of deregulated genes. How do the authors rule out increased acetylation of non-histone proteins?

First point: *Other primary changes in histone acetylation.* We thank the reviewer for this excellent suggestion. In order to assess global acetylation levels – both in the nucleus as well as in the cytoplasm – we performed pan-acetylation Western blots on the two cellular fractions (Extended Data Fig. 1a-b). This demonstrated the vast majority of TSA-induced acetylation is gained on in the nuclear fraction, especially on histone H3 (~20 kDa band) and H4 (~14 kDa band). In the cytoplasmic fraction lysine acetylation is relatively stable, except for tubulin which gets highly acetylated upon TSA treatment. Crucially, all hyperacetylation visible on the below Western blots is efficiently reversed both in the first and second recovery.

Extended Data Figure 1a. Western blots showing the levels of lysine acetylation in nuclear extracts in the following conditions: DMSO control, TSA treatment, 24-hour recovery following TSA washout, sequential TSA treatment and 24-hour recovery from the second TSA treatment. Lamin B1 is shown as loading control. **b.** Western blots showing the levels of lysine acetylation (top panels) and levels of acetylated tubulin (bottom panels) in cytoplasmic protein extracts in conditions as in **a**. Ponceau staining is shown as loading control on the right.

Although we assessed in extensive detail the genome-wide behaviour of H3K27 acetylation, we agree with the reviewer that confirming TSA-induced hyperacetylation on other histone residues is important. Therefore, we performed Western blot and calibrated ChIP-seq against H3K9ac in DMSO, TSA and REC condition (Extended Data Fig. 5h-j).

Extended Data Fig. 5h. Western blot showing TSA-induced H3K9 hyperacetylation and recovery in nuclear extracts. Lamin B1 is shown as loading control. **i.** Heatmaps showing scaled H3K9ac ChIP-seq signal in merged replicates at all enrichment sites in DMSO, TSA and REC conditions. **j.** Diverging bar charts showing the number of differential ChIP-seq peaks identified by DESeq2 analysis in the indicated conditions.

Similarly to H3K27, TSA induced strong hyperacetylation on H3K9 which was restored to pre-TSA level at Western blots. ChIP-seq signal showed similar behaviour, with genome-wide H3K9ac acetylation showing only a very mild increased in the recovery condition compared to DMSO. Nevertheless, our quantitative differential ChIP-seq peak analysis (discussed in detail in Reviewer#1 major comment 2 point (i)) detected a large number of genomic intervals where H3K9ac remained significantly – albeit only very slightly – enriched. In order to test if these could be linked to memory, we plotted the peak frequency distribution of sites with increased H3K9ac relative to TSSs of genes that recover or do not recover their transcriptional states following TSA washout (Extended Data Fig. 8i).

Extended Data Fig. 8i, Differential H3K9ac ChIP-seq peak frequency distribution relative to TSSs of up- (left panels) and down-regulated (right panels) genes that recover or do not recover. Shading represents 95% confidence interval.

This analysis showed no particular association of increased H3K9ac peaks – neither in TSA nor in recovery – with differentially expressed TSSs of either gene groups. This indicates a non-specific gain in H3K9ac and argues against H3K9ac as the signal for memory.

To reflect on these new conclusions, we updated the manuscript text in lines 296-298,

“Excess H3K27ac, H3K4me1 and chromatin accessibility were restored at once (Fig. 4b) and H3K27ac peaks re-gained their enrichment around TSSs (Fig. 4c). An exception to essentially complete recovery was H3K9ac, that appeared to recover at the protein level but quantitative analysis indicated mild persistent enrichment at several genomic loci (Extended Data Fig. 5h-j).”

as well as in lines 434-436 as follows:

“H3K9ac peaks that remained enriched after the first recovery were equally enriched around TSSs that recovered and that did not recover (Extended Data Fig 8i), indicating that H3K9 acetylation is not responsible for the memory effect.”

We would like to add that we considered an additional possibility, namely that H4K16ac might be an important carrier of TSA-induced chromatin memory due to histone hyperacetylation. A very interesting study³⁴ characterized this modification and we interacted with Alex Radzsheuskaya, the first author of the work, who meanwhile established her own laboratory at the Institute for Cancer Research in London. She recently established a dTag mES cell line that allows one to efficiently deplete MSL1, the critical component specifically responsible for deposition of H4K16ac. However, she found that strong depletion of MSL1 does not induce changes in gene expression, even if it does strongly reduce global levels of H4K16Ac. We therefore did not continue this line of analysis.

Second point: *How do the authors rule out increased acetylation of non-histone proteins?*. This concern was also raised by Reviewer#1 (see Reviewer#1, major comment 4: TF targets), thus we duplicated our response below.

To understand if sustained gene expression deregulation could be due to modulated non-histone target activity induced by HDAC inhibition, we carried out motif analysis at deregulated gene promoters. Next, we crossed the list of enriched motifs with a list of characterised non-histone proteins that undergo deacetylation¹⁴, and filtered results for proteins that are targets of Class I, Class II or Class IV HDACs (those being inhibited by TSA) expressed in mESCs. Such proteins are marked with asterisk on the bubble plots below (Extended Data Fig. 6f-g).

This analysis demonstrated that while genes that recovered from TSA treatment could be clearly linked to certain non-histone HDAC targets (eg. YY1, c-Myc, Sp1), genes that remained deregulated could not. The sole exception to this is GATA3 whose targets (4 genes in total) are enriched among upregulated genes at 24 hours. This however, is unlikely to happen through the direct action of HDACs on GATA3, given that HDAC inhibition attenuates – rather than activates - GATA3-dependent gene transcription in lymphoma¹⁵.

Despite the fact that these computational analyses do not support the hypothesis that TF acetylation is linked to memory, we decided to test this idea experimentally. As YY1 was the top enriched motifs among downregulated genes that recover from TSA treatment, we performed ChIP-seq on YY1 in TSA condition (Extended Data Fig. 6e). Importantly, this showed that YY1 binding at downregulated TSSs remains virtually unchanged in TSA, making it unlikely that persistently altered YY1-chromatin interactions (as a result of modulated YY1 acetylation state¹⁶) would be responsible for long-lasting transcriptional effects.

Extended Data Figure 6f-g. Bubble plots showing HOMER motif enrichment analysis at up- (f) and downregulated (g) gene promoters of genes that recover from and genes that remain deregulated following TSA treatment. Direct Class I, II or IV HDAC targets that have detectable expression levels in mouse ESCs are marked with asterisk (*).

Transcription factor binding leads to chromatin opening both at promoter sites as well as distal *cis* regulatory elements (enhancer). Thus, instead of restricting our analysis to gene promoters, we decided to carry out motif enrichment analysis at all ATAC-seq sites that remain open following the first or second recovery periods (Extended Data Fig. 6h). Then, like above, we crossed the list of enriched motifs with a list of characterised non-histone proteins that undergo deacetylation¹⁴, and filtered results for proteins that are targets of Class I, II or IV HDACs expressed in mESCs. This revealed that sites of continually increased chromatin accessibility are not enriched in motifs of direct HDAC targets. An exception to this is Sp1, whose motif is enriched among sites that remain open in the first recovery. However, given that 1) Sp1 can function as a transcriptional repressor by recruiting HDAC1-containing complexes¹⁷ and 2) Sp1 itself is downregulated in TSA treatment, it is difficult to envisage plausible mechanisms that might lead to prolonged chromatin opening at Sp1 motifs.

Extended Data Figure 6e. Normalised YY1 ChIP-seq signal at downregulated TSSs that recover (top) and do not recover (bottom) in DMSO and TSA. **h.** Heatmap showing HOMER motif enrichment analysis at differential ATAC-seq peaks at 24-hour recovery following the first and second TSA treatments.

We of course cannot fully exclude the possibility that some continued gene expression deregulation is due to previously uncharacterised non-histone HDAC target. Nevertheless, these complex analyses strongly support the idea that the transcriptional memory induced by TSA treatment is not principally due to altered acetylation state of non-histone HDAC targets. Therefore, we updated the manuscript text in lines 313-316 as follows:

“Crucially, we did not find any link between non-histone targets of HDACs and sustained gene expression deregulation, minimising the possibility that long term transcriptional consequences would be due to off-target effects of HDAC inhibition (Extended Data Fig, 6e-h).”

Reviewer#2 major comment 2 – Changes in chromatin accessibility

Figure 1b shows H3K27ac peak density on chromosome 18, and Figure 1d show ChIP-seq data for various histone modifications on chromosome 7. Are there any specific acetylation patterns unique to these two chromosomes compared to others? Additionally, Figure 1d depicts ATAC-seq data following TSA treatment in a highly acetylated region. Chromatin accessibility does not appear to change significantly or correlate with increased acetylation change, the authors should describe and interpret these findings.

Chromosomal regions depicted on the above figure panels were chosen completely randomly. We agree with the reviewer that global increase in chromatin accessibility is not obvious on Fig. 1d at first glance, thus we produced additional plots to better illustrate TSA-induced changes in the ATAC-seq landscape.

First, ATAC-seq signal analysis at all sites of chromatin opening indicated that the most visible increase occurred at previously closed – or only slightly open – sites (Reviewer Fig. 6a). This we believe could be due to an upper limit to accessibility which is already (nearly) achieved in DMSO at sites of strong chromatin opening. Via quantitative differential peak analysis, we identified more than 100,000 ATAC-seq sites that show increased signal in TSA, supporting global chromatin opening that is nearly perfectly reversed in the recovery conditions (Reviewer Fig. 6b-c). Finally, we annotated differential ATAC-seq peaks that were located within 500bp vicinity of TSSs and performed comparative Gene Ontology enrichment analysis with differentially expressed genes. These indicated that while downregulated genes and sites of decreased ATAC-seq signal correlated well, chromatin opening occurred almost non-specifically, around genes belonging to Gene Ontology terms linked to up- and downregulated genes (Reviewer Fig. 6d).

Reviewer Figure 6a, Heatmaps showing normalised ATAC-seq signal at all enrichment sites. **b**, Diverging bar charts showing the number of differential ATAC-seq peaks identified by DESeq2 analysis in the indicated conditions. Both recovery periods were 24 hours long. **c**, Genomic snapshot of spike-in normalised ATAC-seq signal over the HoxC cluster. Black bars correspond to ATAC-seq peaks with increased signal in TSA. **d**, Gene Ontology enrichment of among up- and downregulated genes in TSA vs annotated differential ATAC-seq peaks that are within 500 bp vicinity of TSSs.

Reviewer#2 major comment 3 – Chromatin interactions at upregulated genes

The authors develop ultra-deep Micro-C contact maps for both control and TSA-treated mESCs, generating genome-wide interaction maps with unprecedented resolution. TSA treatment leads to an increase in inter-chromosomal contacts and a decrease in A-A compartment interactions. These 3D changes, along with histone modification alterations, partially correlate with gene expression dysregulation. Transcriptomic changes are associated with increased H3K27ac and concomitant H3K4me1 gain in surrounding regions, with no apparent loss of repressive marks (Figure 3). However, no promoter contact changes were observed at upregulated transcription start sites (TSSs). Beyond this correlative evidence, could the authors provide an example or the list of upregulated genes that exhibit increased interactions (measured by Micro-C, precisising which type), H3K27ac, and H3K4me1/3 modifications. Are these genes involved in ESC commitment?

Following the reviewer's suggestion, we crossed the list of genes upregulated in TSA with H3K27ac, H3K4me1 and H3K4me3 peaks that gain signal, and loops that increase in strength in TSA. Based on the number of genes that show increase in the above chromatin features we could confirm our previous observations, namely that gene expression upregulation is best correlated with increase in H3K27ac, followed by simultaneous increase in H3K27ac, H3K4me3 as well as H3K4me1 (Reviewer Fig. 7a). Contrarily, increased chromatin looping was a feature of a very small number of upregulated TSSs (16 in total, out of which 114 recover within 24 hours of TSA removal), once again supporting our conclusion that gene expression upregulation is predominantly happening without a gain in promoter contacts.

Reviewer Figure 7. a, Number of upregulated genes in TSA linked to upregulated chromatin features. **b-c**, Gene Ontology (b) and transcription factor (c) enrichment among upregulated genes linked to different upregulated chromatin features.

Next, we looked at Gene Ontology enrichment among sub-groups of upregulated genes with the different chromatin features (Reviewer Fig. 7b). This analysis revealed that genes with increased looping contribute to different biological processes than genes with increased signal of activating histone marks. Specifically, genes where loop strength increased in TSA belonged to Polycomb-related Gene Ontologies. To verify this, we also carried out transcription factor enrichment analysis (using the Encode-ChEA database) which showed that TSSs

showing increased looping are indeed strongly enriched for Ezh2 and Suz12 binding (Reviewer Fig. 7c). Thus, we were able to confirm that gene upregulation is predominantly associated with an increase in activating histone modifications around TSS. Additionally, we verify that increased chromatin looping is not a feature of most upregulated TSSs. The small (116/3912) subgroup of genes where we observe such enhanced looping are enriched for Polycomb targets, corroborating with our genome-wide analysis of fortified Polycomb domain interactions. The complete list of upregulated genes as well as the corresponding upregulated chromatin features are now listed in Supplementary Table 10.

Reviewer#2 major comment 4 – Longer recovery times

24 hours after TSA washout, H3K27ac and H3K4me1 levels return rapidly to normal level, which could be expected for very dynamic histone modifications. However, some gene expression changes, associated with 3D conformational alterations induced by TSA, persist much longer than the acetylation state including after cell division. This raises several questions about the underlying mechanisms. In these experiments the authors should clarify/discuss how long these changes persist in culture and approximately how many cell divisions are required for this memory to dissipate.

We thank the reviewer for raising this excellent point and encouraging us to look into the longer-term dynamics of recovery from the TSA pulse. We conducted two additional experiments to address this question. First, we looked at longer recovery times (48h, 72h, 96h) following the first TSA treatment. Second, we increased recovery from 24- to 48 hours between the sequential TSA treatments and assessed if memory was still induced at the level of the transcriptome. Reviewer#1 raised similar concerns regarding the recovery dynamics, thus we duplicated our response below.

Interestingly, upon performing RNA-seq in 48-, 72- and 96 hours following TSA washout, we found gene expression differences increasing – rather than dissipating – with increasing recovery times (Extended Data Fig. 6a-b). This is exemplified by an increasing distance between DMSO and consecutive recovery time points in principle component analysis (PCA), as well as the sheer number of differentially expressed genes that pass our filtering thresholds. Conversely, we also found higher number of differentially expressed genes at shorter recovery times (16 hours and 20 hours), indicating that the 24-hour time point represents the most recovered state among the tested.

Extended Data Figure 6. a, Principal component analysis (PCA) analysis showing increasing distance between DMSO and recovery (REC) samples with increasing recovery time. **b,** Diverging bar plot showing the number of differentially expressed genes at the indicated recovery times.

When we analysed the behaviour of the gene groups that recover (REC) and do not recover (nonREC) from TSA, we observed that, while recovery genes have a stable expression on the long term, non-recovery genes undergo progressive gene expression deregulation (Reviewer Fig. 3a). This provides further support to an ongoing cellular memory – that being a sustained cellular response to a transient stimulus. ChEA Transcription Factor Target enrichment among downregulated genes analysis showed that continued gene downregulation on the long term also relies on Polycomb-mediated repression (Reviewer Fig. 3b).

Reviewer Figure 3. a, Heatmaps showing mean expression z-scores of recovered and not recovered genes through the TSA-recovery treatment course. **b,** Bubble plot showing enrichment of downregulated genes that do not recover in the ChEA Transcription Factor Target database.

Gene Ontology (GO) term enrichment among deregulated genes throughout our recovery time series shows partially overlapping developmental categories (Extended Data Fig. 6c-d). Thus, we suspect that mESC initially make a near-complete transcriptional recovery, following which they undergo a pseudo-developmental cascade.

Extended Data Figure 6c-d, Bubble plots showing Gene Ontology (Biological Process) enrichment among up- (c) and downregulated (d) genes at different recovery intervals.

To visualise functional enrichment results at the different recovery time points, genes in the major developmental categories were plotted on network diagrams (Reviewer Figure 4a-b). This highlights complex associations between different days of the recovery, further arguing for an ongoing developmental cascade. Thus, rather than representing an intermittent time point during an ongoing recovery, the 24-hour time point is the closest to the DMSO cellular state.

These new critical results are now summarized in lines 307-311 of the manuscript text as follows:

“In order to exclude the possibility that the near-complete recovery is simply due to an insufficiently long time period, we assayed gene expression changes 48-, 72- and 96-hours following TSA removal. Interestingly, this revealed that the number of differentially expressed genes increases with longer recovery times as cells appear to undergo a seemingly unguided developmental cascade (Extended Data Fig, 6a-d).”

Reviewer Figure 4a-b. Network diagrams showing relationships between non-recovery (**a** – upregulated, **b** – downregulated) genes at the indicated recovery times.

Next, we sought to understand if a second perturbation had a similar effect if the first recovery period was increased from 24- to 48 hours. Thus, we carried out RNA-seq in a second TSA pulse following a 48-hour initial recovery period (48hREC-reTSA). We also monitored recovery from this second pulse at 24 and 48 hours (48hREC-24hREC, 48hREC-48hREC). We found that transcriptional response to the second TSA pulse was highly similar whether the first recovery period was 24 hours or 48 hours: this was visible from the number of differentially expressed genes as well as from the scatterplot showing high correlation between the two conditions (Extended Data Fig. 8d,f-h).

Moreover, the extent of recovery from the second TSA treatment in both cases (24hREC-24hREC, 48hREC-24hREC) was nearly identical in terms of number of differentially expressed genes (Extended Data Fig. 8d), arguing for the case that continued transcriptional changes induced by repeat exposure to TSA are stable over time. Thus, we included the following statement in lines 425-428 of the manuscript text:

“Crucially, transcriptional response and recovery was similar when the first recovery period was increased from 24- to 48-hours (Extended Data Fig. 8f-h), indicating that the cellular memory effect can persist through multiple cell generations.”

Extended Data Figure 8d, Diverging bar plot showing the number of differentially expressed genes at the indicated conditions. **f**, Principal component analysis (PCA) analysis showing increasing distance between DMSO and recovery (reREC) samples following a second TSA treatment with increasing recovery time. **g**, Heatmaps showing mean expression z-scores of recovered and not recovered genes through the TSA-recovery treatment course. **h**, Scatterplot showing correlation between transcriptomic changes induced by sequential TSA treatment following 24-hour and 48-hour recovery.

Reviewer Figure 5.a, Venn diagram showing overlap between differentially expressed gene sets between the 96-hour recovery versus 48-hour + 48-hour recovery conditions. **b-c**, Bubble plots showing unique Gene Ontology terms enriched among up- (**b**) and downregulated (**c**) genes.

Similarly to single exposure to TSA, transcriptional differences compared to the pre-TSA condition increased when recovery time was increased following the second exposure (Extended Data Fig. 8f). However, when we compare differentially expressed genes in the 96-hour recovery versus 48-hour +48-hour recovery condition, we find that ~30% and ~50% of up- and downregulated genes were unique to either condition, respectively. We then

compared Gene Ontology terms that are uniquely enriched in either condition, and we noted that the second TSA treatment modifies the developmental trajectory of cells (Reviewer. Fig. 5a-c). In sum, our data suggest that in response to transient challenges, cells initiate self-propagating responses that are reproducible, but might partly depend on the duration of the challenge on whether it is unique or repeated. Please note that we chose not to include these reviewer analyses in order to avoid adding unnecessary complexity that might blur, rather than sharpen, the message of the manuscript. However, readers that would like to delve into details will easily be able to perform these and other analyses thanks to the extensive raw data, extended and supplementary information provided along with the manuscript.

Reviewer#2 major comment 5 – Clonal transmission of epigenetic memory

To support the claim that cellular memory is changed by TSA repeat treatments the authors must verify the properties of clonally-derived ESCs after the first treatment, and subsequently after the second (If the property is inherited/stable as claimed, then cells can be expanded at each step and the analysed). This is necessary to confirm, at the level of individual cells, that epigenetic memory has been revised.

We thank the reviewer for this insightful comment. While we tried to address most points experimentally, we deemed this suggestion both unfeasible as well as to extend beyond the scope of this study due to the reasons explained below.

First, as discussed in the 'Remarks to the Authors' section, we define cellular memory as a 'sustained response to a transient stimulus'. We believe that our population-based assays are suitable to measure cellular memory as we show that 1) TSA treatment induced hyperacetylation – that is, the initial causative trigger - is efficiently reversed at the protein level as well as genome wide and that 2) it nevertheless leads to a sustained transcriptional response that lasts at least 96 hours. As our TSA treatment is only 4-hours, the 24-hour time point already represents a relatively long recovery time, lasting 6 times the duration of the treatment pulse, while 96 recovery hours represents 24 times the length of the perturbation. Additionally, time-resolved Western blot analysis showed that H3K27 acetylation is restored as quickly as within 3 hours of drug removal, indicating that when cells are assayed for cellular memory, the stimulus has been removed for >20 hours (Extended Data Fig. 5a). Finally, cell generation tracing has shown that the entire TSA treated cell population divides during the recovery period (see detailed response under Reviewer#2 major comment 6), excluding the possibility that lasting alterations in gene expression and chromatin folding would be due to a subpopulation of cells stranded in the TSA state and unable to divide. Thus, if anything, population-based assays would dilute the memory effect if only a subpopulation of cells would exhibit such a response.

Extended Data Figure 5a - Western blots showing the levels of H3K27 acetylation DMSO, TSA and varying time intervals following TSA removal. Lamin B1 is shown as loading control.

Secondly, although we agree with the reviewer that clonal assessment of cellular memory would allow us to address interesting questions regarding the responsiveness of individual cells, such an approach would be technically unfeasible within the time scales of this study. As the growth rate of mESCs slows down when they are seeded to low density, amplification of individual cells to sizeable colonies in 96-well plates takes about 3 weeks (Cavalli lab, personal communication). Additionally, this should be followed by at least 7-10 days of further amplification to obtain sufficient number of cells to perform genomics experiments assaying cellular states (RNA-seq, ATAC-seq, Micro-C). Thus, we estimate at least about 4 weeks between TSA treatment and clonal assessment which goes beyond the time scales assessed by this study. Eventually, due to the vast recovery duration, we would not be able to distinguish the possibility that a metastable epigenetic memory was established in certain clones but then reversed, from the possibility that memory was not established at the clonal level.

Reviewer#2 major comment 6 – 3D genome through cell division

The authors should confirm and extend their view that local 3D genome changes at deregulated genes are sustained when cells divide. For example, by using a cell labelling method (such as CFSE) where cells at successive rounds of proliferation can be examined. Providing such evidence would significantly strengthen the claims of this manuscript.

We would like to thank the reviewer for this excellent suggestion, which we have now addressed. As CFSE inhibited cell proliferation in mESCs (data not shown), we opted for using the low toxicity CellTrace Violet dye. First, we stained cells (0h time point) and let them re-attach to the cell culture dish. The next day, we treated them either with DMSO or TSA between 20- and 24 hours from the staining. At 24 hours, we also harvested a control plate that showed that cells divided since the staining and that cell division is not inhibited by the cell tracer dye. Finally, we harvested DMSO- and TSA-treated cells 48 hours after the staining which corresponds to 24 hours of recovery from the DMSO and TSA treatments. This indicated that all cells divided during the 24-hour recovery period that followed the DMSO or TSA treatments (Extended Data Fig. 5b). This result is of critical importance, as it shows that alterations in 3D architecture observed in the recovery condition must have survived at least one round of cell division and are not due to the presence of a small number of cells stranded in the TSA state.

Extended Data Figure 5b, Cell generation tracing using flow cytometry following DMSO and TSA treatment. Two biological replicates are shown.

These results are now described in lines 291-293 of the manuscript texts:

“Cell generation tracing showed that all cells have divided during the recovery period (Extended Data Fig. 5b), consistent with the lack of cell cycle perturbation induced by the TSA-recovery treatment course (Extended Data Fig. 1d-e).”

Minor comments

Extended Data Figure 1: The authors should include data for H3K4me1 downregulation and H2AK119ub downregulation, as these are currently missing.

This figure panel has been updated with differential ChIP-seq peaks from our revised quantitative analysis (Extended Data Fig. 1h). H2AK119Ub downregulation is not shown as no such peaks were found. This is now stated in the figure legend.

Extended Data Figure 1h, Bar plots showing the distance of differential ChIP-seq peaks (H3K27ac, H3K4me3, H3K4me1, H3K27me3, H2AK119Ub, H3K9me3) from transcription start sites (TSS). For H2AK119Ub no decreased peaks were found.

Reviewers' references

1. Zhang, Y. *et al.* HDAC-6 interacts with and deacetylates tubulin and microtubules in vivo. *EMBO Journal* **22**, 1168–1179 (2003).
2. Hu, B. *et al.* Biological chromodynamics: A general method for measuring protein occupancy across the genome by calibrating ChIP-seq. *Nucleic Acids Res* **43**, (2015).
3. Steinhauser, S., Kurzawa, N., Eils, R. & Herrmann, C. A comprehensive comparison of tools for differential ChIP-seq analysis. *Brief Bioinform* **17**, 953–966 (2016).
4. Nakato, R. & Sakata, T. Methods for ChIP-seq analysis: A practical workflow and advanced applications. *Methods* **187**, 44–53 (2021).
5. Eder, T. & Grebien, F. Comprehensive assessment of differential ChIP-seq tools guides optimal algorithm selection. *Genome Biol* **23**, 1–27 (2022).
6. Stark, R. & Brown, G. DiffBind: Differential binding analysis of ChIPSeq peak data. Preprint at <https://doi.org/10.18129/B9.bioc.DiffBind> (2011).
7. Zhao, J. *et al.* H2AK119ub1 differentially fine-tunes gene expression by modulating canonical PRC1- and H1-dependent chromatin compaction. *Mol Cell* **84**, 1191–1205.e7 (2024).
8. Li, T. *et al.* Non-canonical PRC1.1 licenses transcriptional response to enable Treg plasticity in immune adaptation. *Mol Cell* 1–18 (2025)
doi:10.1016/j.molcel.2025.05.029.
9. Loubiere, V., Papadopoulos, G. L., Szabo, Q., Martinez, A. M. & Cavalli, G. Widespread activation of developmental gene expression characterized by PRC1-dependent chromatin looping. *Sci Adv* **6**, eaax4001 (2020).
10. Paldi, F. & Cavalli, G. 3D genome folding in epigenetic regulation and cellular memory. *Trends in Cell Biology* vol. xx 1–14 Preprint at <https://doi.org/10.1016/j.tcb.2025.03.001> (2025).
11. Boyle, S. *et al.* A central role for canonical PRC1 in shaping the 3D nuclear landscape. *Genes Dev* **34**, 931–949 (2020).
12. Dimitrova, E. *et al.* Distinct roles for CKM–Mediator in controlling Polycomb-dependent chromosomal interactions and priming genes for induction. *Nat Struct Mol Biol* **29**, 1000–1010 (2022).
13. Rosati, D. *et al.* Differential gene expression analysis pipelines and bioinformatic tools for the identification of specific biomarkers: A review. *Comput Struct Biotechnol J* **23**, 1154–1168 (2024).
14. Narita, T., Weinert, B. T. & Choudhary, C. Functions and mechanisms of non-histone protein acetylation. *Nat Rev Mol Cell Biol* **20**, 156–174 (2019).
15. Geng, X. *et al.* GATA-3–dependent Gene Transcription is Impaired upon HDAC Inhibition. *Clinical Cancer Research* **30**, 1054–1066 (2024).
16. Yao, Y.-L., Yang, W.-M. & Seto, E. Regulation of Transcription Factor YY1 by Acetylation and Deacetylation. *Mol Cell Biol* **21**, 5979–5991 (2001).
17. Doetzlhofer, A. *et al.* Histone Deacetylase 1 Can Repress Transcription by Binding to Sp1. *Mol Cell Biol* **19**, 5504–5511 (1999).
18. Ho, T. C. S., Chan, A. H. Y. & Ganesan, A. Thirty Years of HDAC Inhibitors: 2020 Insight and Hindsight. *J Med Chem* **63**, 12460–12484 (2020).

19. Shvedunova, M. & Akhtar, A. Modulation of cellular processes by histone and non- histone protein acetylation. **23**, (2022).
20. Lagger, G. *et al.* Essential function of histone deacetylase 1 in proliferation control and CDK inhibitor repression. *EMBO Journal* **21**, 2672–2681 (2002).
21. Dovey, O. M., Foster, C. T. & Cowley, S. M. Histone deacetylase 1 (HDAC1), but not HDAC2, controls embryonic stem cell differentiation. *Proc Natl Acad Sci U S A* **107**, 8242–8247 (2010).
22. Jamaladdin, S. *et al.* Histone deacetylase (HDAC) 1 and 2 are essential for accurate cell division and the pluripotency of embryonic stem cells. *Proc Natl Acad Sci U S A* **111**, 9840–9845 (2014).
23. Lv, W., Guo, X., Wang, G., Xu, Y. & Kang, J. Histone deacetylase 1 and 3 regulate the mesodermal lineage commitment of mouse embryonic stem cells. *PLoS One* **9**, 1–19 (2014).
24. Fursova, N. A. *et al.* Synergy between Variant PRC1 Complexes Defines Polycomb-Mediated Gene Repression. *Mol Cell* **74**, 1020-1036.e8 (2019).
25. Schoenfelder, S. *et al.* Polycomb repressive complex PRC1 spatially constrains the mouse embryonic stem cell genome. *Nat Genet* **47**, 1179–1186 (2015).
26. Cruz-Molina, S. *et al.* PRC2 Facilitates the Regulatory Topology Required for Poised Enhancer Function during Pluripotent Stem Cell Differentiation. *Cell Stem Cell* **20**, 689–705 (2017).
27. Mas, G. *et al.* Promoter bivalency favors an open chromatin architecture in embryonic stem cells. *Nat Genet* **50**, 1452–1462 (2018).
28. Kraft, K. *et al.* Polycomb-mediated Genome Architecture Enables Long-range Spreading of H3K27 methylation. doi:10.1101/2020.07.27.223438.
29. Murphy, S. E. & Boettiger, A. N. Polycomb repression of Hox genes involves spatial feedback but not domain compaction or phase transition. *Nat Genet* **56**, 493–504 (2024).
30. Cavalli, G. & Heard, E. Advances in epigenetics link genetics to the environment and disease. *Nature* doi:10.1038/s41586-019-1411-0.
31. Henikoff, S. & Grealley, J. M. *Epigenetics, Cellular Memory and Gene Regulation*. (2016) doi:10.1016/j.cub.2016.06.011.
32. Burrill, D. R. & Silver, P. A. Making Cellular Memories. *Cell* **140**, 13–18 (2010).
33. D’Urso, A. & Brickner, J. H. Mechanisms of epigenetic memory. *Trends in Genetics* vol. 30 230–236 Preprint at <https://doi.org/10.1016/j.tig.2014.04.004> (2014).
34. Radzisceuskaya, A. *et al.* Complex-dependent histone acetyltransferase activity of KAT8 determines its role in transcription and cellular homeostasis. *Mol Cell* **81**, 1749-1765.e8 (2021).

Response to reviewers Nature Genetics manuscript NG-A67123

Reviewer #1:

Remarks to the Author:

The revised manuscript by Paldi and colleagues is greatly strengthened. The authors have clearly gone to considerable lengths to address the feedback and concerns, adding a wide array of new experiments and analyses. The inclusion of extensive RNA-seq, ChIP-seq, ATAC-seq, and Micro-C datasets, as well as controls with a selective HDAC inhibitor and analyses in neural progenitor cells and gastruloids, demonstrates commendable rigor. I want to emphasise that it is not just the scale of new data – which is impressive - but the thoughtful design and implementation that greatly advances conclusions. The additions clarify several key points: first, that TSA-induced changes largely recover at the level of histone acetylation, with quantitative ChIP-seq and Western blotting showing rapid restoration within hours; second, that Polycomb domain interactions underlie architecture-based memory, supported by Ring1B ChIP-seq and PCGF2/PCGF4 depletion; and third, that transcriptional and architectural features of memory persist through confirmed cell division. Together with other clarifications, these represent a significant addition, providing mechanistic insight into how transient hyperacetylation pulses can leave a lasting mark on genome architecture and gene regulation.

We thank the reviewer for the appreciation of our work as well as for their critical assessment and insightful comments that allowed us to greatly improve our manuscript.

Overall, the study is a conceptually important advance into how gene regulation is modulated, and its dynamic responses over time. Importantly, the revised version is better controlled and the conclusions more broadly supported. I therefore believe this study provides valuable insight for the community and should be considered a strong candidate for publication. At the same time, some issues remain that the authors may wish to comment on or address. For example, although Polycomb looping emerges as a central player, the contribution of other factors such as transcription factor perturbation or non-histone acetylation cannot be fully excluded. This remains a relatively unclear possibility in the manuscript in current format.

We have now emphasised on this point by adding “Although we cannot fully exclude it, this is minimising the possibility that long-term transcriptional consequences would be due to pleiotropic effects of HDAC inhibition (Extended Data Fig, 6e-h).” to line 221 of the main text.

Moreover, the authors have explained several incongruent results through H2AK119ub potentially acting to promote gene activation (rather than canonical repression) yet none of this is cited or discussed in the manuscript.

We thank the reviewer for pointing this out. We added the following sentence to line 180-182 of the manuscript text: “Sites of increased H2AK119Ub signal were equally found around up- and downregulated TSS, consistent with its recent implication in gene repression as well as activation.”

Perhaps more importantly, the new observation of increasing gene dysregulation over recovery time is to me unexpected and therefore interesting. However, it is not clear how that fits in with the original hypothesis/conclusions, nor that the explanation of a 'psuedo-developmental cascade' fits the paradigm. If there are increasing baseline changes over recovery time, then it follows that the response to a second perturbation may not be 'memory' but an exaggeration of the 'cascade;' that is ongoing. This feels quite important to clarify. Finally, the functional consequences of this memory mechanism for cell fate and physiology, beyond the molecular and architectural signatures described, remain only partially explored.

We agree with the reviewer that it remains a possibility that some of the changes observed after the second perturbation could be due to increasing gene expression deregulation with increasing recovery time. However, we detect more differentially expressed genes (767 versus 378) at 24 hours after the second TSA pulse (reREC) than 48 hours after a single TSA pulse (48hREC), indicating that the second TSA pulse is amplifying the memory effect. We agree that it is indeed important to clarify this, thus we added the following sentence to lines 278-279 of the manuscript text: "We note that this might be partially due to increasing gene expression deregulation with longer times following the first TSA pulse (Extended Data Fig. 6b)."

In sum, the revisions markedly enhance the manuscript and addresses key concerns. The study offers a new angle on the regulation of epigenetic memory, while leaving open interesting questions about mechanistic exclusivity and biological relevance that could inspire future work.

We would like to thank to reviewer again for their careful assessment of our work. We are pleased that the reviewer finds the work significantly improved and of broad interest.

Reviewer #2:

Remarks to the Author:

In this revised manuscript, the authors performed new experiments that reinforce their initial finding of the persistence of a cellular memory induced by transient inhibition of histone deacetylase. These additional experiments provide deeper insight into the molecular changes involved, including analyses of other acetylation targets and examination of longer recovery times following the initial pulse of the epigenetic perturbation. Notably, in this revised version, the authors have addressed the 3D genome maintenance through cell divisions and discussed the clonal transmission of cellular memory.

The authors have meticulously addressed most of the concerns raised in the first submission. They have included significant new set of data and analyses, along with improved clarification of new results and provided a more precise definition of cellular memory.

This study offers new insights into the mechanistic contributions of different epigenetic layers (namely H3K27ac, H3K4me2, and H3K27me3) and genome folding in regulating transcription in ESCs, and how this impact transcriptional memory.

Remarks on code availability:

These look OK.

We thank the reviewer for the positive assessment of our revised manuscript. We would also like to express our gratitude to the reviewer for the critical evaluation of our work and their perceptive comments which allowed us to significantly improve our manuscript.

Additional remarks from authors

As part of the publication process we have now conducted scrupulous verification of all data analysis, processing and editing steps. During this extensive re-assessment of our work we have discovered that some statistical comparisons were conducted under unmet assumptions. Therefore, methodologically inappropriate comparisons were removed in Figure 7h, Extended Data Fig. 5d and Extended Data Fig. 9g as the magnitude and pattern of the differences are readily interpretable from the data itself. Additionally, in order to comply with journal rules we were asked to remove statistical comparisons from Extended Data Fig. 5c, the experiment being performed in biological duplicates.

Furthermore, we found minor mistakes in three figure panels. We have now updated these figure panels in the final manuscript version. The old and new panels are displayed below, on the left (old) and right (new), respectively and a brief explanation is provided below each panel. We sincerely apologize for these mistakes and we note that the minor changes in the new version do not affect results or conclusions in any way.

Figure 4e - H3K27ac up peak count frequency distribution in REC relative to gene bodies that remain up- and down-regulated.

In the old figure, the sparse nature of the data over a small genomic region hindered assessment of randomness relative to the TSSs. The new panel clearly indicates a lack of strong signal accumulation specific to TSSs.

Extended Data Fig. 5f - Western blots showing total level of histone modifications in nuclear extracts. Lamin B is shown as loading control.

The old top panel of Fig. 5f displayed the loading control instead of the H2AK119Ub immunoblot. This is now corrected.

Extended Data Fig. 9d - Aggregate plots of Micro-C signal around loops where anchors overlap with downregulated TSSs that recover (left panels) or do not recover (right panels) (resolution = 4 kb). Quantification of piled-up loop signal ($n = 9$ corresponding to central 3x3 pixels) is shown on the right (paired two-tailed t-test; ns > 0.05, *** $p < 0.001$). Data shown are the median, with hinges corresponding to IQR and whiskers extending to the lowest and highest values within 1.5x IQR.

Old boxplots showed quantification of the central 5x5 pixels. In order to remain consistent with other loop quantifications presented in the manuscript, new boxplots show the values of the central 3x3 pixels while leaving the outcome of statistical comparisons unchanged.